# Procedural Generation of Algorithm Discovery Tasks in Machine Learning

Alexander D. Goldie [* 1]   Zilin Wang [† 1]   Adrian Hayler [† 1]   Deepak Nathani [† 2]   Edan Toledo [† 3]
Ken Thampiratwong [† 2]   Aleksandra Kalisz [† 1]   Michael Beukman [‡ 1]   Alistair Letcher [‡ 1]
Shashank Reddy Chirra [‡ 1]   Clarisse Wibault [‡ 1]   Theo Wolf [‡ 1]   Charles O'Neill [‡ 1]   Uljad Berdica [‡ 1]
Nicholas Roberts [‡ 4]   Saeed Rahmani [‡ 1 5]   Roberta Raileanu [§ 3]   Shimon Whiteson [§ 1]   Jakob N. Foerster [§ 1]

## Abstract

Automating the development of machine learning algorithms has the potential to unlock new breakthroughs. However, our ability to *improve* and *evaluate* algorithm discovery systems has thus far been limited by existing task suites. They suffer from many issues, such as: poor evaluation methodologies; data contamination; and containing saturated or very similar problems. Here, we introduce *DiscoGen*, a procedural generator of algorithm discovery tasks for machine learning, such as developing optimisers for reinforcement learning or loss functions for image classification. Motivated by the success of procedural generation in reinforcement learning, DiscoGen spans billions of tasks of varying difficulty and complexity from a range of machine learning fields. These tasks are specified by a small number of configuration parameters and can be used to optimise algorithm discovery agents (ADAs). We present *DiscoBench*, a fixed, small subset of DiscoGen tasks for principled evaluation of ADAs. Finally, we propose a number of ambitious, impactful research directions enabled by DiscoGen, and demonstrate its use for ADA optimisation through scaling experiments for automated prompt tuning. DiscoGen is released open-source.

## 1. Introduction

Automating the development of machine learning (ML) algorithms with AI offers the potential to unlock new breakthroughs in research. Furthermore, since algorithm discov-

ery agents (ADAs) can alleviate the bottleneck of human ideation, implementation and experimentation, their utility scales directly with computational resources.

However, existing ADA benchmarks (e.g., MLE-Bench (Chan et al., 2025) and MLGym-Bench (Nathani et al., 2025)) suffer from structural problems that inhibit principled evaluation. They generally fail to separate the *discovery* (meta-train) and *evaluation* (meta-test) of algorithms, meaning ADAs discover algorithms for the same problems they are evaluated on. Additionally, they often require agents to write entire codebases, effectively measuring software engineering rather than research skills, or initialise from full file systems, biasing agents away from discovering novelty (Nathani et al., 2025). Finally, these benchmarks run the risk of data contamination from pre-training (Dong et al., 2024; Liang et al., 2025); ADAs may have learned from the fixed task sets during pre-training, and thus change their behaviour or use previously seen problem solutions to compensate for poor research skills (Liang et al., 2025).

Furthermore, our ability to develop better ADAs for ML remains limited, principally because there are *too few* different algorithm discovery problems to learn from. These existing suites of tasks for algorithm discovery are constrained due to a reliance on manual creation. Therefore, developing new approaches and architectures for existing suites, or training ADAs on them, risks overfitting.

To address these issues, we introduce DiscoGen, a procedural generator of algorithm discovery tasks for ML. DiscoGen supports *billions* of different tasks, of varying difficulty. DiscoGen tasks have distinct meta-train/meta-test datasets, where meta-test datasets are hidden from the ADA, ensuring principled evaluation. Furthermore, DiscoGen supports many diverse ML subfields and uses a modular structure that defines *which components* of an algorithm an ADA discovers, meaning tasks vary over a number of axes.

DiscoGen enables the use of an *ADA optimisation loop*, as shown in Figure 1. Thus, we establish our terminology as:

- **Inner-loop:** An algorithm optimises a specific model on a single dataset's train set. When the inner-loop finishes, the model is evaluated on the dataset's test set. For example,

*Lead Author, †Core Contributor, ‡Task Contributor, §Equal Supervision [1] University of Oxford [2] University of California, Santa Barbara [3] University College London [4] University of Wisconsin–Madison [5] Delft University of Technology. Correspondence to: Alexander D. Goldie <goldie@robots.ox.ac.uk>.

*Proceedings of the 43rd International Conference on Machine Learning*, Seoul, South Korea. PMLR 306, 2026. Copyright 2026 by the author(s).

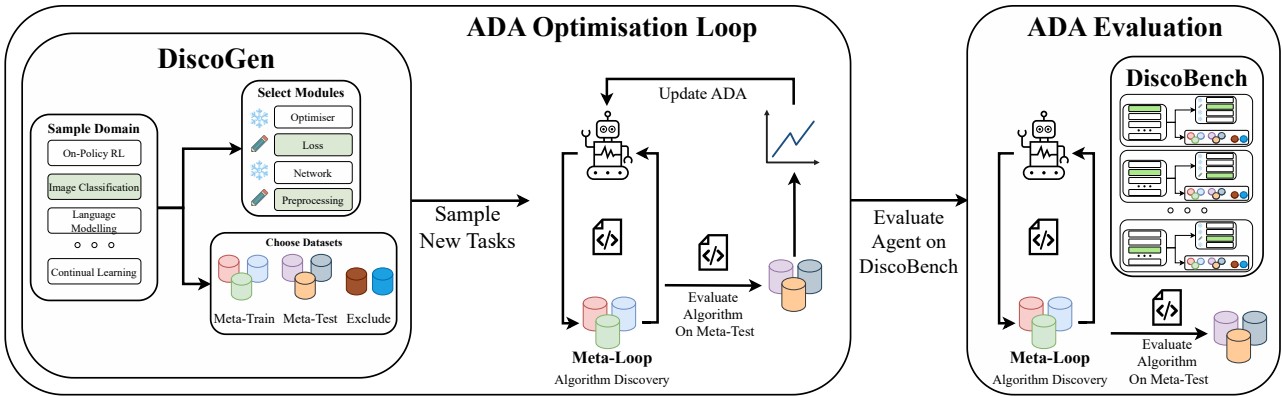

*Figure 1*. A typical DiscoGen setup. DiscoGen procedurally generates new algorithm discovery tasks. For every generated task, an algorithm discovery agent iteratively develops new algorithms (the meta-loop) for training in the task's meta-train datasets (the inner-loops). The developed algorithm is evaluated on meta-test datasets, with the evaluation score used to optimise the agent (the ADA optimisation-loop). Datasets that are available in a task domain can also be excluded from the task. After each step, DiscoGen can be sampled for a new task. After ADA optimisation has completed, the agent is evaluated on DiscoBench, a set of ADA test tasks.

the inner loop could be training an image classifier on the ImageNet train set (Russakovsky et al., 2015), and evaluating it on the ImageNet test set.

- **Meta-loop:** The ADA iteratively improves the algorithm based on inner-loop feedback from its *meta-train* set, which contains many inner-loop datasets. When the meta-loop finishes, the final algorithm is evaluated on a held-out meta-test set. An example meta-loop could involve an ADA developing image classifier loss functions, based on feedback from ImageNet and CIFAR-10 (Krizhevsky et al., 2009) (meta-train), and evaluating the loss by training an image classifier on CIFAR-100 (meta-test).
- **Task:** A single algorithm discovery problem, defining the ADA's objective and the meta-train and meta-test datasets.
- **ADA Optimisation loop:** The ADA is optimised for meta-loop performance in DiscoGen tasks in a *meta-meta*-loop (Schmidhuber, 1987). The ADA optimisation loop produces an ADA for evaluation. ADA optimisation could update ADA weights based on meta-test performance.
- **ADA Evaluation**: The ADA is evaluated on a set of tasks that it has not been directly optimised for (in other words, *meta-meta*-test). This could involve developing a language model optimiser, for instance.

Much as procedural environment generation enabled training of general reinforcement learning agents (Cobbe et al., 2020; Stooke et al., 2021; Bauer et al., 2023; Matthews et al., 2025), DiscoGen enables new research directions in algorithm discovery. Ideas like autocurricula (Leibo et al., 2019; Dennis et al., 2020; Parker-Holder et al., 2022a) or recursive self-improvement (Clune, 2019) *rely* on an ability to sample new, interesting tasks of varying difficulty to prevent overfitting. Given the number of possible tasks in DiscoGen, it is a crucial tool for enabling open-ended learning for algorithm discovery (Stanley, 2019; Hughes et al., 2024).

We create DiscoBench, a set of hand-designed tasks from DiscoGen, for ADA evaluation. Similar to Matthews et al.

(2025) or Samvelyan et al. (2021), which build task suites from procedural environment generators, DiscoBench lies in the support of DiscoGen but should *not* be intentionally optimised over. While DiscoBench, like other task sets, is susceptible to data contamination, it benefits from the design of DiscoGen, such as a meta-train/meta-test distinction. Since our ADA evaluation was run prior to publication, it is not subject to this contamination, and we propose mitigations in Section 9 to overcome this in the future.

We provide further research proposals enabled by DiscoGen in Section 6. Finally, as an example of ADA optimisation, we use procedurally generated tasks from DiscoGen for prompt tuning an ADA in Section 7. We explore how ADA evaluation changes when their prompts are optimised over different numbers of *randomly generated* tasks, finding that prompts developed over a wider range perform better; both in-support, and for completely held out domains.

## 2. Related Work

Due to the depth of prior work in this field, we abridge our related work discussion here and expand it in Appendix B.

### 2.1. Automated Research

Automated machine learning (Hutter et al., 2019, AutoML) focuses on applying machine learning to new problems without expert knowledge. Prior research generally augments machine learning algorithms with methods like hyperparameter tuning (e.g., (Li et al., 2017; Parker-Holder et al., 2020)) or data-cleaning (e.g., (Krishnan et al., 2016)). However, whereas most AutoML research is limited to fitting human-designed solutions to new data, algorithm discovery has the inverse goal: autonomously developing new algorithms.

That said, meta-learning is a subset of AutoML which aims to *learn* algorithms from data (Schmidhuber, 1987; Real et al., 2020; Beck et al., 2025). Often, meta-learned algorithms train a neural network to replace a component

*Table 1.* Overview of selected desirable properties for a number of existing algorithm discovery tasks. ● denotes partial satisfaction.

| Benchmark | # Tasks | Task Property | | | | |
|---|---|---|---|---|---|---|
| | | Meta-Test | Flexible Init. | Low Contam. Risk | Tunable Diff. | Editable Evals |
| MLGym-Bench | 13 | ✗ | ✗ | ✗ | ✗ | ✗ |
| MLAgentBench | 13 | ✗ | ✗ | ✗ | ✗ | ✗ |
| MLE-Bench | 75 | ✗ | ✗ | ✗ | ● | ✗ |
| AIRSBench | 20 | ✗ | ✗ | ● | ● | ✗ |
| REBench | 7 | ✗ | ✗ | ● | ● | ✗ |
| **DiscoGen (Ours)** | **∼100B** | ☑ | ☑ | ● | ☑ | ☑ |

of a machine learning algorithm, such as the optimiser (Andrychowicz et al., 2016; Metz et al., 2022b; Goldie et al., 2024) or loss function (Kirsch et al., 2020; Bechtle et al., 2021). Recently, using large language models (LLMs) to propose new algorithms has proven fruitful (Lu et al., 2024; Romera-Paredes et al., 2024; Novikov et al., 2025). However, optimising and evaluating these systems is difficult due to a lack of diverse, interesting and well-designed tasks. In this paper, we consider how procedural generation can be used to create new algorithm discovery tasks to this end.

As language models have improved (METR, 2025; Chollet et al., 2025), developing more complex research and coding *agents* has emerged as an important pursuit. Agents augment language models with the ability to take actions, use tools and run code (Wang et al., 2024a; Schick et al., 2023; Yang et al., 2024). AI research agents (Toledo et al., 2025) use these tools to automate parts of the research process in a ReAct loop (Yao et al., 2023), where they receive feedback from tools while developing solutions. Rather than focusing on designing new agents, we build a framework for sampling millions of tasks to aid their development.

Agents can be applied throughout the research pipeline. Examples include automating ideation (Si et al., 2025), implementing research ideas with a human-in-the-loop (Gottweis et al., 2025; Weston & Foerster, 2025), judging research papers (Si et al., 2025; Thakkar et al., 2025), or automating the entire research workflow (Lu et al., 2026; Yamada et al., 2025; Intology, 2025; Si et al., 2026). Specifically in algorithm discovery, considerations include how agents should search over new algorithms (Jiang et al., 2025; Toledo et al., 2025) or when to run experiments (Yu et al., 2025; Nathani et al., 2025). In this work, we provide a rigorous and scalable generator of tasks for optimising and evaluating ADAs.

### 2.2. Optimising Agents

Pretraining language models on more data leads to improved performance (Kaplan et al., 2020), and large procedurally generated environments have led to generalist reinforcement learning (RL) policies (Section 2.3). Motivated by these findings, we consider how ADAs can be optimised using procedurally generated algorithm discovery tasks.

Optimising agents specifically for mathematics (e.g., (Lewkowycz et al., 2022; Trinh et al., 2024; Hubert et al.,

2025)) or coding (e.g., (Li et al., 2022; Rozière et al., 2024)) has led to significant gains. Underpinning these advances are large, diverse and verifiable problem sets to research for or train models on (Shao et al., 2024; Wen et al., 2026). For example, there are many suites of mathematical problems (e.g., (Cobbe et al., 2021; Hendrycks et al., 2021b)) or open-source code repositories (Jimenez et al., 2024; Chen et al., 2021). However, developing similarly large sets of algorithm discovery tasks has proven difficult, as manual curation requires expert knowledge and, often, adaptation to integrate different data. Additionally, developing superhuman algorithms requires measuring '*how good*' algorithms are, rather than correctness. Here, we mitigate these limitations of manual task creation using procedural generation.

### 2.3. Procedural Generation

Procedural content generation (PCG) involves creating levels or environments algorithmically, according to rules, rather than manually (Togelius et al., 2013). To do so, PCG environments are defined as Contextual Markov Decision Processes (Hallak et al., 2015, Contextual MDPs) or Underspecified Partially Observable MDPs (Dennis et al., 2020) which define levels by a small number of configuration variables. In deep RL, PCG has proven effective for training agents to generalise over a smooth *distribution* of levels, rather than solving specific levels only. Such approaches have been applied to environments of ranging complexity, from gridworlds (Chevalier-Boisvert et al., 2023) to physics engines (Matthews et al., 2025) or 3-dimensional worlds (Stooke et al., 2021). Procedural generation enables new research directions, such as autocurricula (e.g., (Jiang et al., 2021b; Dennis et al., 2020)) or large scale meta-learning (e.g., (Bauer et al., 2023; Nikulin et al., 2024)). We explore how to apply similar principles to algorithm discovery.

### 3. The Problems With ADA Benchmarks

ADA improvement is bottlenecked by current algorithm discovery benchmarks. They are heavily limited in scale due to a reliance on manual task creation. Task suites like MLGym-Bench (Nathani et al., 2025, 13 tasks), MLAgent-Bench (Huang et al., 2024, 13 tasks), MLE-Bench (Chan et al., 2025, 75 tasks), AIRSBench (Lupidi et al., 2026, 20 tasks) and REBench (Wijk et al., 2025, 7 tasks) assess ADA performance, but none provide enough tasks to optimise

ADAs over, nor separate ADA evaluation from optimisation. Ideally, as in PCG for RL, we want to optimise ADAs over a smooth *distribution* of tasks to robustly develop algorithm discovery capabilities. Furthermore, these benchmarks suffer from flaws which limit their value for evaluation.

### 3.1. Issues With Existing Task Designs

Beyond limited scope, we believe task design in these benchmarks is insufficient. Here, we discuss a number of their structural flaws, which DiscoGen rectifies (Section 4.4).

**Poor Evaluation**  Proper evaluation in machine learning relies on a distinct train/test split (Goodfellow et al., 2016) to avoid overestimating performance. Algorithm discovery is no different; despite not fitting a *model* to test data in the *inner-loop*, hill-climbing algorithms on meta-train datasets is susceptible to the same flawed evaluation as humans using validation signals to design methods (Langford, 2005; Whiteson et al., 2011). However, existing benchmarks miss the proper train-test boundary. Rather than measuring algorithm transfer from *meta*-train to unseen *meta*-test datasets, they evaluate the performance of inner-loop trained models on the test set of the (known) meta-train datasets (e.g., (Nathani et al., 2025; Chan et al., 2025)). In effect, testing an algorithm on the dataset it was developed on, rather than how well it generalises. Whilst this can be valid evaluation in certain settings, it is not generally the objective in algorithm discovery (Goldie et al., 2025). We demonstrate the importance of meta-test evaluation in Section 5.2, where algorithms often perform worse in meta-test over meta-train.

**Limited Diversity**  Due to their limited scale, benchmarks are often restricted to similar classes of problems, such as small Kaggle-style challenges (Chan et al., 2025) or quick-to-run problems (Nathani et al., 2025). As such, rather than understanding the general performance of ADAs, they measure their ability in specific *types* of problems only.

**Limiting Initialisation**  While in-context examples help elicit reasoning in LLMs (Wei et al., 2022), they can limit output diversity (Turpin et al., 2023). Many benchmarks only initialise tasks from full implementations, potentially limiting the creativity of agents; in Nathani et al. (2025), agents devolve to hyperparameter tuning. Similarly, starting from empty files can sometimes prove too difficult for agents, limiting the potential optimisation signal.

**Slow Manual Expansion**  Adding *every* new task to these suites is manual, meaning scaling their quantity is inherently limited by the number of human-hours available.

**Data Contamination**  Data contamination, where evaluation data leaks into training, is an issue in LLM benchmarks due to large-scale pretraining (Dong et al., 2024). It can be especially problematic in 'challenge-based' benchmarks, like MLE-Bench. Since agents can often use the internet, or may have pre-trained on challenges, ADAs can reproduce public solutions to boost their score (Hamin & Edelman,

2025). Such logic can be extended to other contamination types, like agents seeing open-sourced dataset labels. While limiting internet access is a mitigation (e.g., (Yang et al., 2026)), it is significantly more advantageous to design benchmarks that are as robust as possible instead, such that we evaluate a true agent's capabilities.

**Floor and Ceiling Effects**  Many suites include *saturated* or *solved* problems (e.g., (Huang et al., 2024)) with hard-to-change difficulties, limiting their signal for optimisation.

## 4. DiscoGen

DiscoGen generates **tasks**: modular algorithm discovery problems consisting of an objective and meta-train and meta-test datasets. Each task is defined by seven components.

**1. Task Domain**  The task domain is the area of machine learning a task pertains to (e.g., *Image Classification*). It defines the initial codebase, performance metrics, and which datasets and modules are available.

**2. Editable Modules**  For each domain, we identify important building blocks, or *modules*, of an algorithm that can be set as *editable* or *fixed*. Only *editable* modules are to be discovered by the agent; *fixed* modules use standard implementations (e.g., a baseline optimiser or loss). There can be many modules per domain, combinatorially expanding the number of possible tasks that can be generated.

**3. Meta-Training Datasets**  Meta-training datasets are the problems which an agent can run experiments on during algorithm discovery. They are known to the agent.

**4. Meta-Test Datasets**  Meta-test datasets are used to evaluate an algorithm *after* the meta-loop completes. They measure algorithm-transfer to held-out problems, and are unknown to the ADA.

**5. Backend**  Some task domains include extra *backends* that expand the task space. For example, in On-Policy RL, tasks can require either a feed-forward or recurrent policy.

**6. Evaluation Type**  DiscoGen supports different meta-objectives: final score (maximise performance); energy, in kWh, to reach a proportion of the baseline's score (maximise efficiency); or time taken, in seconds, to reach a proportion of the baseline's score (maximise speed, like Zhao et al. (2025); Jordan et al. (2024)). In this paper we focus on performance only, due to resource constraints.

**7. Initialisation**  Tasks are set to either '*empty*', where only the interface of each editable module is provided (to ensure implementations fit the codebase), or '*baseline*', where editable modules start from baseline algorithms. We compare these settings in Section 8.2.

### 4.1. An Example Task

We use Example 1 to demonstrate the task interface. For this on-policy RL task, the ADA will work in JAX (Bradbury

et al., 2018) to maximise the performance of an RL agent. The majority of the codebase is *fixed*: domain-specific code like wrapper functions, environment creation, and the optimiser, training loop, target and activation modules. While ADAs *can* interact with these files, changes are overwritten prior to meta-testing to prevent evaluation hacking.

The ADA must work with two *editable* files: the loss, which starts as an empty function mapping inputs (e.g., data batches) to a scalar loss; and the network architecture, which maps environment observations to an RL policy, value function and recurrent state, and also contains no logic. The ADA can make arbitrary code edits, such as writing additional functions, importing extra packages, and filling templates. The agent can *run* inner-loop training and evaluation for both meta-train environments. To ensure security during agent execution, the ADA operates in a containerised environment. If one editable module is *only* called by another, the ADA can edit its interface, expanding the search-space of algorithms in DiscoGen tasks further.

```
1  task_domain = "OnPolicyRL"
2  meta_train = ["Ant", "Freeway"]
3  meta_test = ["Pusher", "Craftax"]
4  backend = "recurrent"
5  change_optim = False
6  change_loss = True
7  change_networks = True
8  change_train = False
9  change_target = False
10 change_activation = False
11 eval_type = "performance"
12 initialisation = "empty"
```

*Example 1.* An example task configuration. A task is defined by its task domain, meta-train/meta-test datasets, backend and modules.

### 4.2. Procedural Task Generation

DiscoGen is a procedurally generated benchmark of ML tasks. As a generator, it is designed for *improving* ADAs, manually or automatically in an ADA optimisation loop (Figure 1). A task is specified by a small configuration file, like Example 1, which takes the same role as the parameters in other PCG environments (Cobbe et al., 2020; Stooke et al., 2021). This can be randomly generated, user-specified, or sampled using autocurricula. The technical details of sampling tasks in DiscoGen are described in Appendix D.1.

DiscoGen creates tasks in a two-stage process to reduce the risk of meta-test leakage. The meta-train portion of the task is generated first; only *after* the meta-loop is complete does DiscoGen create the meta-test codebase. As such, the agent is never given any details of the meta-test datasets.

PCG involves creating levels, or tasks, programmatically. DiscoGen is no different; the number of tasks for a domain is combinatorial with respect to how many modules and datasets it supports. Specifically, for a task domain with $m$ modules, $d$ datasets, and $b$ backends, and our currently

supported 3 evaluation types and 2 task initialisations,

$$N_{tasks} = 2 \cdot 3 \cdot b \cdot (2^m - 1) \cdot \left(3^d - 2^{(d+1)} + 1\right). \quad (1)$$

Derived in Appendix D.2, this assumes at least one editable module and assigns each dataset to either meta-train, meta-test or unused (with at least one meta-train and meta-test).

### 4.3. Available Task Domains

Table 2 shows the domains in DiscoGen, which we expect to grow from open-source contributions; adding new domains needs only mild adaptation of existing codebases. Due to imbalanced task counts, we recommend stratified sampling over domains to reduce bias, which is supported by the DiscoGen library. We describe each domain, its modules and datasets in Appendix A.

*Table 2.* Overview of domains and their number of supported tasks.

| Task Domain | $m$ | $d$ | $b$ | $N_{tasks}$ |
|---|---|---|---|---|
| Bayesian Optimisation | 6 | 11 | 1 | 65,413,656 |
| Brain Speech Detection | 3 | 7 | 1 | 81,144 |
| Computer Vision Classification | 4 | 9 | 1 | 1,679,400 |
| Continual Learning | 5 | 3 | 3 | 6,696 |
| Greenhouse Gas Prediction | 2 | 4 | 1 | 900 |
| Language Modelling | 3 | 4 | 2 | 4,200 |
| Model Unlearning[1] | 1 | 3 | 1 | 85,176 |
| Neural Cellular Automata | 5 | 5 | 1 | 33,480 |
| Off-Policy RL | 7 | 4 | 1 | 38,100 |
| Offline RL | 5 | 10 | 1 | 10,602,372 |
| On-Policy MARL | 6 | 17 | 2 | 97,431,783,120 |
| On-Policy RL | 6 | 13 | 3 | 1,789,383,960 |
| Trajectory Prediction | 4 | 3 | 3 | 1,080 |
| Unsupervised Environment Design | 3 | 4 | 1 | 2,100 |
| Total | | | | 99,299,115,384 |
| Median | | | | 59,622 |

DiscoGen exhibits useful diversity across these millions of tasks; in Section 7, we show that ADA optimisation improves as more DiscoGen tasks are experienced. In Section 8.1 we demonstrate how performance varies across different editable module combinations; as the number of editable modules lowers the ADA's success rate. The domains supported in DiscoGen span a range of machine learning fields, incorporate datasets of varying complexity and difficulty, and are built upon codebases of differing scales. Most task domains include unique modules, and when there is overlap, implementations are domain-specific.

We further validate this diversity through *rank correlation analysis* over the performance of different ADAs in DiscoBench (Section 5.2), a subset of DiscoGen tasks, in Figures 3 & 4 (Appendix F). Hierarchical clustering of the correlation matrix reveals distinct patterns; while correlation is often high between similar modules in different domains or different modules in the same domain, there are also anti-correlations where strong performance in one task

---

[1]Model unlearning entails finetuning pretrained models. For $n$ models, $N_{tasks} = 2 \cdot 3 \cdot b \cdot (2^m - 1) \cdot \left((2n+1)^d - 2(n+1)^d + 1\right)$

implies poor performance in another, including within the same domain. The Fisher-Z transformed mean Spearman correlation is $\sim 0.4$; high enough to indicate non-random signal, while sufficiently small to show low redundancy in DiscoGen. Notably, we also find that clustering patterns are *distinct* between meta-train and meta-test, demonstrating how the *same algorithm* can rank differently across datasets.

Per-dataset analysis in each task reinforces the meaningfulness of DiscoGen's large task space. Considering Appendix J, the ranking of the discovered algorithms changes across datasets within the *same task*. This is intuitive, given prior literature suggests optimal algorithms differ between datasets or RL environments (e.g., in reinforcement learning (Goldie et al., 2024; Jackson et al., 2025), computer vision (Rodrigo et al., 2024; Takahashi et al., 2024) or language modelling (Dao & Gu, 2024; Jelassi et al., 2024)).

### 4.4. Advantages of DiscoGen

Individual task design in DiscoGen also overcomes the many flaws of previous task definitions raised in Section 3.1.

**Principled Evaluation & Contamination Resistance** DiscoGen tasks clearly distinguish between meta-train and meta-test. Since DiscoGen is procedural, and there is no knowledge of the meta-test datasets in meta-training, the potential for test leakage is limited. Even as DiscoGen enters pre-training datasets, this is a step towards fairer evaluation, ensuring DiscoGen remains pertinent for a long time. Furthermore, DiscoGen supports different evaluation *types*, enabling discovery for factors other than performance.

**High Diversity** DiscoGen generates highly diverse tasks. As demonstrated in Section 4.2, DiscoGen supports a range of domains with different data structures, filesystem complexities and modules. Since DiscoGen tasks are parameterised combinatorially, its tasks represent a smooth range of difficulties between "*implement a single module for one easy dataset*" to "*implement many modules for many hard datasets*", rather than simply *easy* or *hard* subsets.

**Different Initialisations** Agents need not implement full codebases, and the initialisation of editable modules can be set to just the inputs and outputs of the module (empty) or full baseline implementations. In essence, tasks are either '*improve a baseline*' or '*de novo discovery*'. This enables better analysis of the biases elicited by ADAs, can make tasks easier or harder, and may increase agent creativity.

**Ease of Adding Tasks** Beyond our currently implemented domains, adding many tasks to DiscoGen is significantly easier than for other suites. For similar effort to adding one task to, say, MLE-Bench (Chan et al., 2025) or MLGym-Bench (Nathani et al., 2025), DiscoGen can gain potentially millions of new tasks in its support. When the base code for a task domain is complete, adding more tasks is even easier; isolating a new module effectively doubles the number of possible tasks, and adding a new dataset near-triples it.

**Unsaturated Problems** Since tasks can be made more or less difficult by changing the module and dataset configurations (e.g., Section 8.1), DiscoGen tasks span a wide range of difficulties. We find that the more modules there are to implement, the harder the problem but the higher the potential ceiling. Additionally, almost all datasets currently in DiscoGen have yet to be solved by humans, let alone agents, and adding more, harder datasets is straightforward.

## 5. ADA Evaluation: DiscoBench

Despite the contamination risk that arises from releasing a fixed public task suite, there is still merit to evaluating ADA performance over a set of DiscoGen tasks that resolve the other flaws from prior benchmarks (Section 3.1).

DiscoBench is a manually curated ADA evaluation set akin to hand-designed levels built in PCG environments (e.g., (Matthews et al., 2025; Nikulin et al., 2024)). For each domain in DiscoGen, we create $m + 1$ tasks; $m$ tasks where each of the $m$ modules is active (*Single Edit*), and 1 where all modules are active simultaneously (*All Edit*). DiscoBench is the union of these sets; we separate them here for analysis only. We do not include other module combinations to ensure DiscoBench stays manageable, and to enable principled expansion with new domains. Meta-train and meta-test sets are fixed between tasks to enable comparison, and are selected pseudo-randomly; long-to-run datasets are reserved for meta-testing, for computational reasons. These splits are included in Appendix E. ADAs should not be optimised on DiscoBench to ensure it is an appropriate test suite (though the probability of sampling tasks from DiscoBench is non-zero, as in other PCG environments). Since DiscoBench is not yet public, our evaluation is not subject to contamination; however, this work entering the public domain exposes it to pretraining or internet search agents. As such, we plan to release a private DiscoBench 'API' using datasets not mentioned publicly.

### 5.1. Experimental Setup

We explore the performance of different LLMs with the MLGym ADA (Nathani et al., 2025), a ReAct agent (Yao et al., 2023) which can run code, read files and choose when to submit, within a fixed action budget. Due to resource constraints, and for reproducibility purposes, we evaluate three open-source language models: Deepseek-v3.2 (DeepSeek-AI et al., 2025), Devstral2 (Mistral AI, 2025) and GPT-OSS 120B (OpenAI et al., 2025). We include an '*all fixed*' baseline (i.e., the code with no editable modules) for comparison; it is *always* possible for the ADA to implement this. We provide experimental details and hyperparameters in Appendix C, include our generic ADA system prompt in Appendix K.1, and detail cost and compute usage in Appendix C.5.

In both *DiscoBench Single* and *DiscoBench All*, we aggregate scores over three seeds per task and model. We report two per-model success-rates and Elo ratings (Elo, 1978)

*Table 3.* ADA evaluation performance in DiscoBench (Elo Scores with 95% CIs). Bold indicates best mean performance.

| Model | DiscoBench (Single Edit) | | | DiscoBench (All Edit) | | | DiscoBench | | |
|---|---|---|---|---|---|---|---|---|---|
| | Succ. | Meta-Train | Meta-Test | Succ. | Meta-Train | Meta-Test | Succ. | Meta-Train | Meta-Test |
| Baseline (All Fixed) | — | **1111** [1088, 1133] | **1133** [1114, 1153] | — | **1443** [1340, 1621] | **1388** [1264, 1575] | — | **1129** [1109, 1150] | **1144** [1120, 1166] |
| GPT-OSS 120B | 59.5% | 907 [885, 931] | 944 [924, 966] | 17.8% | 596 [366, 712] | 595 [142, 758] | 52.0% | 892 [871, 914] | 931 [911, 955] |
| Devstral2 | 55.2% | 963 [932, 986] | 924 [900, 945] | 39.3% | 790 [634, 890] | 1010 [909, 1127] | 52.4% | 948 [928, 969] | 928 [909, 951] |
| Deepseek-v3.2 | 71.3% | 1019 [997, 1042] | 999 [978, 1020] | 40.5% | 1171 [1090, 1320] | 1007 [926, 1174] | 65.7% | 1031 [1013, 1049] | 998 [974, 1016] |

based on same-dataset comparisons; one for meta-train, and one for meta-test. We report 95% confidence intervals for Elo, estimated using 100 bootstrap samples as in Appendix C. Since agents frequently fail to consistently produce valid solutions for many tasks[2], we penalise failure such that a model with more successful runs dominates one with fewer; this penalty does not apply in baseline comparisons. We also report the total success rate for each model.

To understand how models could perform without failure, we run extra experiments on *DiscoBench (Single Edit)* until each model has three successful attempts; we include results in Appendix I.2. Due to low success rates, this was unaffordable for *DiscoBench (All Edit)* and a small number of tasks in *DiscoBench (Single Edit)*. This failure is expected; in Nathani et al. (2025), many frontier closed source models failed over their four attempts in MLGym-Bench tasks. These tasks are omitted from the *Until Success* analysis.

### 5.2. Results

We report results in Table 3, and include a per-task score breakdown in Appendix J. To demonstrate that agents can discover interesting and performant algorithms, we discuss two hand-selected algorithms in Appendix H.

Firstly, success rates in *All Edit* are significantly lower than for *Single Edit*, confirming the hypothesis that including more editable modules increases task difficulty. This is explored further in Section 8.1, where we sweep over all module combinations in On-Policy RL and find that the success rates of ADAs *consistently* fall as more editable modules are added. In fact, average success rates for all three models are low. In contrast, we examine *Success@3* rates (i.e., what proportion of tasks had *at least one* successful solution from 3 attempts) in Appendix I.1, and find that they are 10-30 percentage points higher than the aggregated rates. Considering this performance gap, and that *all* baselines follow the same interface as the editable modules, it is clear that agents struggle to **robustly** produce even simple, well-known algorithms. We find that failures are broadly driven by syntax errors or, often, code overfitting to the meta-train datasets (e.g., hardcoded shapes).

Elo shows a similar pattern; the baseline has a much higher score in *All Edit* than in *Single Edit*. Even when agents have three successful solutions, no agent consistently out-

performs the baseline to the point of having a higher Elo (Appendix I.2). Since ADAs do not yet match well-known human algorithms, even though they *could* be implemented and are often *suboptimal* algorithms for the datasets, there is clearly a significant margin for ADAs to improve.

Comparing ADAs, Deepseek-v3.2 performs well in *DiscoBench (Single Edit)*, both in meta-train and meta-test, as well as on aggregate in the full *DiscoBench* set. However, due to currently low success rates, drawing conclusions for *DiscoBench (All Edit)* is difficult. Overall, the relative baseline performance increases between meta-train and meta-test, suggesting there are signs of meta-overfitting for ADAs; there is clearly space for ADA improvement.

It is important to ensure DiscoBench is diverse; a single model being uniformly dominant would suggest DiscoBench only measures general ability. We explore this using rank-correlation analysis in Appendix F and find high variation in rankings between tasks. In fact, the per-task results (Appendix J) show that even the ranking over datasets within the *same* task varies, demonstrating how different algorithms are better for different datasets and justifying claims of diversity in DiscoGen. Furthermore, this per-task breakdown confirms the range of task difficulties; sometimes agents outperform the baseline, usually they produce weak-but-valid solutions, and often they fail completely.

## 6. Enabling New Research

In addition to enabling discovery of better algorithms, by extracting discovered artifacts, DiscoGen is a platform for a plethora of new research directions. To serve as inspiration to the wider research community, we propose some ideas here. In Section 7, we show how DiscoGen can be used for prompt optimisation, demonstrating one such use-case.

### 6.1. Understanding The Pathologies of ADAs

Our ability to analyse the pathologies of algorithm discovery systems is hindered by the design and limited controllability of existing benchmarks. DiscoGen provides an expansive space in which to run such analysis. For example, research could seek to understand creativity differences (Haase et al., 2025; Franceschelli & Musolesi, 2025) as task initialisation changes, the limits of instruction-following in complex filesystems (Zeng et al., 2024; Ouyang et al., 2022), or whether ADAs are biased towards certain domains or modules (e.g., they might be better at designing optimisers than losses). This enables more scientific development of ADAs.

---

[2]Since at least one model produced a valid solution for every task, we have an existence proof that all tasks are solvable.

### 6.2. Learning to Discover Algorithms

LLM reasoning has significantly improved since the introduction of RL (Shao et al., 2024; Ouyang et al., 2022; Gehring et al., 2025; Kazemnejad et al., 2025) or evolution (Sarkar et al., 2025; Qiu et al., 2025) post-training. Prior work generally focuses on mathematics or programming, where there are many verifiable problems to learn from. As DiscoGen enables sampling of *billions* of unique algorithm discovery tasks, findings from these other task-rich domains can be transferred towards training better ADAs. Such *meta-meta-learning* could optimise for efficient, quick, or performant algorithms, or all three.

### 6.3. Sampling Hard-Yet-Learnable Discovery Tasks

Prior work has shown how autocurricula (Dennis et al., 2020; Parker-Holder et al., 2022a; Foster et al., 2025) can sample hard-yet-learnable tasks to improve minimum expected performance bounds (Beukman et al., 2024) and improve training efficiency (Foster et al., 2025). Since tasks in DiscoGen are of varying difficulty, it is naturally suited to curriculum methods, and their application could improve the performance and efficiency over random task sampling. This is especially important in ADA optimisation, where task completion times can vary by orders of magnitude.

### 6.4. Algorithm World Models

Copet et al. (2025) train a "Code World Model" to replicate a code interpreter's state as it ran, to improve an LLM's programming abilities. However their work necessitates vast amounts of data. Combining DiscoGen with similar computational resources could enable collection of a similarly large-scale, structured dataset of algorithm-performance pairs. This could be used to train an '*algorithm world model*' (AWM) that predicts an algorithm's performance. Such a model could be used directly in an ADA, fine-tuned as above, or integrated into tree-search agents as below.

### 6.5. Training LLM-As-A-Judge in Tree-Search ADAs

"*How to explore?*" is an open question in algorithm discovery and AI research (Toledo et al., 2025). Some agents designed for automated research and algorithm discovery use tree-search (Jiang et al., 2025; Toledo et al., 2025), but evaluating a tree's leaves requires running expensive inner-loop trainings. Instead, using an LLM or otherwise trained model to evaluate algorithms could act as a filter, selecting promising leaves to run (Yu et al., 2025; Herr et al., 2025) or acting as a value function (Wang et al., 2025) or reward model (Zhang et al., 2025) to skip inner-loop training and reduce the cost of search.

However, off-the-shelf judge performance would likely be poor, since we ideally want to evaluate super-human (and thus, out-of-distribution) algorithms. Using data generated using DiscoGen, it would be possible to train either a full model or a value prediction head that could be integrated into Monte-Carlo Tree Search (Kocsis & Szepesvári, 2006).

### 6.6. Symbolic Evolution of Algorithms

Since DiscoGen provides templates for module (i.e., defining inputs and outputs), and the rest of an algorithm's code is pre-implemented, it is well-placed for working with non LLM-based algorithm discovery methods; for example, symbolic search or black-box meta-learning. This could include developing methods using genetic programming, as in (Ramachandran et al., 2017; Zheng et al., 2022; Chen et al., 2023; Goldie et al., 2025), or black-box evolution (e.g., (Lu et al., 2022; Metz et al., 2022b; Goldie et al., 2024)).

### 6.7. Self-Improving ADAs

An alternative to *designing* ADAs is letting them build their *own* scaffolds (e.g., (Hu et al., 2025; Zhang et al., 2026a; Wang et al., 2026; Zhang et al., 2026b)), for data-driven open-ended self-improvement. However, optimising an agent's scaffold on existing algorithm discovery suites could lead to overfitting. With DiscoGen, such systems could be run near-indefinitely with low risk of overfitting to individual tasks, enabling the discovery of super-human ADAs.

## 7. ADA Optimisation Using DiscoGen

Given the impact of prompting on LLM performance (Lester et al., 2021; Fernando et al., 2024), we optimise an ADA's prompt for DiscoGen tasks. We query an 'ADA-Optimisation' LLM, distinct from the ADA, to iteratively update the prompt in the *ADA optimisation loop*, based on past meta-train and meta-test performances in sampled DiscoGen tasks. Its objective is to optimise **meta-test performance**. Since the 'prompting' LLM is queried infrequently, it uses a more expensive closed-source model; Claude Sonnet 4.5 (Anthropic, 2025b). As the best performing LLM tested in DiscoBench, the ADA uses DeepSeek-V3.2.

We explore how task quantity, $K_{tasks}$, correlates to ADA optimisation performance over 30 updates. We sample from DiscoGen at different frequencies; when $K_{tasks} = 1$, we tune the prompt on the same task 30 times, and when $K_{tasks} = 30$, we use a different task each iteration. To prevent bias, we uniformly sample task domains *before* task configurations. To test if ADA optimisation in DiscoGen improves ADA generalisation, we hold out a small number of domains during ADA optimisation (Appendix C.4); in practice, all domains should be used to maximise the breadth of experience. Results are presented in Table 4, using DiscoBench for ADA evaluation and the Elo methodology from Section 5.2, with additional analysis in Appendix G. We include the prompt-tuner prompt in Appendix K.

Our results suggest a positive relationship between increasing $K_{tasks}$ and meta-test performance (i.e., the ADA optimisation objective): as task count increases, ADA performance improves. While this pattern is strongest for the in-distribution domains (but still unseen tasks), the monotonic trend holds in the held-out set, which are out-of-support for the optimised ADAs. Such results demonstrate the value of

*Table 4.* ADA evaluation performance after prompt optimisation (Elo Scores with 95% CIs). Bold indicates the best (point-estimate) Elo.

| $K_{tasks}$ | In-Distribution Domains | | | Held-Out Domains | | |
|---|---|---|---|---|---|---|
| | **Succ.** | **Meta-Train** | **Meta-Test** | **Succ.** | **Meta-Train** | **Meta-Test** |
| 1 | 69.9% | 955 [931, 980] | 965 [940, 990] | 65.0% | 979 [947, 1012] | 983 [954, 1011] |
| 5 | 74.2% | **1063** [1038, 1087] | 978 [955, 1002] | 68.5% | 985 [952, 1018] | 994 [966, 1022] |
| 10 | 71.6% | 935 [911, 961] | 991 [967, 1014] | 72.4% | **1031** [998, 1065] | 1000 [973, 1028] |
| 30 | 75.2% | 1047 [1023, 1071] | **1067** [1042, 1093] | 72.3% | 1005 [971, 1037] | **1024** [995, 1052] |

procedural generation for ADA optimisation; given $K_{tasks}$ can still be dramatically scaled, with additional resources, ADA performance can likely be improved much further. Interestingly, this pattern does *not* exist for meta-train (which we do not optimise for), meaning conventional evaluations would have misrepresented the ADA performance.

Considering the prompts themselves (Appendix K.3), there is a noticeable progression from $K_{tasks} = 1$ to $K_{tasks} = 30$. Whereas lower task counts over-index on specific tasks, prompts developed over more tasks emphasise broader discovery and machine learning principles.

## 8. Analysing DiscoGen

Here, we analyse the design of DiscoGen for the three ADAs from Section 5.2. Task redundancy and the meta-train/meta-test performance gap are analysed in Appendix F.

### 8.1. Changing Modules

Using PCG in DiscoGen provides a smooth distribution of tasks to optimise over, potentially enabling autocurricula methods. To verify this smoothness, we evaluate ADAs for a sweep over the *editable* modules in On-Policy RL domain. We measure the success rate for each LLM over every possible combination of 6 modules (i.e., $2^6 - 1 = 63$ tasks), for a fixed meta-train/meta-test split, in Figure 2.

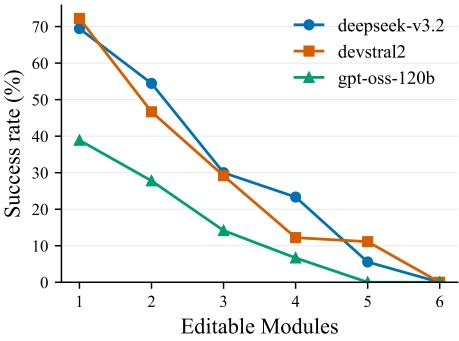

*Figure 2.* Success rate vs. editable module count in On-Policy RL.

For every model, as the number of editable modules *increases*, the success rate *decreases*. As such, tasks in DiscoGen can smoothly be made more or less difficult simply by increasing or decreasing its number of editable modules.

### 8.2. Using a Different Initialisation

DiscoGen enables different task initialisations; either *empty* files, which only provide a "function interface" to the agent,

or starting from *baseline* implementations. While we experiment with *empty* initialisations through most of this paper, we compare the two settings here to verify that *empty* initialisations are more difficult than *baseline* ones. Due to computational limitations, we restrict these experiments to one LLM (Deepseek-v3.2) and a set of quick-to-run Disco-Gen domains, as set out in Appendix C.4.

We find that success rates for starting from *baseline* (92.4%) are significantly higher than for *empty* initialisations (75.4%), and Elo scores are higher (1101 vs 899, respectively). Given all else is kept equal, baseline initialisation clearly makes tasks easier. Furthermore, a non-exhaustive, qualitative analysis of the discovered algorithms from the *baseline* experiments shows that they often do remain close to the original implementation.

## 9. Conclusion

In this paper, we introduced DiscoGen, a procedural generator of algorithm discovery tasks. We motivated the design of DiscoGen by the shortcomings of existing algorithm discovery suites, such as poor evaluation methodology and limited scale. We demonstrated that DiscoGen overcame many of these flaws, and established DiscoBench, an ADA evaluation set. We subsequently introduced a number of possible research avenues enabled by DiscoGen. As demonstration of the potential of DiscoGen, we used it for ADA prompt optimisation and showed that prompt performance improved with the number of tasks it was optimised for; both within and beyond the support of the optimisation distribution. Finally, we analysed the design of DiscoGen, showing how different axes of task variation affect difficulty.

**Future Work**    Beyond the research *enabled* by DiscoGen (Section 6), we highlight a number of avenues for future work. Firstly, DiscoGen is non-exhaustive; expanding its support of domains, modules and datasets, via open-source contributions, would increase its utility. Additionally, ADA evaluation is still affected by contamination (though less than other benchmarks). Implementing an API-only DiscoBench suite, with datasets not available in DiscoGen, could solve this issue. Testing better models and scaffolds in DiscoBench would provide a greater understanding of the current ADA ceiling. Finally, our ADA optimisation experiments only focus on prompt tuning, a relatively limited form of training, due to cost constraints; exploring scaffold- or weight-based improvement is an obvious next step.

## Acknowledgements

The authors would like to credit **Hannah Erlebach** with the Neural Cellular Automata task domain implementation.

**AG** is funded by the EPSRC Centre for Doctoral Training in Autonomous Intelligent Machines and Systems EP/S024050/1. **ZW** is funded by a generous grant from Waymo. **AK** is supported by Exscientia and the SABS CDT. **CW** is funded by the EPSRC DTP Research Studentship. **TW** is funded by the EPSRC Centre for Doctoral Training in Autonomous Intelligent Machines and Systems EP/Y035070/1. **UB** is supported by the EPSRC Centre for Doctoral Training in Autonomous Intelligent Machines and Systems and the Rhodes Scholarship. **NR** is supported by the Defense Advanced Research Projects Agency (DARPA). Our experiments were made possible by an equipment grant from NVIDIA. **SR** is funded by the Transport and Mobility Institute at Delft University of Technology. **JF** is partially funded by the UKRI grant EP/Y028481/1, which was originally selected for funding by the ERC. **JF** is also supported by the JPMC Research Award and the Amazon Research Award. This project received compute resources from a generous grant provided by the Isambard-AI National AI Research Resource, for the projects "Robustness via Self-Play RL" and "FLAIR Summer Moonshots".

The authors thank **Jonathan Cook** and **Edward Hughes** for their comments and advice in this project.

## Contributions

**Alexander D. Goldie** led the project, created and designed DiscoGen, ran all experiments, wrote the paper and implemented the On-Policy RL task domain.

**Zilin Wang** contributed to the design of DiscoGen and implemented the Computer Vision Classification and Brain Speech Detection task domains.

**Adrian Hayler** contributed to the design of DiscoGen and implemented the Language Modelling task domain.

**Deepak Nathani** contributed to the design of DiscoGen and the structure of the DiscoGen repository.

**Edan Toledo** contributed to the design of DiscoGen and helped edit the paper.

**Ken Thampiratwong** contributed to the design of DiscoGen.

**Aleksandra Kalisz** contributed to the design of DiscoGen.

**Michael Beukman** implemented the Unsupervised Environment Design task domain and helped edit the paper.

**Alistair Letcher** implemented the Model Unlearning task domain.

**Shashank Reddy Chirra** implemented the Off-Policy RL and Multi-Agent RL task domains, and contributed additional modules for the On-Policy RL task domain.

**Clarisse Wibault** implemented the Bayesian Optimisation task domain.

**Theo Wolf** implemented the Greenhouse Gas Prediction task domain.

**Charles O'Neill** implemented the Continual Learning task domain.

**Uljad Berdica** implemented the Offline RL task domain.

**Nicholas Roberts** implemented the state-space model backend for the Language Modelling task domain.

**Saeed Rahmani** implemented the Trajectory Prediction task domain.

**Roberta Raileanu**, **Shimon Whiteson** and **Jakob N. Foerster** provided equal supervision over the course of the project.

## Impact Statement

Considering the social, economic and safety effects of automated algorithm discovery is crucial due to the potential magnitude of its impacts. In this work, we are considering not only how a specific algorithm can be meta-learned (which has more limited impact), as in much prior work, but how to *optimise* agents towards the development of new machine learning algorithms. While we believe there are a range of significant benefits that such work can lead to, we are also wary and considerate of the risks.

It is often preferable to develop models with specialist capabilities; consistently focusing on generalist improvements can introduce safety concerns (Bommasani et al., 2022; Weidinger et al., 2021), and may not even be the most effective way to enhance desired capabilities (Belcak et al., 2025). We believe algorithm discovery provides a surgical tool with which to automate research, offering a much more specific objective than '*automating science*', as in a lot of automated AI scientist literature (Lu et al., 2026; Intology, 2025; Yamada et al., 2025). Rather than giving black-box agents the ability to decide what problems they work on freely, algorithm discovery (and specifically, DiscoGen) is designed to involve a human-in-the-loop; discovery agents are only allowed to make changes to certain files, with *all* other changes being overwritten at test time. Furthermore, despite the massive task space available in DiscoGen, all tasks are well-defined, scoped and based on human-selected codebases, meaning they are constrained. Given the analysed suite of 14 domains, we do not believe any tasks deemed unsafe *could* be sampled from DiscoGen. However, since we hope for community contributions in the future, it is important to ensure that this remains the case moving forward.

As we emphasise optimisation for algorithm discovery *only*, DiscoGen provides a platform for AI-Human Co-Improvement (Weston & Foerster, 2025). Crucially, rather than training towards more generally-able agents (where algorithm discovery capabilities are improved simultaneously with adverse capabilities), optimising on DiscoGen improves performance *directly* for human-designed algorithm discovery tasks. We believe this type of scoped optimisation is a necessary path towards safe super-human agents; it is safer to develop specifically super-human agents than developing poorly understood general superintelligence.

It is important to recognise that development of automated algorithm discovery systems can pose ethical and economic issues. While we focus on general and beneficial task domains, harmful actors could instead look to optimise agents in unethical domains. Fully mitigating this misuse is beyond the scope of our paper, but it is important to ensure DiscoGen is comprised only of broadly beneficial domains to limit any negative behaviours. Similarly, discovering better machine learning algorithms automatically enables them to be used for harmful datasets; while this is a risk of all machine learning research, we must acknowledge that DiscoGen accelerates their development. At the least, to reduce the risk of directly meta-learning on these datasets, it is important to ensure there is broad human-alignment in foundation models, making it hard to optimise for misaligned behaviour (Bai et al., 2022). A counter-measure to these risks is to ensure DiscoGen *supports* AI safety tasks. For example, discovering algorithms for the *Model Unlearning* task in DiscoGen (Yao et al., 2024) is one avenue for reducing unwanted behaviours in foundation models.

Ensuring people from affected fields are kept in-the-loop as these systems develop is necessary for ensuring automated algorithm systems *complement*, rather than *replace*, humans. The potential impact of AI on the labour market is significant (Eloundou et al., 2024), and thus developing systems which protect the role of humans is important. We believe that automated algorithm discovery provides a strong balance of developing research breakthroughs using AI, while maintaining the agency and empowerment of humans who design the problem settings for it to operate in. In general, this fulfils the idea that research agents are most effective when provided as tools to humans (Gottweis et al., 2025; Shneiderman, 2022), rather than acting to substitute them.

Moreover, the requirements for operating in this field are high; running or querying large language models is expensive, and optimising research agents is more so. This introduces large financial (Chen et al., 2024) and environmental (Strubell et al., 2019; Faiz et al., 2024) costs, which should be considered when experimenting with DiscoGen. One high-impact area for future research would involve developing energy-efficient automated algorithm discovery systems; in fact, these could be developed automatically, using a well-

defined objective in combination with the research proposal of Section 6.7. Furthermore, using *energy* as the evaluation criterion to discover new, hyper-efficient algorithms could offset the costs of running ADAs, given the potential downstream savings enabled by such algorithms.

Democratisation of AI is one necessary tool for ensuring it can act in the benefit of all, rather than a few major players. We believe DiscoGen helps with this in two ways. Firstly, in making a large-scale research environment for algorithm discovery available, we believe that such research is made more feasible outside of industrial frontier labs. Secondly, by emphasising the development of *specialist*, rather than *generalist*, agents, we hope to enable a landscape of research using smaller models which require less pre-training data. However, it would be naïve to suggest that research in automated science and algorithm discovery is not expensive, or doesn't require significant hardware resources. Moving forward, it is necessary to ensure the development of these tools is accessible to a wide range of researchers, from academia through to small and large parts of industry.

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

# Appendix

Our appendix is structured as follows:

- Appendix A provides an overview of each task domain included in DiscoGen. We provide a high-level overview of the goal of each domain, its implementation details, and what editable modules and datasets it supports.

- Appendix B extends the abridged related work from the main text (Section 2) to include discussion of the wider field.

- Appendix C provides hyperparameters and experimental details for our paper. It also includes a description of our Elo calculation and discussion of the experimental compute and cost for the paper.

- Appendix D introduces implementation details of DiscoGen. We discuss how to derive the expression for its task count, and how DiscoGen creates new tasks when sampled from.

- Appendix E provides a breakdown of what the meta-train and meta-test datasets are for each domain in DiscoBench.

- Appendix F examines the redundancy of tasks in DiscoBench, using average-rank correlation, to demonstrate the semantic difference between different tasks.

- Appendix G expands the prompt optimisation results from Section 7 and provides further experimental details and analysis of the system.

- Appendix H introduces two example discovered algorithms, demonstrating that the most performant meta-train/meta-test runs are making novel discoveries rather than just rehashing baselines.

- Appendix I reports and discusses success@3 metrics (i.e., the rate of *at least* one of the three seeds producing a successful solution), and *until success* scores (i.e., the Elo scores when ADAs are run until they have three successful independent seeded runs).

- Appendix J provides per-task results for all experiments in the paper.

- Appendix K includes all prompts used in this paper. This includes prompts developed by the prompt optimisation loop.

# A. Task Domain Overview

In this section, we provide a brief overview of the implementations of each task domain included in DiscoGen, as well as all references covering their original implementations and the origin of their datasets.

## A.1. Bayesian Optimisation

**Task Domain**    Bayesian Optimisation (Jones et al., 1998; Garnett, 2023) addresses the problem of optimising an expensive, black-box objective function under a limited evaluation budget. A probabilistic surrogate model is fit to observed function evaluations, and an acquisition function, balancing exploration of uncertain regions with exploitation of promising candidates, is used to select the next evaluation point. The objective is to identify the global minimum or maximum of the function within a fixed number of queries.

**Implementation**    Our Bayesian Optimisation implementation is based on Boax (Löper, 2023).

**Editable modules**    We include six editable modules for Bayesian Optimisation: the surrogate model; the optimiser used to fit the surrogate model; the acquisition function; the optimiser used to maximise the acquisition function; the initial domain sampling strategy; and the query selection policy.

**Datasets**    We include 11 synthetic optimisation functions that are standard in Bayesian Optimisation literature (Surjanovic & Bingham): Ackley 1D, Ackley 2D, Branin 2D, Bukin 2D, Cosine 8D, Drop-Wave 2D, Egg-Holder 2D, Griewank 5D, Hartmann 6D, Holder-Table 2D and Levy 6D.

## A.2. Brain Speech Detection

**Task Domain**    Brain Speech Detection is a neural signal processing task in which a model predicts the presence of speech from non-invasive or invasive brain recordings. The goal is to learn a two-class classifier conditioned on neural activity (e.g., MEG signals).

**Implementation**    The code for Brain Speech Detection is adapted from the official LibriBrain competition (Landau et al., 2025) codebase, which provides a standardised pipeline for neural signal preprocessing, model training, and evaluation.

**Editable modules**    We include three editable modules in Brain Speech Detection: the loss function; the optimiser; and the network architecture.

**Datasets**    We split the original LibriBrain dataset (Özdogan et al., 2025) into seven parts, constituting seven datasets, each of which contains MEG data and labels collected during the process of the same participant listening to one chapter of Sherlock Holmes in spoken English.

## A.3. Computer Vision Classification

**Task Domain**    Computer Vision Classification is a supervised learning task in which a model assigns a discrete semantic label to an input image. The objective is to learn robust visual representations that generalise across variations in appearance, scale, illumination, and data distribution. This task is a foundational benchmark in computer vision and is widely used to evaluate model architectures, optimisation strategies, and robustness to dataset shift. In DiscoGen, the Computer Vision Classification task spans standard, corrupted, long-tailed, and fine-grained classification settings.

**Implementation**    The code for Computer Vision Classification is adapted from the MLGym (Nathani et al., 2025) benchmarking infrastructure. The implementation provides a unified training and evaluation pipeline for image classification models, including dataset loading via HuggingFace Datasets, model initialisation, optimisation, and metric computation.

**Editable modules**    We include four editable modules in Computer Vision Classification: the loss function; the optimiser; the network architecture; and the image preprocessing.

**Datasets**   We support nine widely used image classification datasets, covering a range of difficulty levels and distributional properties. These include MNIST (LeCun et al., 2010) and Fashion-MNIST (Xiao et al., 2017) for grayscale digit and apparel classification; CIFAR-10 and CIFAR-100 (Krizhevsky et al., 2009) for small-scale natural image classification; Tiny ImageNet (Russakovsky et al., 2015; Wu et al., 2017) for large-class-count evaluation; CIFAR-10-C (Hendrycks & Dietterich, 2019) for corruption robustness; CIFAR-10-LT (Krizhevsky et al., 2009) for long-tailed class imbalance; Oxford Flowers-102 (Nilsback & Zisserman, 2008) and Stanford Cars (Krause et al., 2013) for fine-grained object classification. All datasets are accessed through HuggingFace Datasets and follow standardised train, validation, and test splits.

## A.4. Continual Learning

**Task Domain**   Continual learning is a broadly defined field in which a model must learn from non-stationary data sources without forgetting its previous learnings (Wang et al., 2024b). Non-stationarity can arise through a number of means; in our tasks, which are focused on image classification under nonstationarity, these include randomly permuting the labels attached to images or randomly subsampling classes shown throughout training.

**Implementation**   Our default implementation is based on elastic weight consolidation (Kirkpatrick et al., 2017).

**Editable modules**   We support five different modules in continual learning: the optimisation algorithm; the regulariser for mitigating catastrophic forgetting; the replay buffer for storing past experience; the sampler for mixing replay data with new data; and the learning rate scheduler.

**Additional Backends**   Our default backend uses a ResNet-18 for its base network (He et al., 2016). However, we also offer backends supporting vision transformers (Dosovitskiy et al., 2021; Wightman, 2019) and parameter isolation models (Rusu et al., 2022), a common architecture for preventing catastrophic forgetting (Kirkpatrick et al., 2017) in continual learning.

**Datasets**   We support three datasets for continual learning: PermutedMNIST (LeCun et al., 2010), SplitCIFAR100 (Krizhevsky et al., 2009) and TinyImageNetSplit (Wu et al., 2017).

## A.5. Greenhouse Gas Prediction

**Task Domain**   Forecasting the concentration of greenhouse gases in the atmosphere is an important tool for predicting and mitigating the effects of climate change.

**Implementation**   Our base implementation is based on a standard scikit-learn (Pedregosa et al., 2011) training loop.

**Editable modules**   We support two modules in Greenhouse Gas Prediction: the model architecture and the way the data is processed before modelling.

**Datasets**   We support four datasets from Lan & Keeling that are used for predicting concentrations of $CO_2$, $CH_4$, $N_2O$ and $SF_6$ in the atmosphere. Each dataset is split into a training dataset (pre-2014) and a validation dataset (2015-2025).

## A.6. Language Modelling

**Task Domain**   Language Modelling is the task of learning the underlying distribution of text data via next-token prediction over vast bodies of text, mostly scraped from the internet. We evaluate the quality of a trained language model by computing the exponential of the average negative log-likelihood across next-token predictions on a validation set, also called perplexity.

**Implementation**   Our default implementation builds on a modified codebase based on the `modded-nanogpt` repository (Jordan et al., 2024).

**Editable modules**   We support three editable modules: the network architecture, the loss function, and the optimiser.

**Additional Backends**   In addition to a base nanogpt implementation, we support a state-space model backend based on Mamba (Gu & Dao, 2024; Dao & Gu, 2024).

**Datasets**  We support the following four datasets: LMFineWeb 10B (Penedo et al., 2024), TinyStories (Eldan & Li, 2023), OPC-FineWeb Math and OPC-FineWeb Code (Huang et al., 2025).

## A.7. Model Unlearning

**Task Domain**  Model Unlearning, also called Machine Unlearning (Cao & Yang, 2015), is the task of modifying (e.g. fine-tuning) a model to "forget" targeted information such as sensitive personal data, copyrighted content, or harmful knowledge, all while preserving the model's overall capabilities on unrelated tasks. We specifically focus on LLM unlearning (Yao et al., 2024) across 3 datasets and a variety of open-weight models (see below).

**Implementation**  The code for all tasks is adapted from OpenUnlearning (Dorna et al., 2025). Preservation of general capability is evaluated using LMEvalHarness (Gao et al., 2024).

**Editable modules**  We include a single editable module in Model Unlearning: the loss function, which should typically balance two objectives: unlearning specific knowledge from the forget set while preserving performance on the retain set.

**Datasets**  We support 3 different datasets for Model Unlearning: TOFU (Maini et al., 2024), MUSE (Shi et al., 2024), and WMDP-Cyber (Li et al., 2024).

**Models**  We support 13 different open-weight LLMs from 5 providers: Llama-2-7b-chat-hf, Llama-2-7b-hf, Llama-2-13b-hf, Llama-3.2-1B-Instruct, Llama-3.2-3B-Instruct, Llama-3.1-8B-Instruct (Touvron et al., 2023; Llama Team, 2024), Gemma-7b-it (Mesnard & et al., 2024), Phi-1.5, Phi-3.5-mini-instruct (Abdin et al., 2024), Qwen2.5-1.5B-Instruct, Qwen2.5-3B-Instruct, Qwen2.5-7B-Instruct (Qwen, 2024) and Zephyr-7b-beta (Tunstall et al., 2024).

## A.8. Neural Cellular Automata

**Task Domain**  Neural Cellular Automata (NCAs) are a form of cellular automata in which updates to cells (a function of their neighbours' states) are parameterised and learned by a neural network (Mordvintsev et al., 2020). This task is about designing the components that the NCA uses to perceive its neighbours and learn its update rule.

**Implementation**  The code for Neural Cellular Automata is adapted from CAX (Faldor & Cully, 2025), a library which provides a unified JAX framework for implementing cellular automata.

**Editable modules**  We include five editable modules: (i) the perceive module, (ii) the update module, and (iii) the loss, (iv) optimiser and (v) train modules for training the update module.

**Datasets**  We support five datasets: two growing tasks (Mordvintsev et al., 2020), a self-classifying MNIST task (Randazzo et al., 2020), matrix operations (Béna et al., 2025) and MNIST inpainting (Tesfaldet et al., 2022).

## A.9. Off-Policy RL

**Task Domain**  Off-policy RL refers to a class of reinforcement learning approaches in which an agent learns from experience generated by a behaviour policy that may differ from the policy being optimised, including experience collected in the past or by other agents. In this task, we focus on value-based methods that learn value functions by minimising the temporal-difference (TD) error (Sutton & Barto, 2020).

**Implementation**  The code for Off-Policy RL is adapted from the Deep Q-learning (Mnih et al., 2013, DQN) implementation from PureJaxRL (Lu et al., 2022), which is itself based on CleanRL (Huang et al., 2022a;b).

**Editable modules**  We consider six editable modules: (i) the loss function, which determines the prediction targets for the value network; (ii) the optimiser; (iii) the network architecture; (iv) the replay mechanism, which specifies how experience is stored and sampled during training; (v) the policy, which governs the trade-off between exploration and exploitation; and (vi) the training loop.

**Datasets**   We support Off-Policy RL on four environments from the MinAtar suite (Young & Tian, 2019)—Asterix, Breakout, Freeway, and Space Invaders—as reimplemented in Gymnax (Lange, 2022) using JAX. These environments are simplified versions of their corresponding Atari games (Bellemare et al., 2013).

### A.10. Offline RL

**Task Domain**   Offline Reinforcement Learning (offline RL) is a class of algorithms that trains a policy from previously collected environment transitions, without any other additional environment interactions (Levine et al., 2020). Unlike in online RL, the agent cannot explore and collect new trajectories from the *online* environment and must extract the policy from pre-existing datasets. Such datasets may contain suboptimal trajectories or insufficient state coverage (Kumar et al., 2022). In practice, the real environment—the *online* domain inaccessible during training—can be used to tune the algorithm's hyperparameters (Jackson et al., 2025).

**Implementation**   We base our implementation on Revisited Behaviour Regularized Actor-Critic (ReBRAC) (Tarasov et al., 2023), a high-performing and hyperparameter-robust offline RL algorithm. Our JAX implementation is based on those by Jackson et al. (2025) and Park et al. (2025b).

**Editable Modules**   We include five editable modules for the offline RL task domain: (i) the actor loss, (ii) the critic loss, (iii) the network architectures, (iv) the optimizer, and (v) the training loop.

**Datasets**   We support all single-task, reward-labeled datasets from OGBench (Park et al., 2025a), a currently unsaturated offline RL benchmark. This encompasses several robot morphologies across various locomotion and manipulation tasks.

### A.11. On-Policy Multi-Agent RL

**Task Domain**   On-policy multi-agent reinforcement learning (MARL) refers to a class of multi-agent methods in which each agent updates its policy using an on-policy RL algorithm, thereby relying solely on trajectories generated by their current policy.

**Implementation**   The implementation of our on-policy MARL task is adapted from the Independent PPO (IPPO) algorithm (Yu et al., 2022), as provided in the JaxMARL library (Rutherford et al., 2024b).

**Editable modules**   We include six editable modules in On-Policy MARL: the advantage estimation and critic target computation, the loss function, the optimiser, the network architecture, the activation function used by the network, and the training loop.

**Additional Backends**   In addition to the default, we support a recurrent architecture. In this case, the training loop must handle the recurrent state produced by the agent's network.

**Datasets**   We support 17 environments from the JaxMARL library (Rutherford et al., 2024b), comprising 5 multi-agent Brax tasks, 11 tasks from SMACv2 (Ellis et al., 2023), and the MPE Spread environment (Lowe et al., 2017).

### A.12. On-Policy RL

**Task Domain**   On-Policy RL is a subset of reinforcement learning in which an agent learns from experience collected by its own policy (Sutton & Barto, 2020), rather than from data collected by a different behaviour policy.

**Implementation**   The code for On-Policy RL is adapted from the Proximal Policy Optimisation (Schulman et al., 2017, PPO) implementation from PureJaxRL (Lu et al., 2022), which is itself based on CleanRL (Huang et al., 2022a;b).

**Editable modules**   We include four editable modules in On-Policy RL: the loss function; the optimiser; the network architecture; and the training loop.

**Additional Backends**   In addition to the default, we support two backends that can be used to augment tasks in On-Policy RL. These are: a recurrent agent, in which the train loop must support a recurrent variable produced by the agent's networks;

and a transformer agent (Vaswani et al., 2017), in which the agent architecture uses attention.

**Datasets** We support 13 different environments for On-Policy RL, from three different environment suites. These include: Ant, HalfCheetah, Hopper, Humanoid, Pusher, Reacher and Walker2D from Brax (Freeman et al., 2021), a JAX-based (Bradbury et al., 2018) implementation of MuJoCo (Todorov et al., 2012); Asterix, Breakout, Freeway and SpaceInvaders from Gymnax (Lange, 2022), a reimplementation of four MinAtar environments (Young & Tian, 2019) which are simplifications of some Atari games (Bellemare et al., 2013); and Craftax and Craftax-Classic (Matthews et al., 2024), which are based on Crafter (Hafner, 2022).

### A.13. Trajectory Prediction

**Task Domain** Trajectory prediction is the task of forecasting the future positions of traffic participants, such as vehicles, pedestrians, and cyclists, given their observed past motion and surrounding road context (Huang et al., 2022c). Effective trajectory prediction must account for the multi-modal nature of human behaviour; at any moment, an agent may accelerate, brake, turn, or change lanes, which can lead to multiple plausible futures. The objective is to produce a set of $K$ diverse future trajectories for a target agent, each with an associated probability, that collectively cover the space of likely outcomes. Models are evaluated on how closely their best predictions match the ground truth using minimum Average Displacement Error (minADE), minimum Final Displacement Error (minFDE), miss rate, and Brier-minFDE. The ability to anticipate these possibilities is critical for safe motion planning in self-driving systems.

**Implementation** The base code for Trajectory Prediction is adapted from UniTraj (Feng et al., 2024), a unified framework for scalable vehicle trajectory prediction, and AutoBot (Girgis et al., 2022), a latent-variable sequential set transformer for joint multi-agent motion prediction. The implementation provides a standardised pipeline for encoding past agent trajectories and road geometry via attention-based modules, decoding multi-modal future trajectories with uncertainty estimates, and evaluating predictions against ground truth.

**Editable modules** We include four editable modules in Trajectory Prediction: (i) the loss function, which defines the multi-modal training objective balancing trajectory likelihood with mode diversity; (ii) the optimiser, which controls weight updates and learning rate scheduling; (iii) the network architecture, which encodes agent dynamics, social interactions, and map context before decoding future trajectories; and (iv) the training loop, which orchestrates data loading, model training, validation, and model selection.

**Datasets** We support three large-scale autonomous driving datasets: Argoverse 2 (Wilson et al., 2023), collected across six US cities (Austin, Detroit, Miami, Pittsburgh, Palo Alto, and Washington D.C.); nuScenes (Caesar et al., 2020), recorded in Boston and Singapore and covers diverse urban conditions; and the Waymo Open Motion Dataset (Ettinger et al., 2021), spanning six US cities (San Francisco, Phoenix, Mountain View, Los Angeles, Detroit, and Seattle). Each dataset is preprocessed into a standardised format with 21 observed timesteps (2.1 s at 10 Hz), up to 32 surrounding agents, 128 map polylines, and a 60-timestep (6 s) prediction horizon. All three datasets contain 850 preprocessed scenarios, hosted on HuggingFace.

### A.14. Unsupervised Environment Design

**Task Domain** Unsupervised Environment Design (UED) is a field focused on training agents that are robust, and able to generalise to a wide distribution of tasks (Dennis et al., 2020; Jiang et al., 2021b; Parker-Holder et al., 2022a; Samvelyan et al., 2023). In particular, this tends to be posed as a two-player game between a task-proposing adversary and a task-solving student (Dennis et al., 2020). There are several objective functions that the adversary can use, ranging from theoretically principled ones such as regret (Dennis et al., 2020) to more empirically motivated ones like positive value loss (Jiang et al., 2021a) and learnability (Rutherford et al., 2024a).

**Implementation** We use the base scaffold from Sampling for Learnability (Rutherford et al., 2024a), which uses PPO as the underlying learning algorithm, and training comprises two stages. The first is sampling a large batch of random environments, then rolling out the agent on these, and using these trajectories to score each level. The second phase then trains the agent on a mixture of the high-scoring levels, and uniform random environments.

**Editable modules** The first editable module is responsible for rolling the agent out on candidate environments, and then providing a score for each of them. The second is the train step, which controls the actual RL agent update, as well as how the filtered levels are used during training. Finally, the last module consists of the hyperparameters for the sampling and training process.

**Datasets** We consider two distinct domains; the first is Minigrid (Chevalier-Boisvert et al., 2023), which is a partially-observable 2D navigation task. The second is Kinetix (Matthews et al., 2025), an open-ended domain of 2D physics tasks. The task distribution consists of a broad distribution of randomly-generated physics puzzles, and the goal is to generalise to interesting, human-designed problems. There are three levels of difficulty in the Kinetix benchmark, corresponding to how many entities there are in a scene. We use Minigrid and the Small Kinetix setting as training tasks, and Medium and Large as the testing tasks.

# B. Additional Related Work

## B.1. Automated Research

AutoML involves trying to apply machine learning to new data without needing human experts (Hutter et al., 2019; He et al., 2021; Parker-Holder et al., 2022b). Historically, AutoML has focused on areas like hyperparameter tuning (e.g., (Lindauer et al., 2022; Li et al., 2017; Parker-Holder et al., 2020; Eimer et al., 2023)), selecting algorithms from a possible set (Feurer et al., 2015; Lindauer et al., 2017; Feurer et al., 2022), neural architecture search (Stanley & Miikkulainen, 2002; Stanley et al., 2009; Zoph & Le, 2017; Zoph et al., 2018; White et al., 2023) and data cleaning and augmentation (Zhang et al., 2017; Neutatz et al., 2022; Krishnan et al., 2016). While many of these techniques involve applying *existing* machine learning algorithms, the objective of automated algorithm discovery is to develop *new* machine learning algorithms without relying on manual design.

Meta-learning involves learning algorithms from data in a 'meta-loop' (Schmidhuber, 1987; Real et al., 2020; Beck et al., 2025). Often, meta-learned algorithms are black-box, taking the form of a neural network. For example, prior work has considered meta-learning optimisation algorithms (e.g., (Andrychowicz et al., 2016; Metz et al., 2019; Oh et al., 2020; Metz et al., 2022a;b; Goldie et al., 2024; Lan et al., 2024; Oh et al., 2025)) or loss functions (e.g., (Kirsch et al., 2020; Bechtle et al., 2021; Lu et al., 2022; Alfano et al., 2023)). Optimising these black-box algorithms frequently relies on evolution (Metz et al., 2022b; Lu et al., 2022; Jackson et al., 2024; Goldie et al., 2024; 2025) or meta-gradients (Oh et al., 2020; 2025). However, more interpretable algorithms can also be discovered using symbolic evolution or search (Ramachandran et al., 2017; Cranmer et al., 2020; Zheng et al., 2022; Chen et al., 2023), or, driven by the rise of highly capable language models (e.g., (OpenAI, 2025c; Pichai et al., 2025; Meta AI, 2025; Anthropic, 2025b)), repeatedly prompting language models for new algorithm suggestions (Lu et al., 2024; Romera-Paredes et al., 2024; Novikov et al., 2025; Nathani et al., 2025; Toledo et al., 2025; Gideoni et al., 2025). In this paper, we frame each algorithm discovery problem as its own *task*, and show how the DiscoGen procedural task generator can help optimise these algorithm discovery systems.

Given a rise in language model capabilities (METR, 2025; Chollet et al., 2025; Phan et al., 2025; Paglieri et al., 2025), creating better coding (Yang et al., 2024; Jiang et al., 2025; Anthropic, 2025a; OpenAI, 2025b) and research agents (Lu et al., 2026; Nathani et al., 2025; Toledo et al., 2025) has developed into an important sub-field. Such systems have been used for increasingly complex mathematical (Hubert et al., 2025; Luong et al., 2025), software engineering (Yang et al., 2024; Liu et al., 2026) and research (Taylor et al., 2022; Lu et al., 2026; Yamada et al., 2025; Karpathy, 2026) tasks. LLM agents augment base language models with the ability to take actions or use tools (Wang et al., 2024a; Schick et al., 2023), enabling them to run code (Nathani et al., 2025; Yang et al., 2024; Anthropic, 2025a), search the internet (Nakano et al., 2021; Thoppilan et al., 2022; Komeili et al., 2021; Gao et al., 2023; OpenAI, 2025a; Google), or access applications like calculators for verifiable tasks (Cobbe et al., 2021). These tools are used by AI research agents to automate all or part of the research process in a ReAct loop (Yao et al., 2023), in which tools are used intermittently throughout a task. In this work, we propose a new procedural task generator for algorithm discovery tasks to help drive forward development of new agents and models for algorithm discovery.

The development and analysis of such agents includes developing agents purely for ideation (Si et al., 2025), implementing research ideas proposed by a human-in-the-loop (Gottweis et al., 2025; Weston & Foerster, 2025), judging the quality of research papers (Lu et al., 2026; Si et al., 2025; Thakkar et al., 2025), or even automating the entire workflow to write scientific articles (Lu et al., 2026; Yamada et al., 2025; Intology, 2025; Si et al., 2026). Methods for integrating LLMs into algorithm discovery systems have taken a range of forms. For example, LLMs are often used as mutation and crossover operators in evolutionary algorithms (Ma et al., 2024; Klissarov et al., 2025; Romera-Paredes et al., 2024). However, such systems vary widely in how much agency they give to the LLM. Design decisions include how the agent should search over new algorithms using, for example, tree-search algorithms (Jiang et al., 2025; Toledo et al., 2025), whether the LLM should decide which experiments to run (Yu et al., 2025), or which algorithm to submit as the 'final' version (Nathani et al., 2025).

## B.2. Optimising Agents

To optimise a model, architecture, or system for generalising over a problem space, we need to be able to evaluate it over a large amount of data (Kaplan et al., 2020; Hoffmann et al., 2022; Chung et al., 2024). For example, language models improve as they are pretrained on more data (Kaplan et al., 2020), and generalisation in deep reinforcement learning (RL) has benefitted from procedurally generated and larger environments (Cobbe et al. (2020), Section 2.3). However, it is often preferable to develop models with specialist capabilities; consistently focusing on generalist improvements can introduce safety concerns (Bommasani et al., 2022; Weidinger et al., 2021), and may not be the most effective way to enhance desired

capabilities (Belcak et al., 2025).

Instead, prior work has often focused on building agentic architectures or training models for specific domains. For example, significant effort has gone into developing better agents for mathematics (e.g., (Lewkowycz et al., 2022; Trinh et al., 2024; Hubert et al., 2025)) or programming (e.g., (Li et al., 2022; Jimenez et al., 2024; Rozière et al., 2024)). Underpinning many of these advances is access to an evaluation signal from a large, diverse and verifiable problem set which can be optimised by both models, via training (Shao et al., 2024; Wen et al., 2026), and researchers, through agent and prompt design. For example, there are large suites of mathematical problems (e.g., (Cobbe et al., 2021; Hendrycks et al., 2021b)), factual questions (Joshi et al., 2017; Hendrycks et al., 2021a), and programming suites can leverage open-source code repositories to verify that inputs lead to expected outputs (Jimenez et al., 2024; Chen et al., 2021). Developing these large suites for algorithm discovery has proven more difficult; manual curation of codebases requires expert knowledge, and often needs adaptation to be applied to different data. Additionally, copying outputs from existing code is insufficient for algorithm discovery, wherein the objective is develop *superhuman* algorithms that are better than those already in existence (Romera-Paredes et al., 2024; Novikov et al., 2025). In this work, we propose a procedural algorithm discovery task generator that spans millions of possible algorithm discovery tasks, and does not require domain-specific expert knowledge for generating new tasks.

### B.3. Procedural Generation

Procedural generation involves automatically creating levels or environments, according to rules, rather than manually designing every instance individually (Togelius et al., 2013). To do so, environments in procedural generation are defined as Contextual Markov Decision Processes (Hallak et al., 2015, Contextual MDPs) or Underspecified Partially Observable MDPs (Dennis et al., 2020), where each level or task is defined by a small number of configuration variables. In deep RL, procedural environment generation has proven effective for training agents on a *distribution* of levels, such that they generalise over a distribution rather than just fitting the specific environment levels seen (Cobbe et al., 2020; Frans & Isola, 2023). Such approaches have been applied to generate levels in environments of ranging complexity, from gridworlds (Chevalier-Boisvert et al., 2023) to physics engines (Frans & Isola, 2023; Matthews et al., 2025), 3-dimensional worlds (Stooke et al., 2021), or games (Shaker et al., 2016; Küttler et al., 2020; Cobbe et al., 2020; Hafner, 2022; Wang et al., 2021; Matthews et al., 2024). Procedural generation opens up new avenues of research too, by enabling the development of autocurricula (e.g., (Jiang et al., 2021b; Dennis et al., 2020; Parker-Holder et al., 2022a)) or large scale meta-learning (e.g., (Nikulin et al., 2024)).

# C. Experimental Details

## C.1. Hyperparameters

All DiscoBench experiments are run over three meta-seeds; that is, three independent algorithm discovery runs. In all experiments, agents are given a **24 hour time limit**, and either an 80 (for *DiscoBench Single*) or 100 (for *DiscoBench All*, since tasks are more complicated) action budget, in which to complete a task. Within said limit, the agent is allowed to run many experiments for different implementations, each of which returning performance metrics for all meta-train datasets. Furthermore, for *DiscoBench Single*, we run additional experiments until each agent has 3 valid attempts. For this setting, we remove a small number of tasks (On-Policy RL/Off-Policy RL train_loop and Model Unlearning) since they proved too difficult to produce three successful attempts.

In all experiments, we use the MLGym agent with different open-source LLMs. We do not add or change the implementations of tools from MLGym. We use default hyperparameters from MLGym for all experiments (Nathani et al., 2025), besides those specified below:

Table 5. Non-default MLGym hyperparameters.

| Setting | gpus_per_agent | max_steps |
|---|---|---|
| *DiscoBench (Single Edit)* | 1 | 80 |
| *DiscoBench (All Edit)* | 1 | 100 |
| *Prompt Tuning (Meta-Training)* | 4 | 50 |
| *Prompt Tuning Single* | 1 | 50 |
| *Prompt Tuning All* | 1 | 60 |

## C.2. Elo Calculation

Our results are comparative between models, and based on an Elo score. In Elo, each model's score changes based on pairwise win-loss comparisons of their performance in individual tasks, and the Elo disparity between the model. Before calculating the Elo, we aggregate the per-dataset scores for each model over its meta-seeds; we also include a penalty to scores which ensures that a model with more successful attempts out of its three meta-seeds dominates a model with less successful attempts.

For model 1 with rating $R_1$ and model 2 with rating $R_2$, the expected outcome score of model 1 is

$$E_1 = \frac{1}{1 + 10^{(R_2 - R_1)/400}},$$

where a 400 point difference corresponds to a $\approx 91\%$ chance of winning.

After calculating $E_1$, the rating of model 1 is updated to reflect the difference between the expected and true score as

$$R_1' = R_1 + K \cdot (S_1 - E_1).$$

In this expression, $S_1$ is the score for model 1, and is set as 1.0 for a win, 0.5 for a draw and 0.0 for a loss. Scores for all models are initialised at 1000, and $K = 32$ to balance volatility of score calculations. We loop through the data for 1000 epochs, shuffling each epoch and annealing $K$ to 1 for stability. We use 100 bootstrapping rounds to estimate 95% confidence intervals.

## C.3. Prompt Optimisation Experimental Details

For our prompt optimisation experiment, we use Claude-4.5 Sonnet (Anthropic, 2025b) to automatically refine prompts for an ADA; here the ADA is the MLGym agent with Deepseek-V3.2 (DeepSeek-AI et al., 2025; Nathani et al., 2025). We sweep over different numbers of tasks for prompt optimisation by changing the *frequency* at which a task is sampled. As such, a new task is sampled every $30/N_{tasks}$ steps.

We explored the generalisation of each prompt to a set of four held-out domains (which is a larger distribution shift than for the 10 in-distribution domains, despite all task specifics being held out from the agent). The four held-out domains we select are: *On-Policy MARL, Offline RL, Neural Cellular Automata, and Trajectory Prediction.*

The task sampling process involved first sampling a task domain from the ten possible in-distribution set from DiscoGen. Each dataset from the task domain had a $40\%$ chance of inclusion in meta-train or meta-test, and a $20\%$ chance of being excluded from the task. Every module has an independent $30\%$ chance of being marked as editable. Any invalid task (i.e., no editable modules or an empty meta-train/meta-test set) is discarded, and we resample from DiscoGen. We cap the maximum amount that each domain can be sampled to 10, to prevent domain bias, but this limit is not reached in practice over 30 ADA optimisation iterations.

To prevent unbounded context growth for the ADA Optimisation LLM (Claude-4.5 Sonnet), we do not provide full conversation history. Instead, at every ADA optimisation step, we provide: a system prompt (Appendix K); the current and three preceding prompts with their performance; and up to 15 previous task domain-performance pairs without their corresponding prompt, to help ground scores. It is not told the specific task configuration, to aid development of a general prompt. Claude is given per-dataset breakdowns, as well as whether each dataset is from meta-train or meta-test in a given task. We also tell Claude if there were any failures or error messages, but do not provide full error messages; again to limit the context size. If Claude does not produce a prompt fitting the prescribed template within three attempts, the task is discarded and resampled without increasing the ADA optimisation iteration (i.e., the task is not counted towards the total).

## C.4. Different Initialisation Domains

Due to computational constraints, we are unable to run an exhaustive comparison between initialisation strategies for *all* tasks in the DiscoGen ADA evaluation set. Instead, we select a number of the quicker-to-run domains to run our ablation on. In particular, we use tasks for the following domains: *Bayesian Optimisation; Brain Speech Detection; Greenhouse Gas Prediction; Computer Vision Classification; Neural Cellular Automata; On-Policy RL; Off-Policy RL; and Trajectory Prediction*.

## C.5. Experimental Compute

All experiments are run using Nvidia H200 GPUs. We run the majority of experiments on a single GPU, besides prompt optimisation experiments which are run on 4 GPUs to accelerate experimentation, and On-Policy MARL experiments which use 4 GPUs due to environment memory requirements. For all experimentation (including most development and experiments which are not directly included in this paper), we use an estimated 35,000 GPU-hours of compute. However, such high-end compute is not generally a requirement for usage. To ensure DiscoGen is accessible to researchers under different constraints, we verified that many DiscoGen task domains can run on consumer-grade (e.g., Nvidia 2080Ti/3080) and mid-range (Nvidia L40s) hardware.

All models were run through an API. The total cost of API credits used in development and experiments is estimated to be about $1600. A task generally costs on the order of $0.10-$0.50 to run with open-source models.

# D. DiscoGen Details

## D.1. Technical DiscoGen Implementation

Here, we clarify the implementation details of DiscoGen; in particular, how tasks are generated.

DiscoGen operates by creating *file systems*. To sample a random task from DiscoGen, a random DiscoGen configuration must be created. Doing so involves selecting a task domains, which defines the availability of modules, datasets and backends, and randomising each of these categories. An example task configuration is shown in Example 1.

After a configuration has been created, DiscoGen is queried to build the meta-train portion of the task. To do so, DiscoGen creates: (1) all necessary files for *each* meta-training dataset (in per-dataset folders), including downloading and caching any data; (2) a script for running an inner-loop on all meta-training datasets; (3) a directory which includes all discovered algorithms; and (4) a procedurally generated `description` file. The task description covers *four* things: a high-level overview of algorithm discovery; a description of the task domain; information about each of the editable modules, including what purpose it serves and what interface it must have; and descriptions of each dataset in meta-training. We provide all of these descriptions in our open-source repository, which also includes annotated task-generation code (`make_files.py`).

Any ADA operates in a meta-loop over the meta-train files to develop new algorithms. When the meta-loop is complete, DiscoGen is queried again to build the meta-*test* task. As this is created, **all** files besides the editable modules in `discovered/` are overwritten, and rebuilt from scratch for different datasets. This is done to lower the risk of evaluation hacking. Since the meta-test datasets are not known to the agent, we also dramatically reduce the risk of train-test leakage.

## D.2. Deriving Task Counts

For $m$ different modules, $d$ different datasets, and $b$ different backends in a task domain, we derive the number of valid tasks below.

Each dataset can be marked as part of the *meta-train* set, *meta-test* set, or *excluded* set, given 3 possible options per dataset. The same dataset can not be included more than once, to prevent leakage between the meta-train and meta-test sets (i.e., $\mathcal{D}_{train} \cap \mathcal{D}_{test} = \varnothing$. Therefore, the number of possible combinations of datasets is $3^d$. However, we require *at least* one dataset to be in the meta-train, and *at least* one dataset in meta-test. We remove the two $2^d$ cases where this is not true (i.e., where either meta-train or meta-test are excluded as options), but add back the double-counted case of `all exclude`. This produces $(3^d - 2 \times 2^d + 1) = (3^d - 2^{(d+1)} + 1)$ valid dataset configurations.

For $m$ modules, each modules can be marked as *editable* or *fixed*, meaning each module has 2 possible states. This means there are $2^m$ possible module combinations. By removing the case where all modules are *fixed*, this gives $2^m - 1$ valid modules configurations.

We currently support 2 types of initialisation (start-from-interface and start-from-baseline), and 3 types of evaluation (performance, energy and time).

By combining the number of configurations for datasets and modules, and multiplying by the number of backends, initialisations and evaluation types, this gives

$$N_{tasks} = 2 \cdot 3 \cdot b \cdot (2^m - 1) \cdot \left( 3^d - 2^{(d+1)} + 1 \right).$$

For Model Unlearning, where we can provide one of $n$ different base models for each dataset, the computation changes slightly. As opposed to counting 3 options for every dataset there are $2n + 1$ possible choices; any combination of meta-train/model ($n$), meta-test/model ($n$), or exclude (1). This provides $(2n + 1)^d$ combinations. The degenerate cases now count as removing $(n + 1)^d$ combinations each, meaning we remove $2(n + 1)^d$ in place of the previous $2 \times 2^d$. This gives

$$N_{tasks} = 2 \cdot 3 \cdot b \cdot (2^m - 1) \cdot \left( (2n + 1)^d - 2(n + 1)^d + 1 \right).$$

# E. DiscoBench Task List

Here, we provide a list of the meta-train/meta-test dataset splits for each DiscoBench task in this paper. All datasets are described and referenced in Appendix A.

## E.1. Bayesian Optimisation

**Meta-Train Datasets:**    Ackley1D, Branin2D, Cosine8D, Eggholder2D, Hartmann6D, Levy6D,

**Meta-Test Datasets:**    Ackley2D, Bukin2D, DropWave2D, Griewank5D, HolderTable2D

## E.2. Brain Speech Detection

**Meta-Train Datasets:**    LibriBrain Sherlock Holmes 1-3

**Meta-Test Datasets:**    LibriBrain Sherlock Holmes 4-7

## E.3. Computer Vision Classification

**Meta-Train Datasets:**    CIFAR10, FashionMNIST, MNIST, OxfordFlowers

**Meta-Test Datasets:**    CIFAR100, CIFAR10C, CIFAR10LT, StanfordCars, TinyImageNet

## E.4. Continual Learning

**Meta-Train Datasets:**    SplitCIFAR100

**Meta-Test Datasets:**    PermutedMNIST, TinyImageNetSplit

## E.5. Greenhouse Gas Prediction

**Meta-Train Datasets:**    CH4, SF6

**Meta-Test Datasets:**    CO2, N2O

## E.6. Language Modelling

**Meta-Train Datasets:**    LMFineWeb, OPCFineWebMath

**Meta-Test Datasets:**    OPCFineWebCode, TinyStories

## E.7. Model Unlearning

**Meta-Train Datasets:**    MUSE

**Meta-Test Datasets:**    TOFU, WMDP-Cyber

**Base Model:**    Qwen2.5-1.5B-Instruct

## E.8. Neural Cellular Automata

**Meta-Train Datasets:**    GrowingLizard, SelfClassifyingMNIST

**Meta-Test Datasets:**    GrowingButterfly, MatrixOperations, MNISTInpainting

### E.9. Off-Policy RL

**Meta-Train Datasets:**   MinAtar/Breakout, MinAtar/Freeway

**Meta-Test Datasets:**   MinAtar/Asterix, MinAtar/SpaceInvaders

### E.10. Offline RL

**Meta-Train Datasets:**   OGBench/antmaze-large-navigate, OGBench/cube-single-play, OGBench/humanoidmaze-medium-navigate, OGBench/puzzle-3x3-play, OGBench/scene-play

**Meta-Test Datasets:**   OGBench/antmaze-giant-navigate, OGBench/antsoccer-arena-navigate,OGBench/cube-double-play, OGBench/humanoidmaze-large-navigate, OGBench/puzzle-4x4-play

### E.11. On-Policy MARL

**Meta-Train Datasets:**   MABrax/Ant, MABrax/Walker, SMAX/2s3z, SMAX/6h_vs_8z, SMAX/27m_vs_30m, SMAX/s-macv2_10_units

**Meta-Test Datasets:**   MABrax/HalfCheetah, MABrax/Hopper, MABrax/Humanoid, MPE/Spread, SMAX/3s_vs_5z, SMAX/3s5z, SMAX/3s5z_vs_3s6z, SMAX/5m_vs_6m, SMAX/10m_vs_11m, SMAX/smacv2_5_units, SMAX/s-macv2_20_units

### E.12. On-Policy RL

**Meta-Train Datasets:**   MinAtar/Breakout, MinAtar/Freeway

**Meta-Test Datasets:**   MinAtar/Asterix, MinAtar/SpaceInvaders

### E.13. Trajectory Prediction

**Meta-Train Datasets:**   nuScenes, Argoverse2

**Meta-Test Datasets:**   Waymo

### E.14. Unsupervised Environment Design

**Meta-Train Datasets:**   Kinetix/Small, Minigrid

**Meta-Test Datasets:**   Kinetix/Medium, Kinetix/Large

# F. Redundancy Analysis

To ensure DiscoGen tasks exhibit semantic diversity, rather than redundancy, in this section we produce and analyse three Spearman Rank Correlation plots based on *DiscoBench Single (Until Success)* results from Section 5.2.

For each task, we compute the *average* rank of the models and baseline over datasets in the task. In Figure 3, we combine meta-train and meta-test results into one set and compute average rank over all datasets. We compute the Spearman Rank correlation between all tasks, which is shown through a heatmap. In Figures 4a and 4b, we show the rank correlations over only the meta-train and meta-test sets for each task respectively.

We use hierarchical clustering in Figures 3 and 4a to group together tasks with similar average rankings. For Figure 4b, we *keep* the task-ordering from the meta-train clustering; this helps visualise the consistency in rank ordering between meta-train and meta-test for the same tasks.

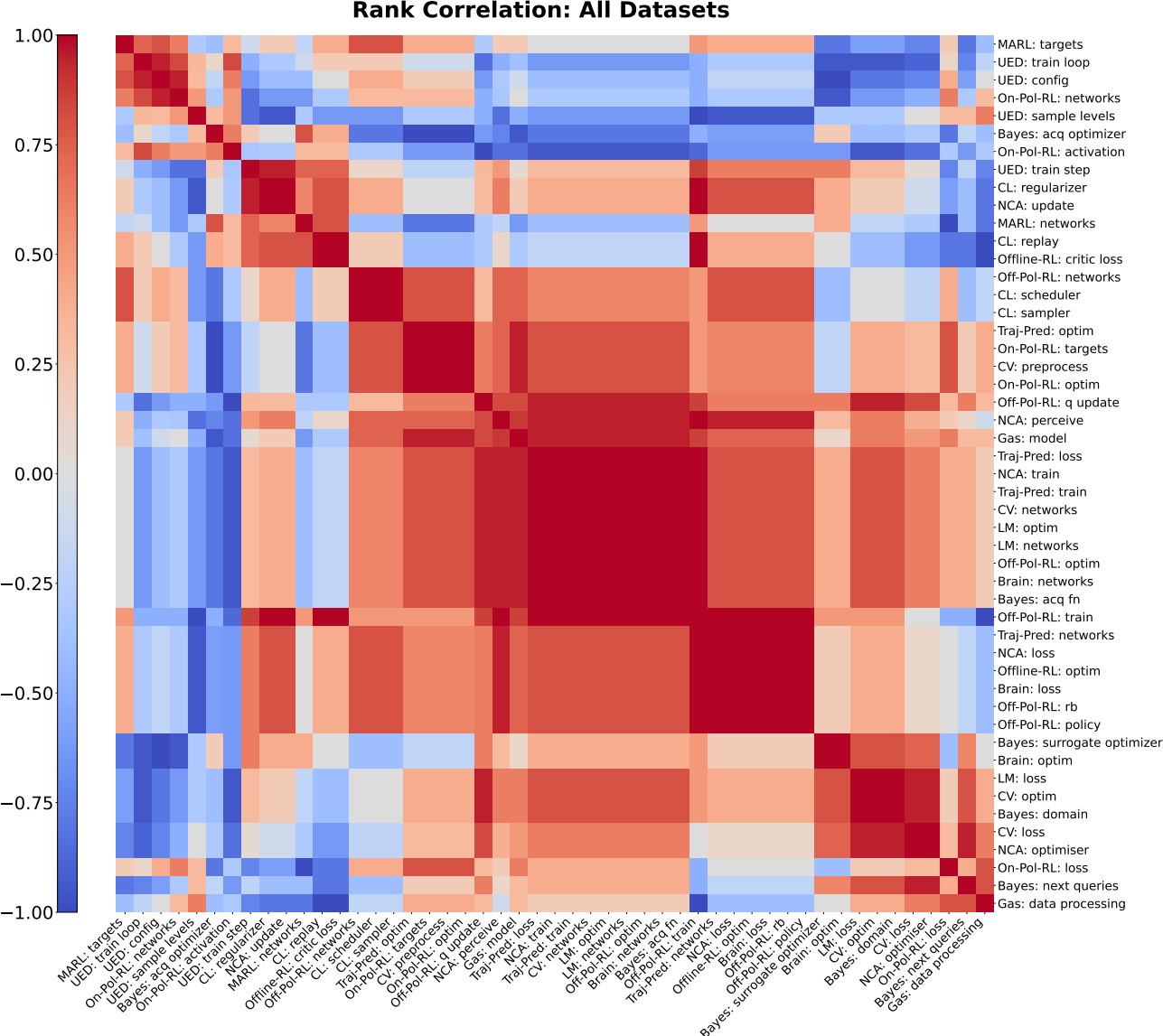

*Figure 3.* Clustered average rank correlation over all datasets (meta-train and meta-test) for each task.

In Figure 3, we find clusters of high-correlation tasks that are occasionally, though not always, intuitive. There is often high correlation between tasks in which the network architecture is the editable module. Alternatively, there is notably *less* correlation when the editable module is the optimiser. We also find frequently mixed correlation between tasks from the

same domain; for example, tasks within on-policy RL or Bayesian Optimisation are frequently anti-correlated.

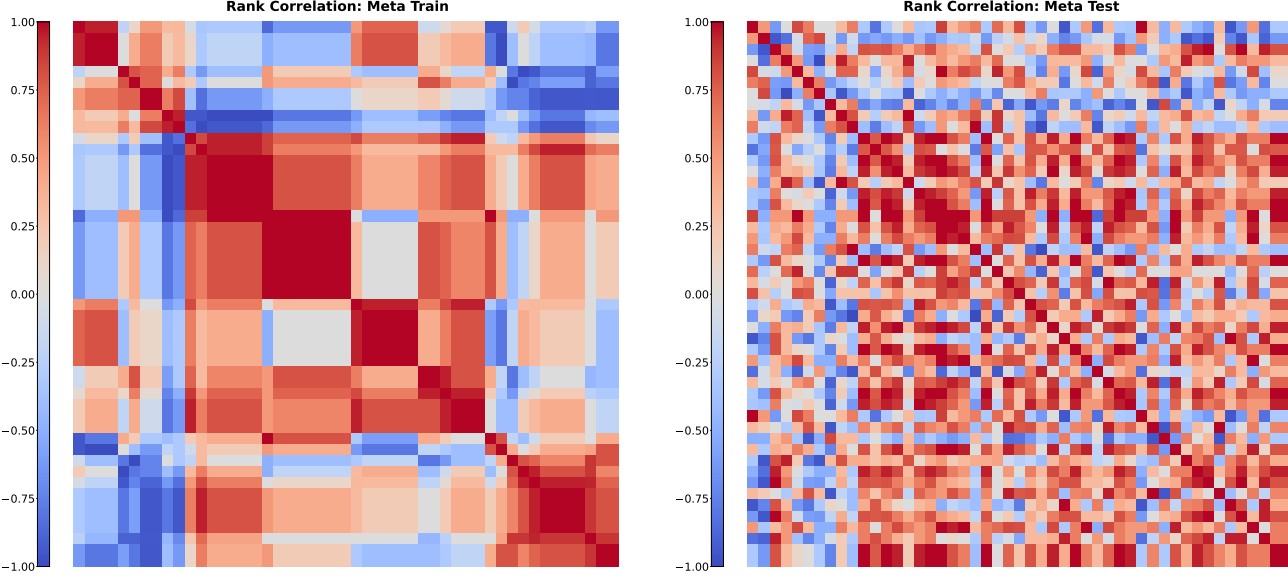

*(a)* Clustered average rank correlation over meta-train datasets.

*(b)* Average rank correlation over meta-test datasets, ordering the tasks based on meta-train clustering.

*Figure 4.* Rank correlations for Meta-Train (left) and Meta-Test (right). The Meta-Test plot inherits the clustering order from Meta-Train to visualize consistency across splits.

In Figure 4, we consider the consistency of clusters between meta-train and meta-test; essentially, exploring whether high and low correlation patterns are reflected between the two regimes, to explore alignment when *only* the datasets are changed. To do so, we first compute the order of labels by clustering the *meta-train* heatmap, and keep this order fixed in the *meta-test* plot. If patterns persist, it would suggest that the rank correlations are similar between the two regimes.

Compared to the clustered plot for meta-train (Figure 4a), the correlation structure in meta-test (Figure 4b) effectively dissolves to noise. Despite the *only* difference between these plots being the datasets (i.e., Figure 4a plotting over meta-train datasets that the agent develops algorithms for, and Figure 4b plotting over the meta-test datasets that the agent doesn't see during its meta-loop), the average algorithm rank changes **dramatically** to the extent that old patterns become broadly unrecognisable. This provides strong justification to the semantic diversity, and lack of redundancy, in the combinatorial task space of DiscoGen; simply changing datasets completely changes the ranking of ADAs. It also justifies the fact that the meta-train evaluation of existing benchmarks is insufficient.

# G. Prompt Tuning Analysis

We do not visualise ADA optimisation curves for $K_{tasks} \neq 1$, since tasks are frequently changing and performance is incomparable between different datasets, domains, and module combinations. However, to demonstrate the validity of our prompt improvement ADA optimisation loop, we visualise the ADA optimisation curves over the meta-train and meta-test environments (both of which are known during ADA optimisation, since meta-test is only held out from the ADA itself).

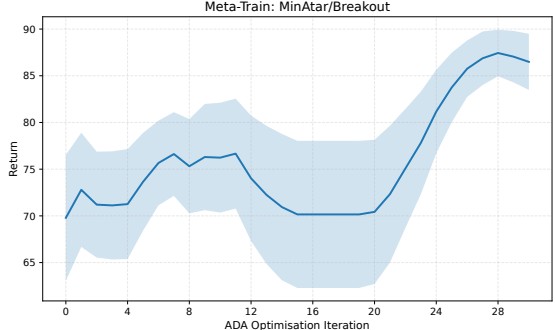

*(a)* Performance in Breakout (a meta-train environment) over the course of ADA optimisation.

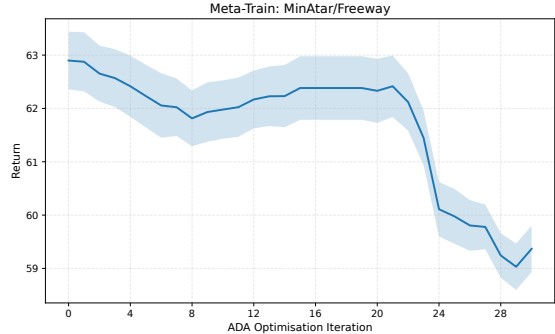

*(b)* Performance in Freeway (a meta-train environment) over the course of ADA optimisation.

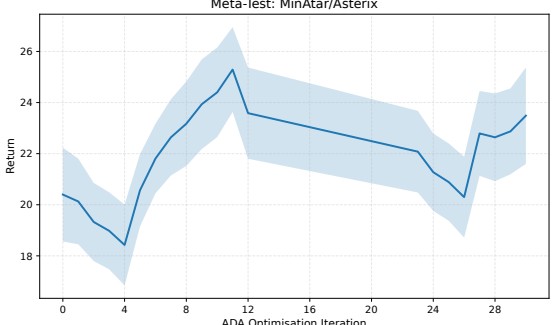

*(c)* Performance in Asterix (a meta-test environment) over the course of ADA optimisation.

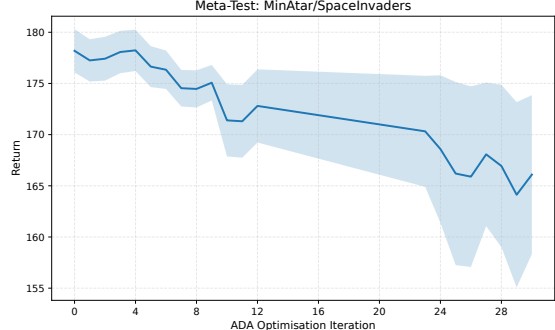

*(d)* Performance in Space Invaders (a meta-test environment) over the course of ADA optimisation.

*Figure 5.* Visualisation of performance for the four environments that are used in the $K_{tasks} = 1$ prompt optimisation experiment. We plot mean and standard error of the evaluation return achieved by the final policy for each environment over 8 inner-loop seeds, as defined by the On-Policy RL task domain implementation.

Given prompt optimisation using a language model is effectively a search problem, the curves are (expectedly) noisy. However, there is a general trend of improvement. Considering these curves in parallel with the final prompt (Section K.3.1) helps elucidate these curves. In particular, the prompt emphasises *improvement* in Breakout and *maintaining* in Freeway. Furthermore, the vast majority of the prompt is dedicated to scores in Breakout; it suggests making changes based on the variance of Breakout, for instance.

# H. Example Discovered Algorithms

One question to explore is whether ADAs in DiscoGen *actually* discover new algorithms, or whether successful solutions end up reproducing baselines and subsequently tweaking hyperparameters. Searching through all discovered algorithms would be difficult, given the number of independent ADA runs in our work, for different domains and module combinations. As such, we select two interesting discovered algorithms to briefly discuss here. These are chosen as two algorithms which generalised especially well from meta-train to meta-test. We note that these algorithms were *not* the best performers in meta-train (though neither performed badly), which generally did not transfer well to meta-test.

## H.1. Language Modelling:

We firstly introduce a discovered loss function for language modelling. We include the loss function code in Example 2.

```python
from typing import Sequence

import torch.nn.functional as F
import torch
import torch.nn as nn
from typing import Dict, Optional

def compute_loss(outputs: Dict[str, torch.Tensor], labels: torch.Tensor,
        num_items_in_batch: Optional[int] = None) -> torch.FloatTensor:

    """Calculate the loss for the model's outputs against the true labels.
    Args:
        outputs (dict): The model's outputs, typically containing logits {"logits":
    logits}.
        labels (torch.Tensor): The true labels for the batch.
        num_items_in_batch (int, optional): The number of items in the batch. Defaults
     to None.

    Returns:
        torch.FloatTensor: The computed loss value.
    """
    logits = outputs["logits"]

    # Additive margin softmax (AM-Softmax) with margin=0.1 and scale=10
    margin = 0.1
    scale = 10.0

    one_hot = torch.zeros_like(logits).scatter_(1, labels.unsqueeze(1), 1.0)
    logits_margin = logits - margin * one_hot
    logits_margin = scale * logits_margin

    # Cross-entropy with label smoothing
    loss = F.cross_entropy(logits_margin, labels, label_smoothing=0.05)

    # Entropy regularization
    probs = F.softmax(logits, dim=1)
    entropy = -torch.sum(probs * torch.log(probs + 1e-8), dim=1).mean()
    loss = loss + 0.01 * entropy
    return loss
```

*Example 2.* A discovered loss function for language modelling.

The ADA makes a number of unintuitive design decisions that, when combined, performed close to the maximum in both meta-train and meta-test. Firstly, the agent uses additive margin softmax (Wang et al., 2018), a technique used in face verification to encourage large margins between network outputs, which is generally undesirable in language modelling where synonyms should be treated similarly. This is used in the cross-entropy calculation with a scale of 10; effectively making the target distribution significantly more sharp. Secondly, the agent introduces conflicting objectives: label-smoothing, which optimises towards *reduced* certainty in outputs (Szegedy et al., 2016), and an entropy penalty which *encourages* certainty. This all combines to give an interesting loss function; one which encourages and penalises certainty

simultaneously.

## H.2. On-Policy RL: A CNN-MLP Dual-Path Architecture For MinAtar

We also explore a dual-path network architecture (Chen et al., 2017) for on-policy reinforcement learning in MinAtar (Young & Tian, 2019). We provide discovered python code in Example 3, removing any redundant code for clarity. In the DiscoGen setup, the activation is imported from a separate file.

```python
from typing import Sequence

import distrax
import flax.linen as nn
import jax
import jax.numpy as jnp
import numpy as np
from flax.linen.initializers import constant, orthogonal

class ActorCritic(nn.Module):
    action_dim: Sequence[int]
    config: dict

    @nn.compact
    def __call__(self, x):
        """Insert your network logic here."""
        # Input = x. x is the environment observation.
        # x shape: (batch, 400) after flatten.
        hsize = self.config.get("HSIZE", 64)

        # Map activation string to function
        activation = nn.elu

        # Reshape to (..., 10, 10, 4)
        x_img = jnp.reshape(x, (*x.shape[:-1], 10, 10, 4))
        # Apply CNN
        cnn = nn.Conv(features=32, kernel_size=(5,5), strides=(1,1), padding='SAME',
                      kernel_init=orthogonal(np.sqrt(2)), bias_init=constant(0.0))(x_img
        )
        cnn = activation(cnn)
        cnn = nn.LayerNorm()(cnn)
        cnn = nn.Conv(features=64, kernel_size=(3,3), strides=(1,1), padding='SAME',
                      kernel_init=orthogonal(np.sqrt(2)), bias_init=constant(0.0))(cnn)
        cnn = activation(cnn)
        cnn = nn.LayerNorm()(cnn)
        cnn = nn.Conv(features=64, kernel_size=(3,3), strides=(1,1), padding='SAME',
                      kernel_init=orthogonal(np.sqrt(2)), bias_init=constant(0.0))(cnn)
        cnn = activation(cnn)
        cnn = nn.LayerNorm()(cnn)
        cnn = cnn.reshape((*cnn.shape[:-3], -1))  # flatten
        # Reduce dimension
        cnn = nn.Dense(256, kernel_init=orthogonal(np.sqrt(2)), bias_init=constant
        (0.0))(cnn)
        cnn = activation(cnn)
        cnn = nn.LayerNorm()(cnn)
        # MLP on flattened input
        mlp = nn.Dense(hsize, kernel_init=orthogonal(np.sqrt(2)), bias_init=constant
        (0.0))(x)
        mlp = activation(mlp)
        mlp = nn.LayerNorm()(mlp)
        mlp = nn.Dense(hsize, kernel_init=orthogonal(np.sqrt(2)), bias_init=constant
        (0.0))(mlp)
        mlp = activation(mlp)
        mlp = nn.LayerNorm()(mlp)
        # Concatenate
```

```
53          x = jnp.concatenate([cnn, mlp], axis=-1)
54          # Dense layer to combine
55          x = nn.Dense(hsize, kernel_init=orthogonal(np.sqrt(2)), bias_init=constant
    (0.0))(x)
56          x = activation(x)
57          x = nn.LayerNorm()(x)
58
59          # Policy head
60          logits = nn.Dense(self.action_dim, kernel_init=orthogonal(0.01), bias_init=
    constant(0.0))(x)
61          # Value head
62          v = nn.Dense(1, kernel_init=orthogonal(1.0), bias_init=constant(0.0))(x)
63
64          pi = distrax.Categorical(logits=logits)
65
66          return pi, jnp.squeeze(v, axis=-1)
```

*Example 3.* A discovered neural network architecture for on-policy reinforcement learning.

The agent makes two particularly interesting and uncommon design decisions in its discovered architecture. Firstly, rather than using a ReLU activation (Nair & Hinton, 2010), as is common when optimising the PPO objective (Schulman et al., 2017; Huang et al., 2022a), the agent uses the *exponential linear unit* (Clevert et al., 2016, ELU). Secondly, while using convolutional neural networks (LeCun et al., 2015, CNN) is common in Atari games (Mnih et al., 2015; Huang et al., 2022a), and MLP policies are often used for MinAtar (Lu et al., 2022), here the ADA combines the two. Specifically, the agent builds an architecture which learns features with both local inductive bias (from the CNN) and global features (from the MLP), effectively overcoming the shortcomings of each approach, and concatenates them before the policy- and value-head. This network architecture performed exceptionally well, on average, in all four MinAtar environments.

# I. Additional DiscoBench Analysis

## I.1. Success@3 Results

We believe that the principal objective of ADA research should be to develop robust and performant algorithm discovery agents. As such, in the main body of our text we report aggregated success metrics over independent seeds of algorithm discovery; an ADA which only produces a valid algorithm in one out of its three seeds would receive a success rate of 33%. However, such reporting introduces the question of whether tasks are too difficult to provide signal for either evaluation or optimisation. As such, in Table 6 we report *success@3*. This acts as a complement to Table 3 and verifies that, for most tasks, agents are able to create *at least one* valid solution.

*Table 6. Success@3 rates in DiscoBench. Bold indicates best performance.*

| | DiscoBench (Single Edit) | DiscoBench (All Edit) |
|---|---|---|
| **Model** | **Success@3** | **Success@3** |
| GPT-OSS 120B | 82.1% | 26.5% |
| Devstral2 | 79.7% | 46.2% |
| Deepseek-v3.2 | **89.0%** | **63.1%** |

*Success@3* illuminates that ADAs are *able* to produce valid solutions to most tasks, albeit not consistently. As such, the large gap between average success over 3 seeds, compared to *success@3* broadly stems from low ADA robustness as opposed to an insurmountable set of tasks in DiscoBench. Furthermore, by extrapolating these results from DiscoBench to DiscoGen, they confirm that there is *signal* for ADA optimisation; as long as models can sometimes produce valid solutions, there is sufficient evidence to differentiate between meta-loop trajectories, and thus to optimise ADAs.

## I.2. Successful Seeds Analysis

We include additional results over a subset of domains, where affordable and feasible, until each ADA has 3 **successful** seeds (i.e., 3 independent runs with syntactically valid submissions). Such results allow us to disentangle Elo penalties from *task failure* and Elo comparisons from *better or worse solutions*. These results are included in Table 7.

*Table 7.* ADA evaluation performance in DiscoBench (Elo Scores with 95% CIs). Bold indicates best mean performance.

| | DiscoBench (Until Success) | |
|---|---|---|
| **Model** | **Meta-Train** | **Meta-Test** |
| Baseline (All Fixed) | **1080** [1050, 1112] | **1153** [1122, 1187] |
| GPT-OSS 120B | 885 [850, 917] | 894 [867, 924] |
| Devstral2 | 986 [960, 1014] | 954 [928, 982] |
| Deepseek-v3.2 | 1049 [1026, 1074] | 998 [976, 1020] |

We find that, even after accounting for failed runs (as in our DiscoBench results from Section 5.2), models are unable to consistently outperform the all fixed baseline; in fact, the baseline is arguably *more* dominant in meta-test performance. While improving robustness of ADAs is necessary as above, it is also clearly essential to develop ADAs which produce *better* solutions when they do succeed.

# J. Per-Task DiscoBench Results

Here, we present per-task DiscoBench results, which are aggregated into a per-model ELO score over meta-train and meta-test sets. These are separated into *DiscoBench Single*, *DiscoBench All*, and *DiscoBench Single (Until Success)*. The metric is provided on the x-axis; in some cases, said metric should be *maximised*, and in some it should be *minimised*.

Results are reported over 3 or less meta-seeds, depending on how many runs were successful for each model. We highlight the baseline scores in red; these are the scores obtained by the default implementation for each task, when all files are set to fixed. For all results, we calculate bootstrap stratified confidence intervals (95%) and interquartile means, following Agarwal et al. (2021).

In our *Until Success* experiments (Appendices J.5 & J.6), we omit a small number of tasks in which we were unable to reliably produce three seeds for every model without incurring excessive cost.

## J.1. DiscoBench (Single Edit) – Meta-Train

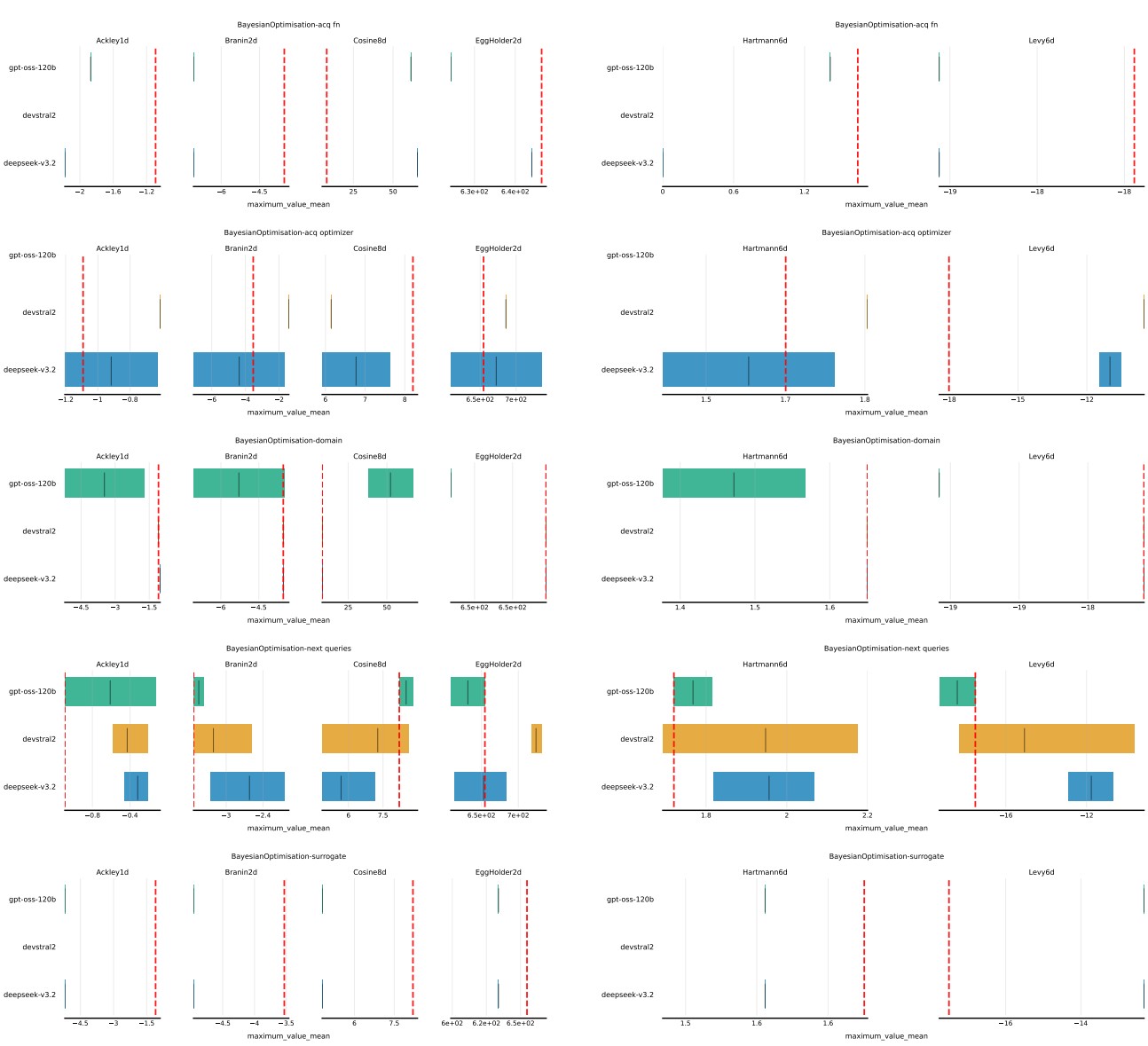

*Figure 6.* DiscoBench (Single Edit) results on Meta-Train tasks. (Part 1/7)

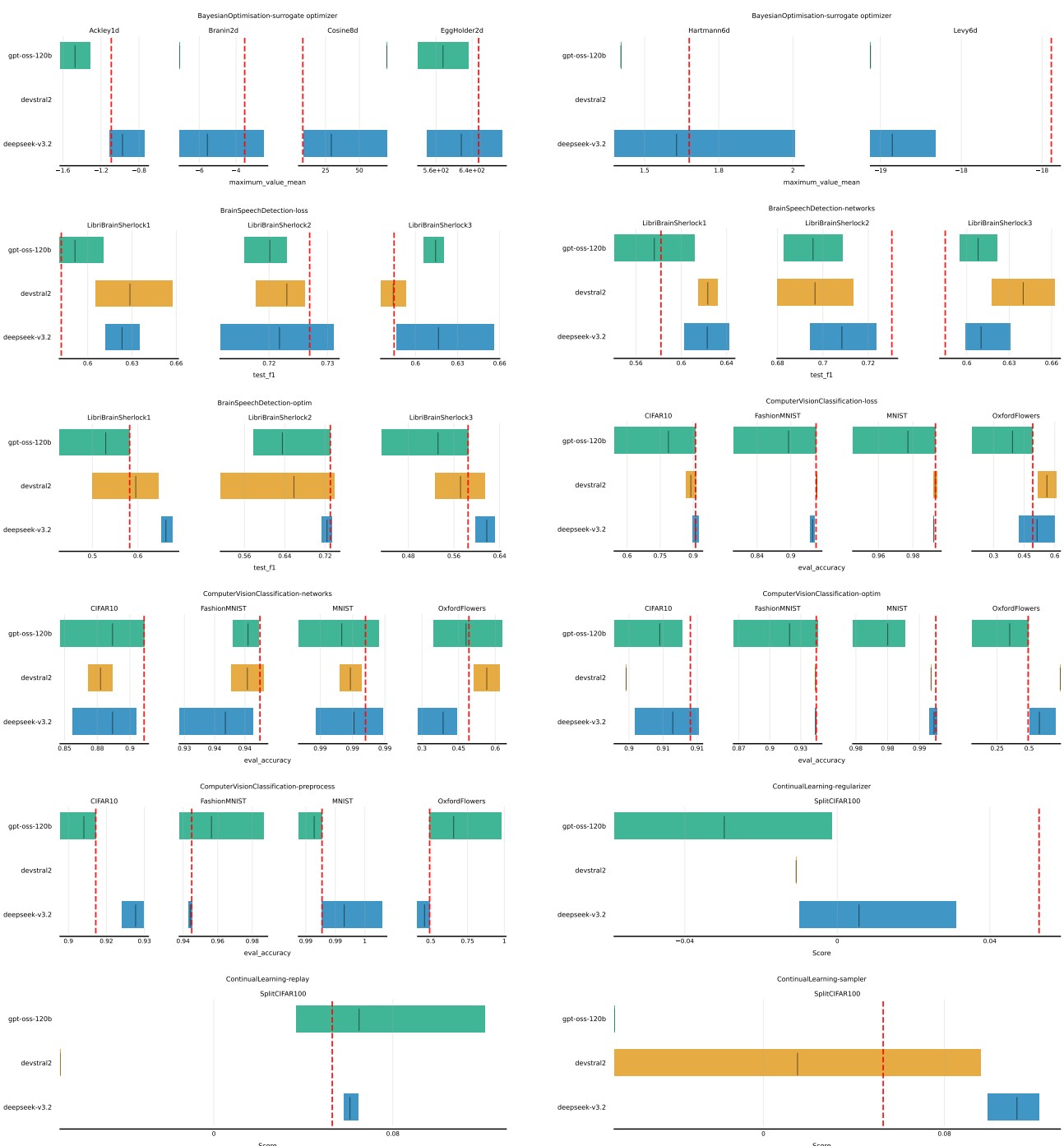

*Figure 7.* DiscoBench (Single Edit) results on Meta-Train tasks. (Part 2/7)

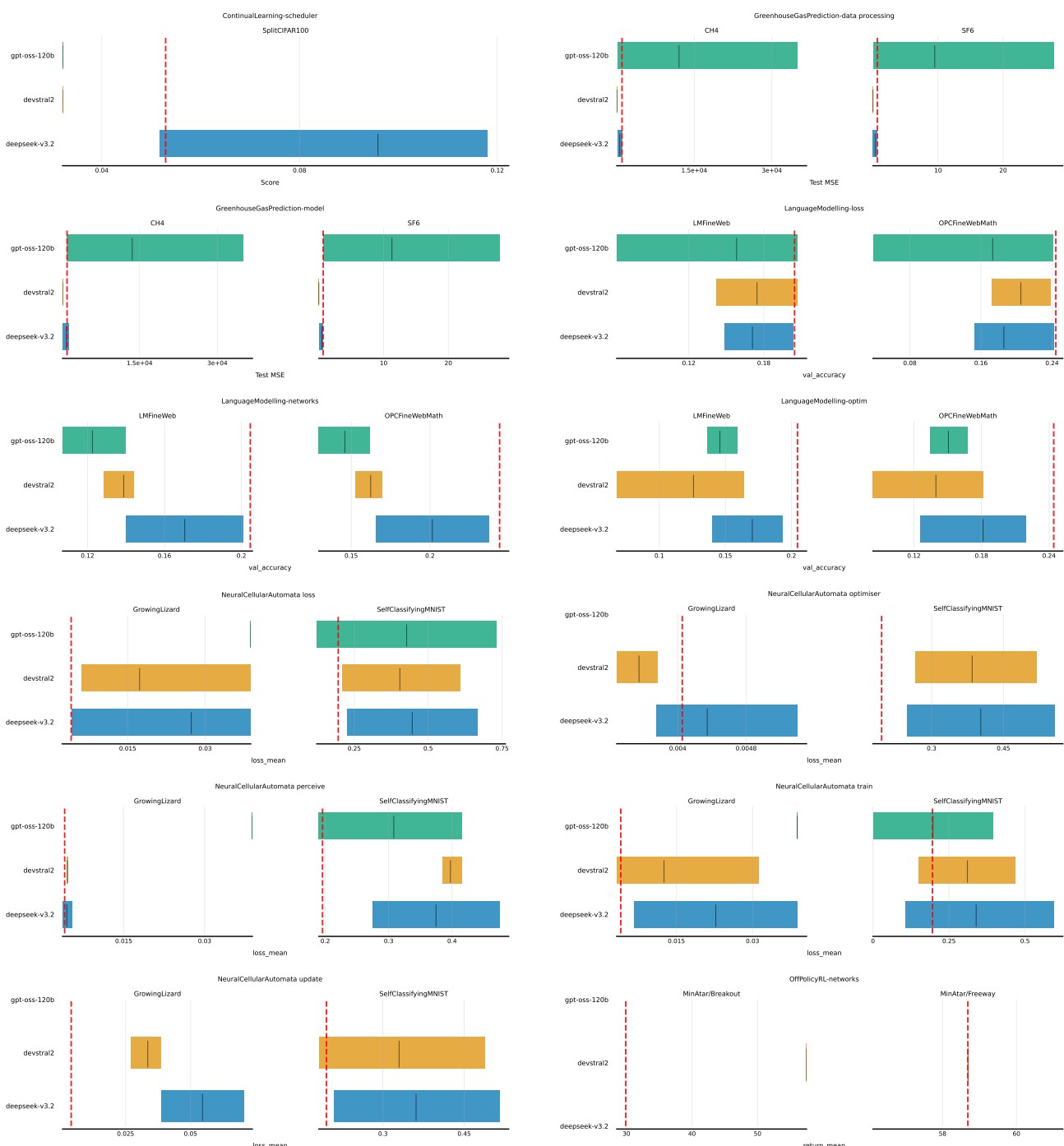

*Figure 8.* DiscoBench (Single Edit) results on Meta-Train tasks. (Part 3/7)

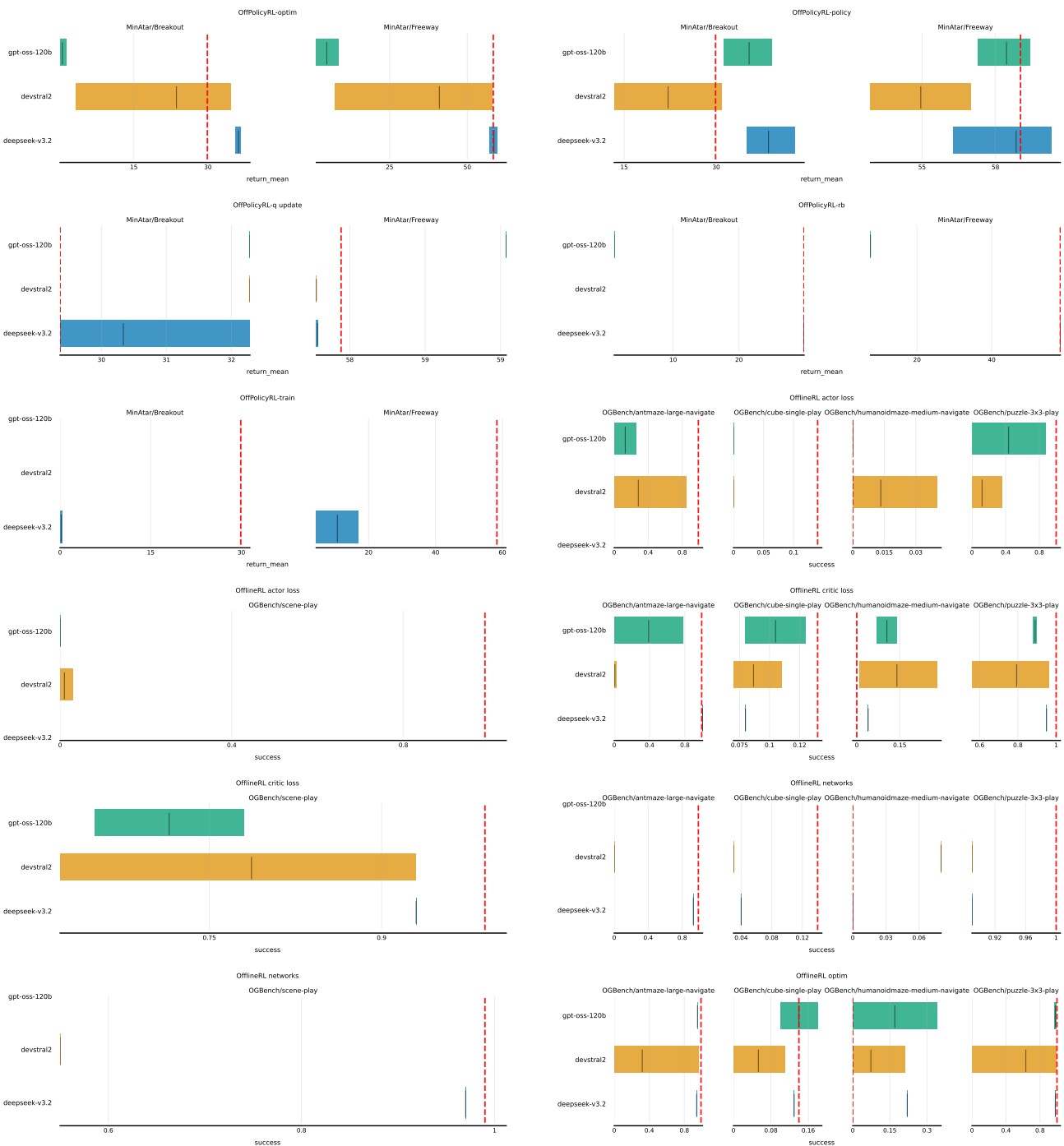

*Figure 9.* DiscoBench (Single Edit) results on Meta-Train tasks. (Part 4/7)

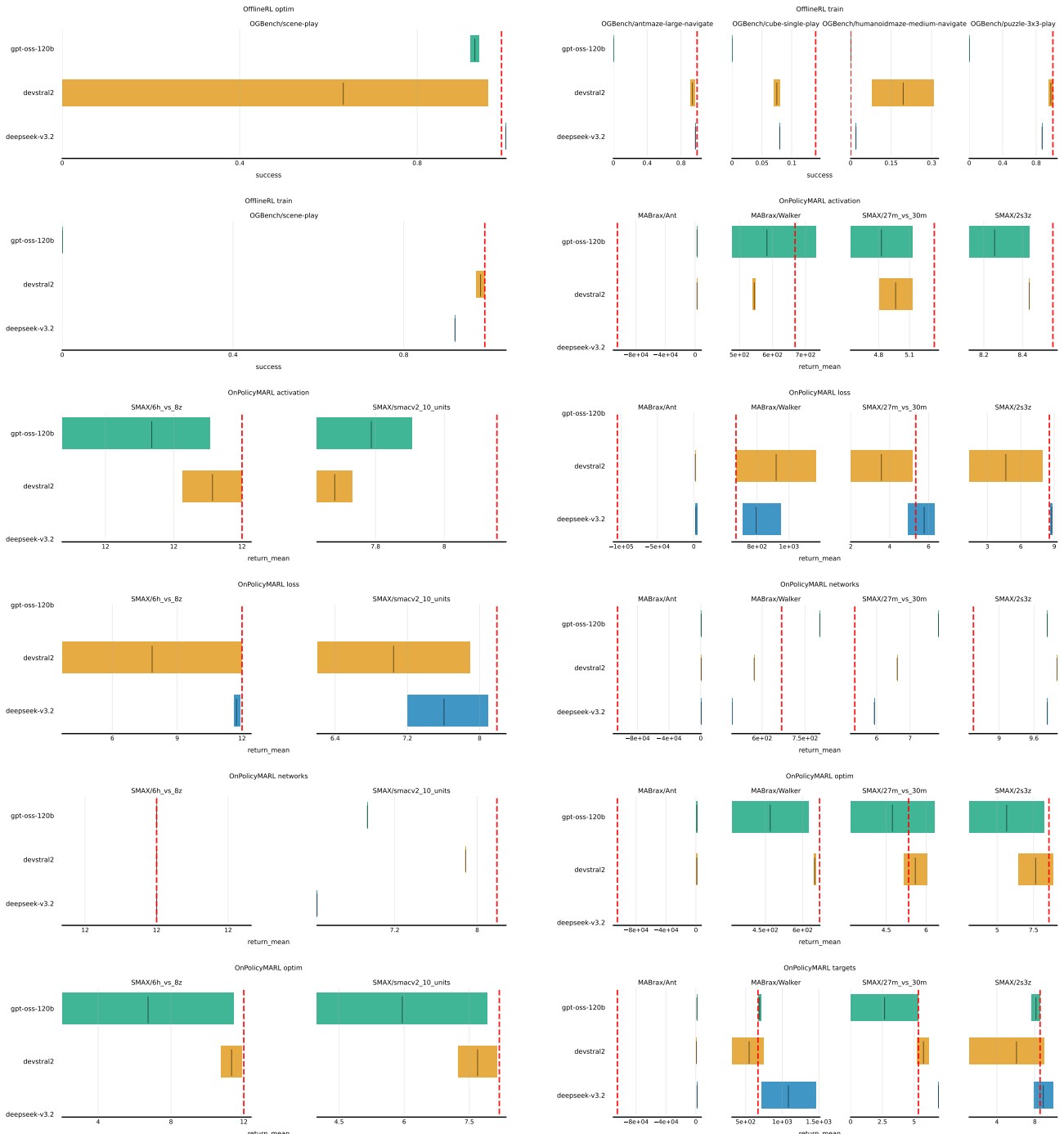

*Figure 10.* DiscoBench (Single Edit) results on Meta-Train tasks. (Part 5/7)

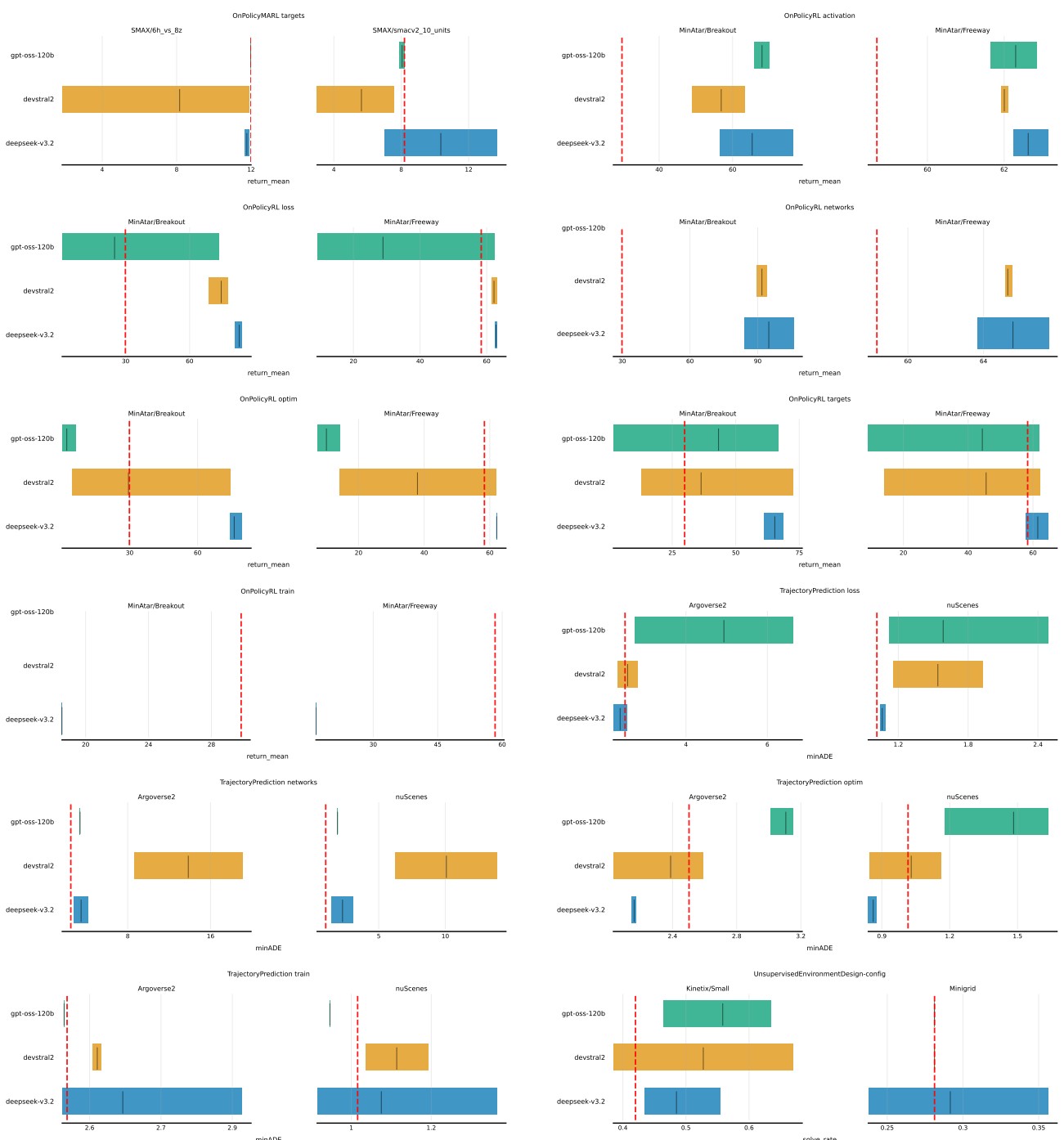

*Figure 11.* DiscoBench (Single Edit) results on Meta-Train tasks. (Part 6/7)

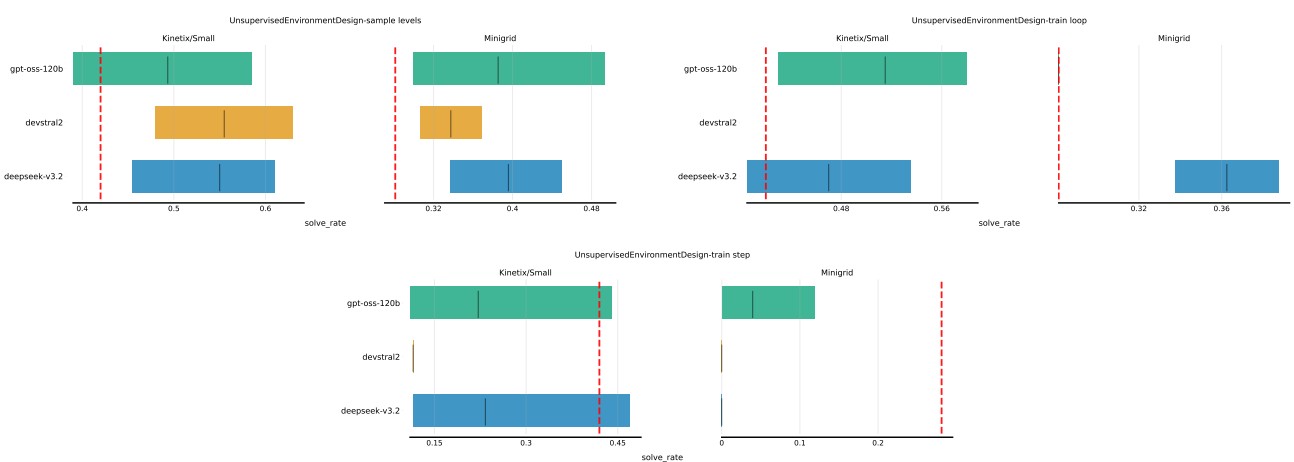

*Figure 12.* DiscoBench (Single Edit) results on Meta-Train tasks. (Part 7/7)

## J.2. DiscoBench (Single Edit) – Meta-Test

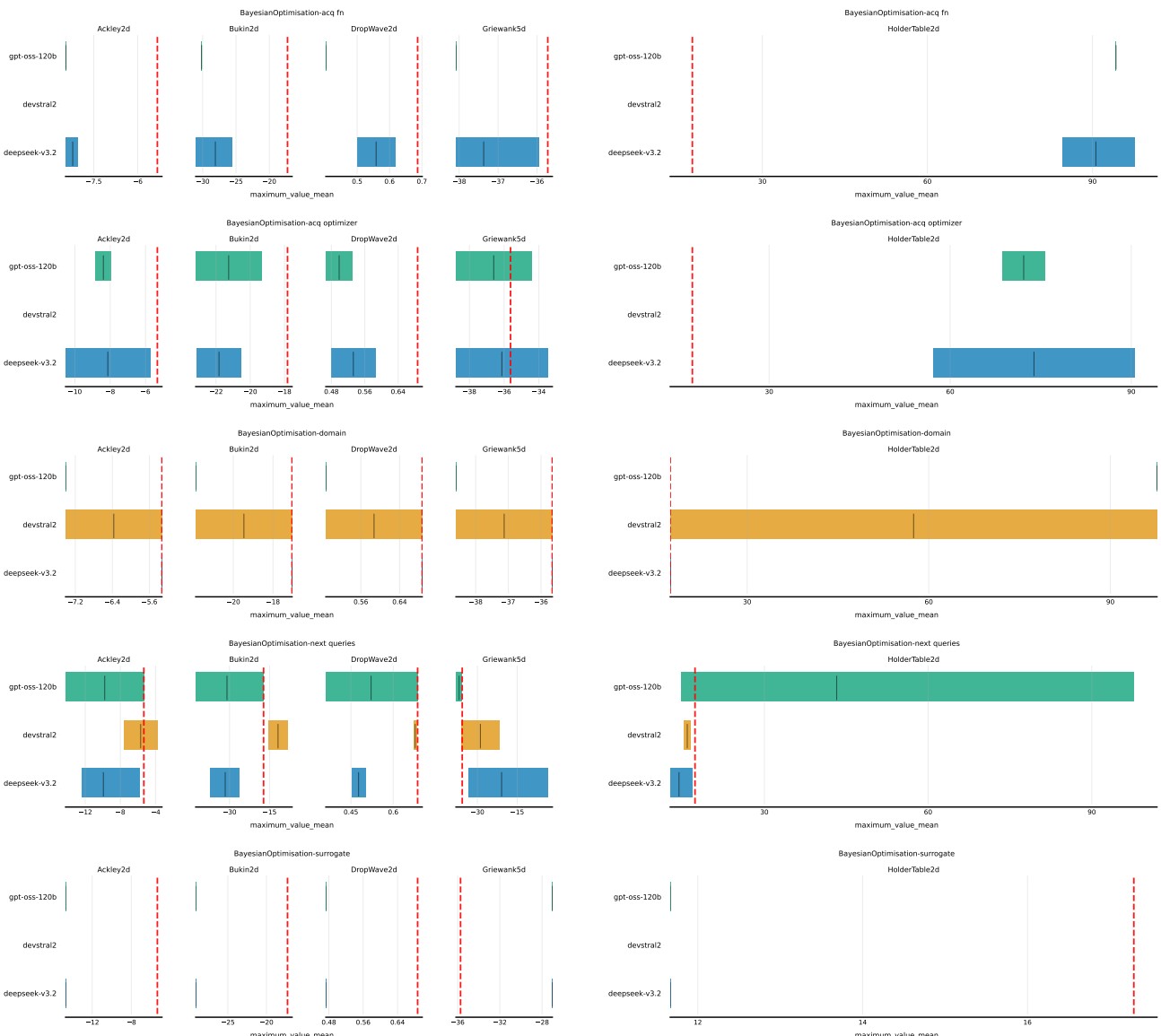

*Figure 13.* DiscoBench (Single Edit) results on Meta-Test tasks. (Part 1/7)

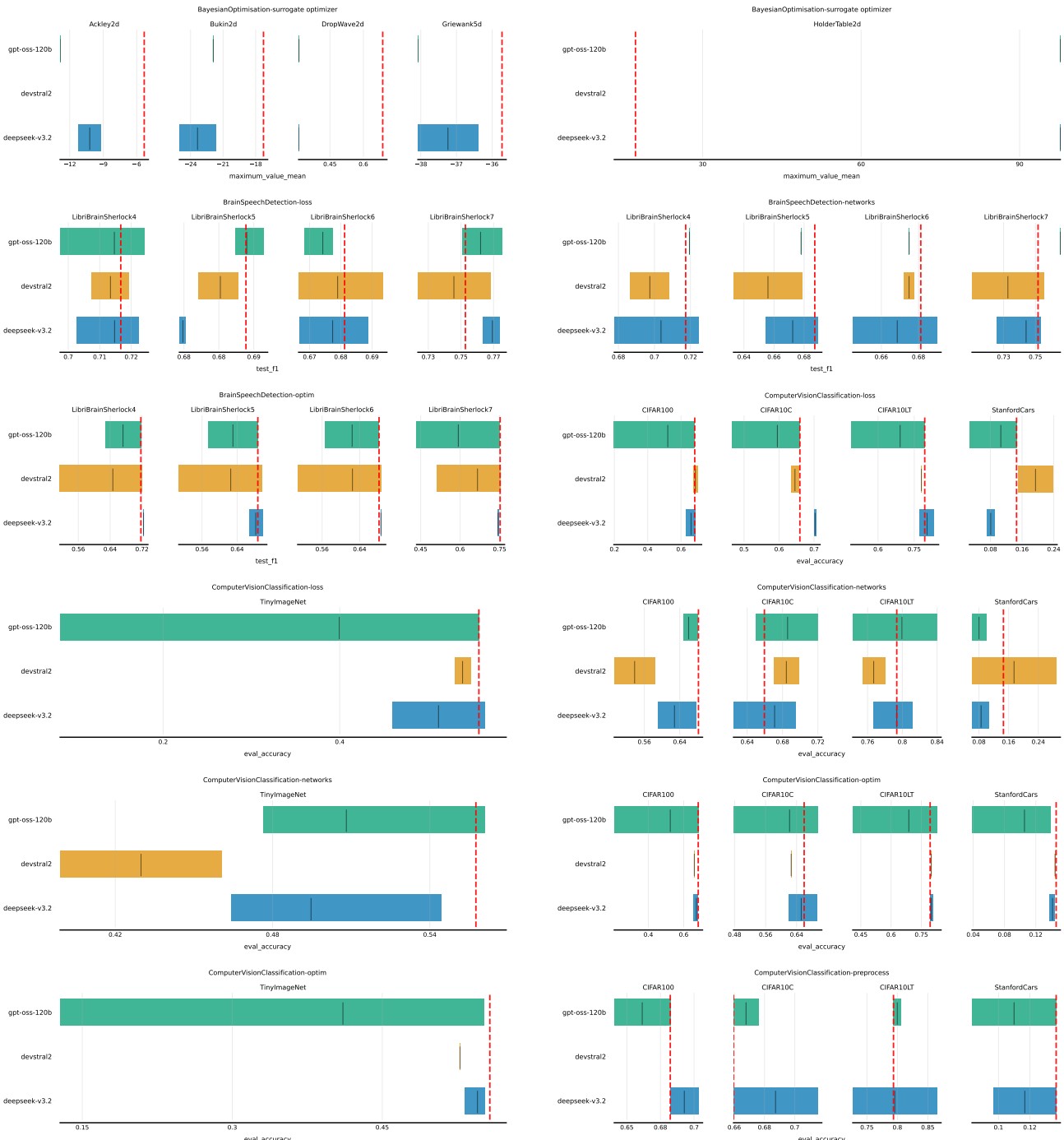

*Figure 14.* DiscoBench (Single Edit) results on Meta-Test tasks. (Part 2/7)

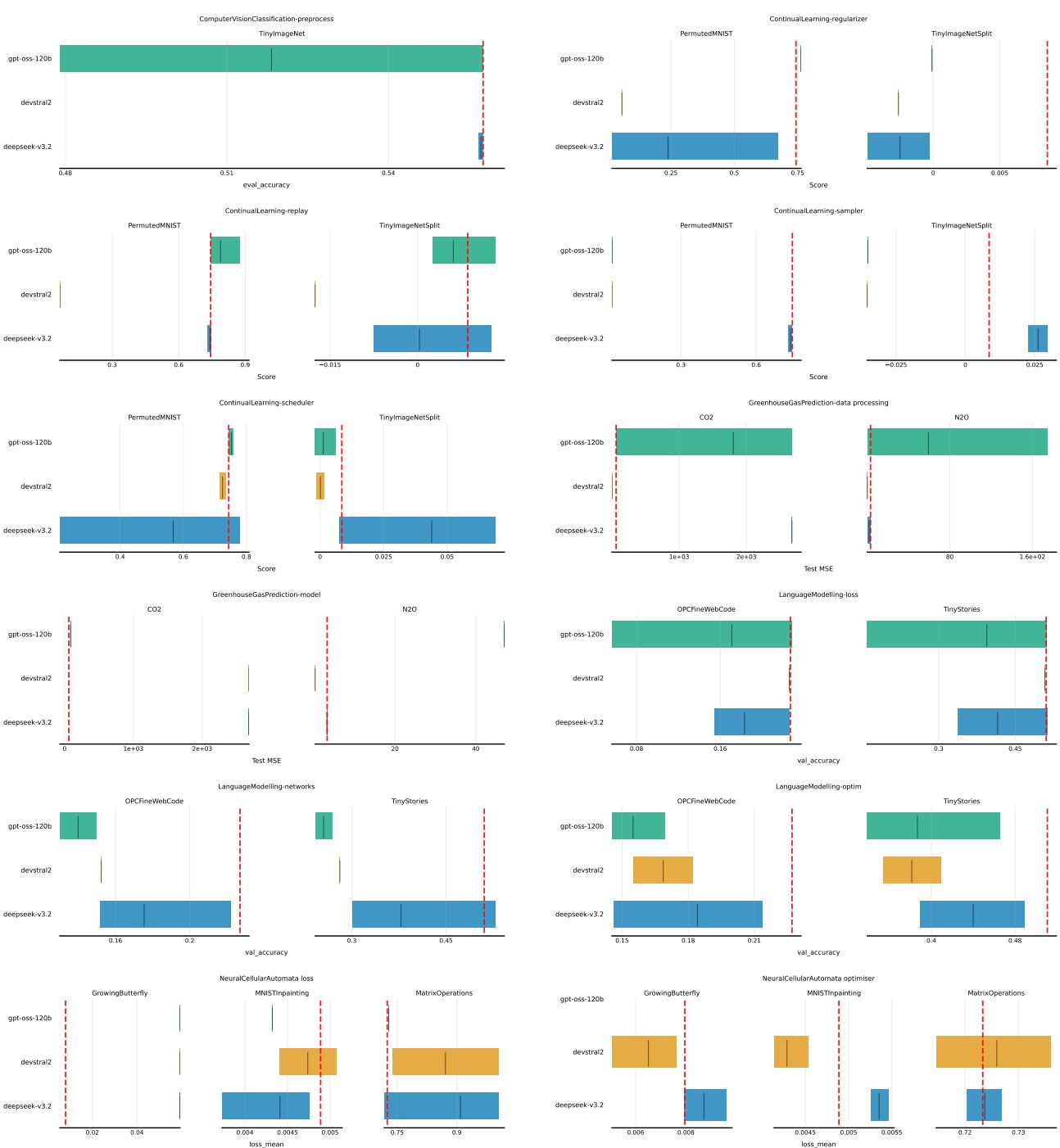

*Figure 15.* DiscoBench (Single Edit) results on Meta-Test tasks. (Part 3/7)

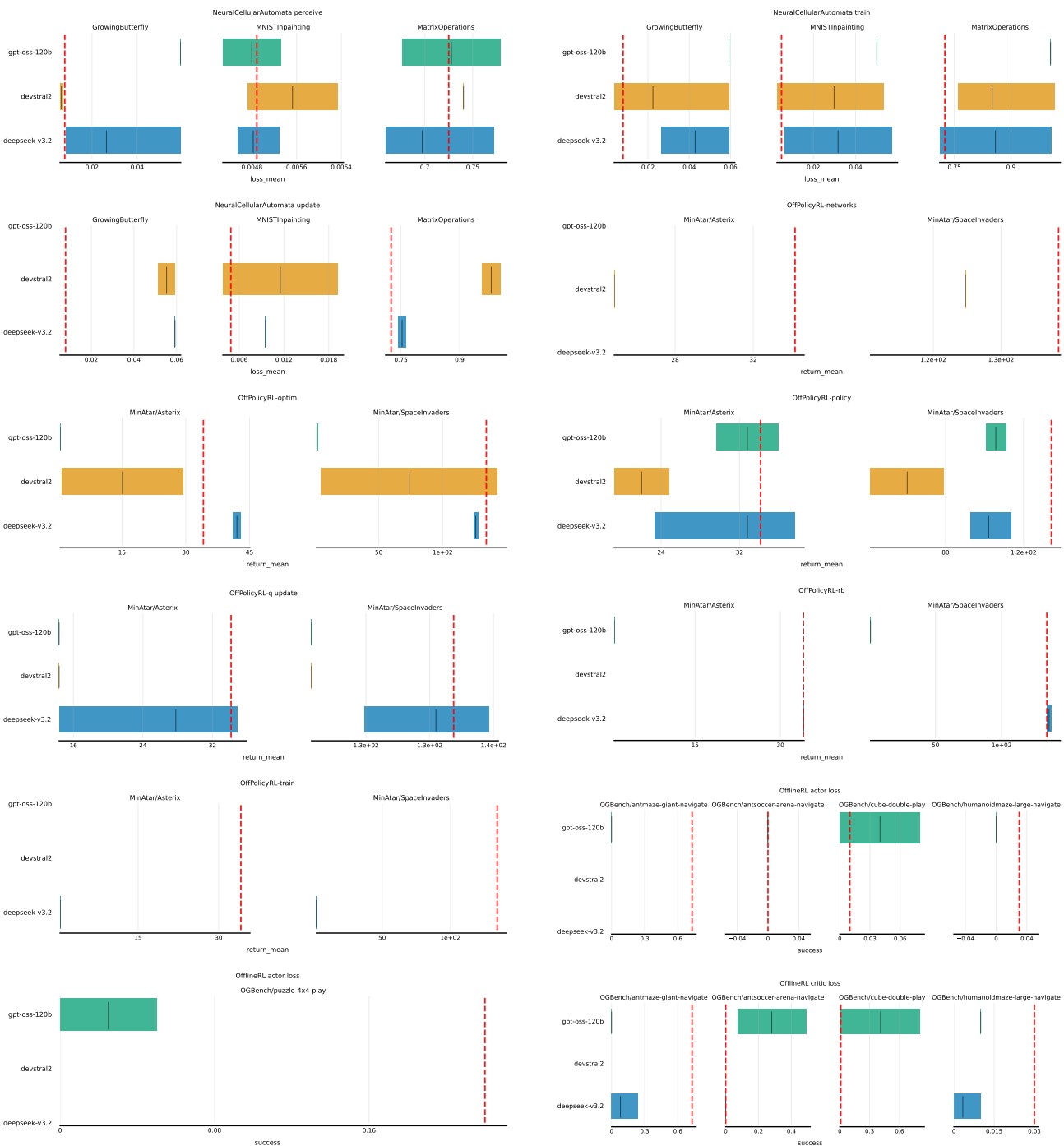

*Figure 16.* DiscoBench (Single Edit) results on Meta-Test tasks. (Part 4/7)

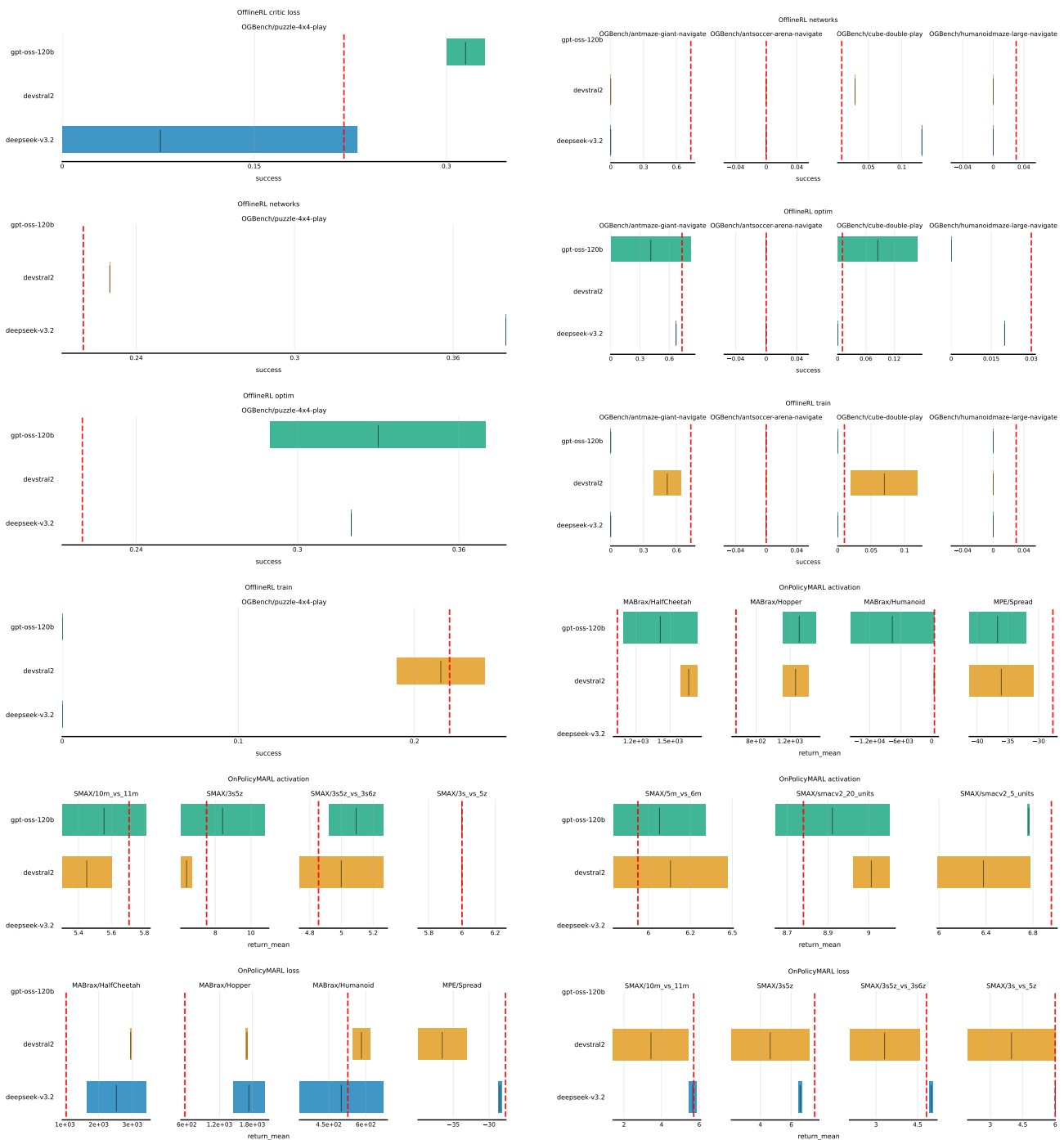

*Figure 17.* DiscoBench (Single Edit) results on Meta-Test tasks. (Part 5/7)

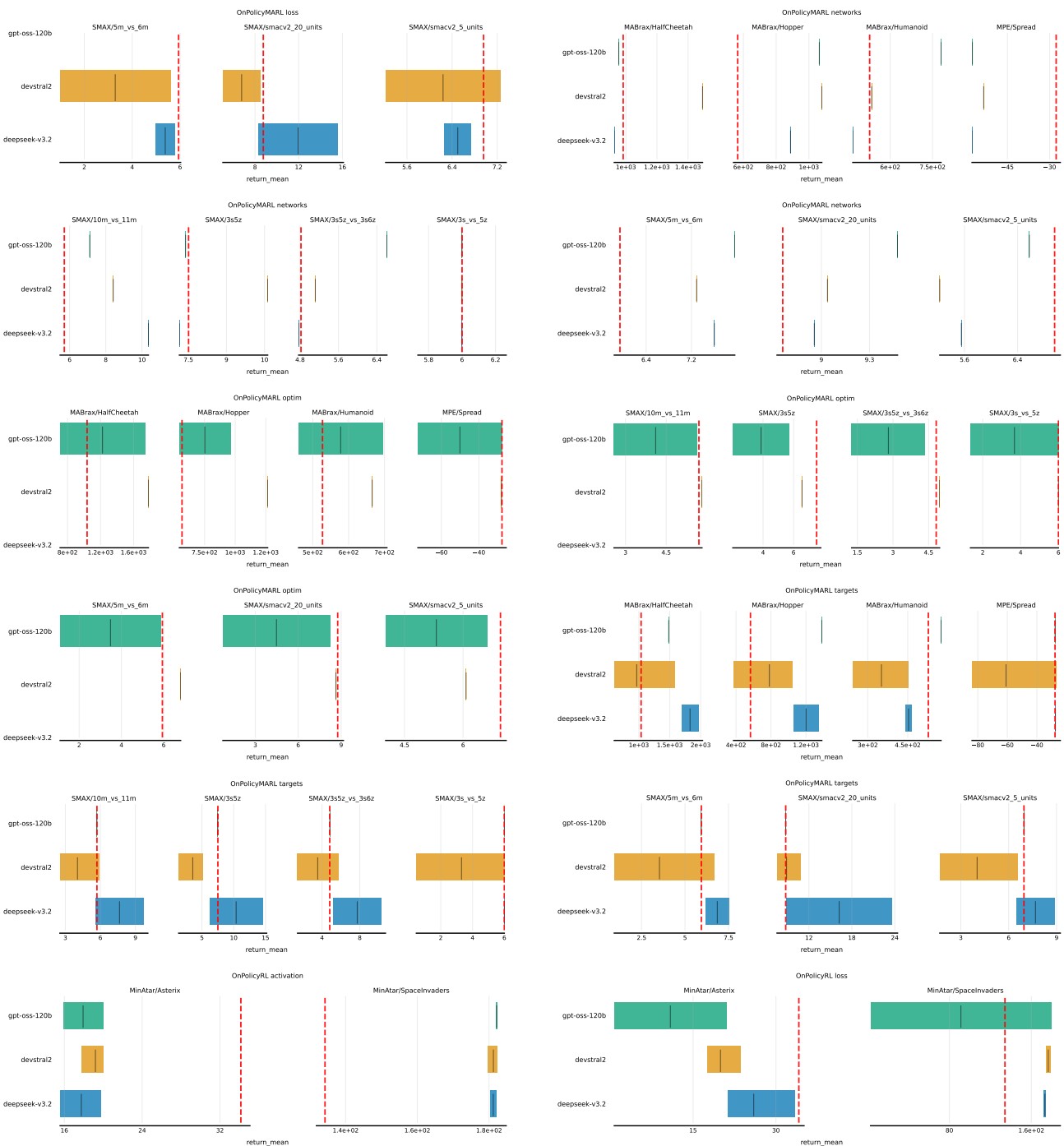

*Figure 18.* DiscoBench (Single Edit) results on Meta-Test tasks. (Part 6/7)

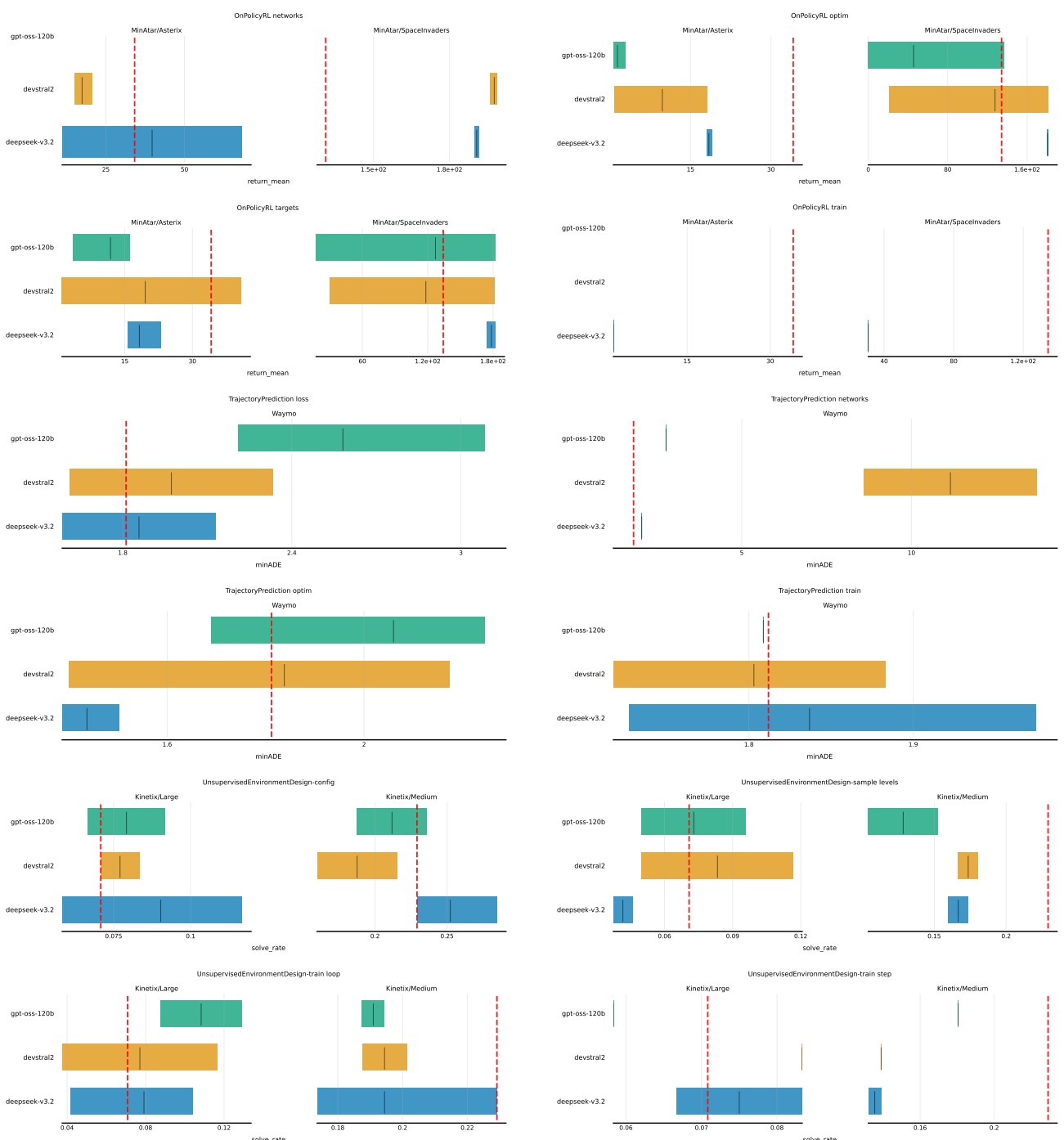

*Figure 19.* DiscoBench (Single Edit) results on Meta-Test tasks. (Part 7/7)

## J.3. DiscoBench (All Edit) – Meta-Train

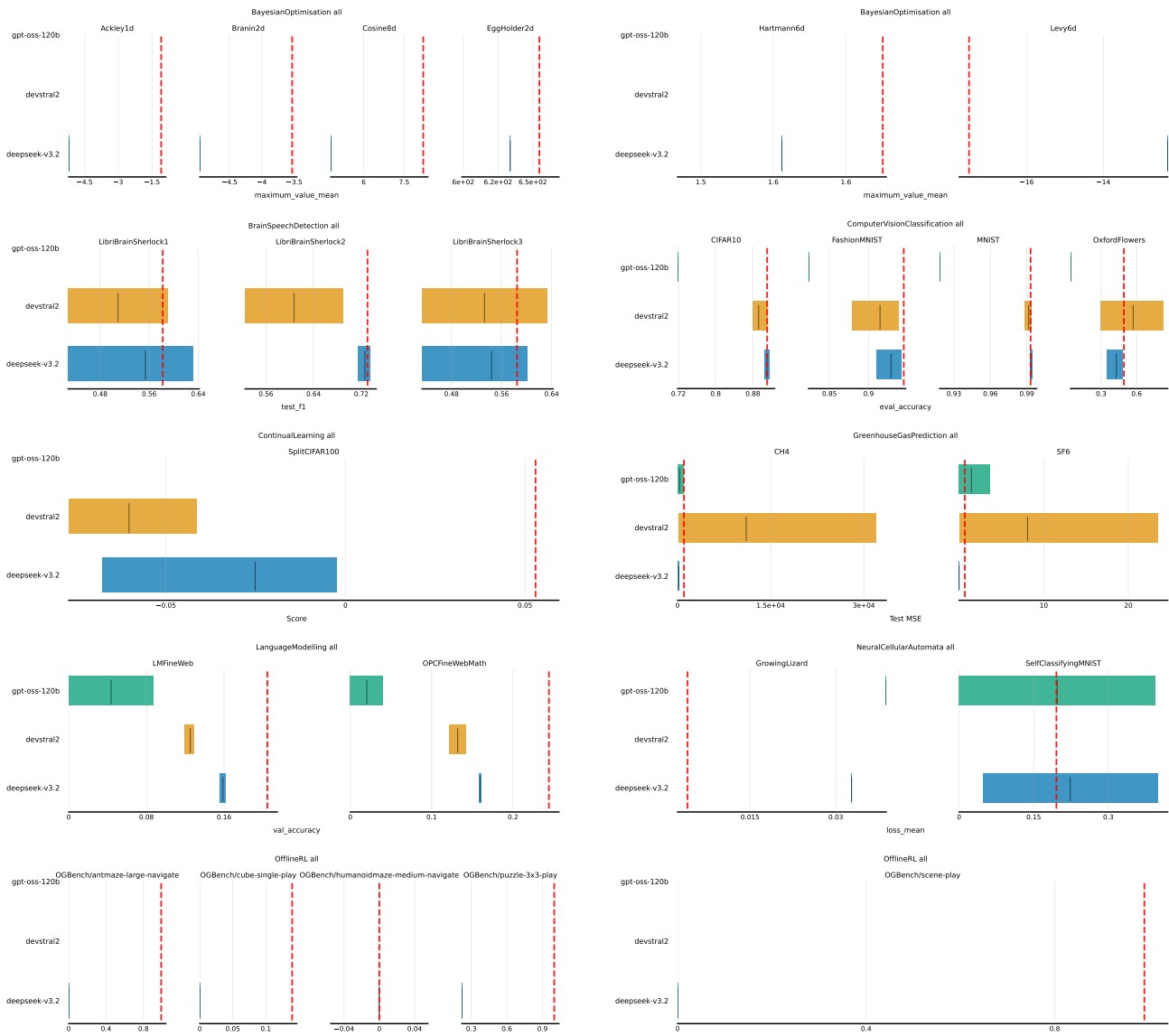

*Figure 20.* DiscoBench (All Edit) results on Meta-Train tasks. (Part 1/2)

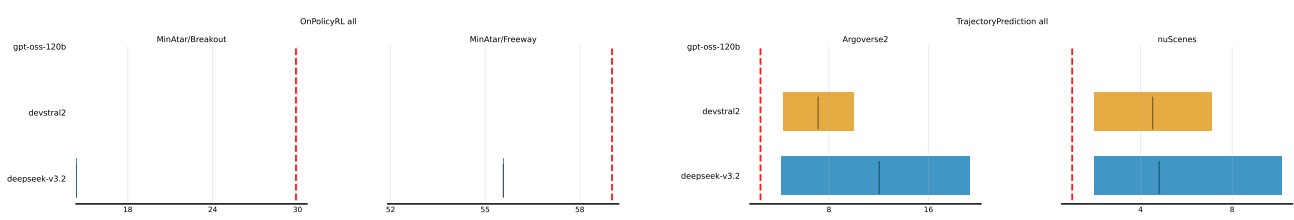

*Figure 21.* DiscoBench (All Edit) results on Meta-Train tasks. (Part 2/2)

## J.4. DiscoBench (All Edit) – Meta-Test

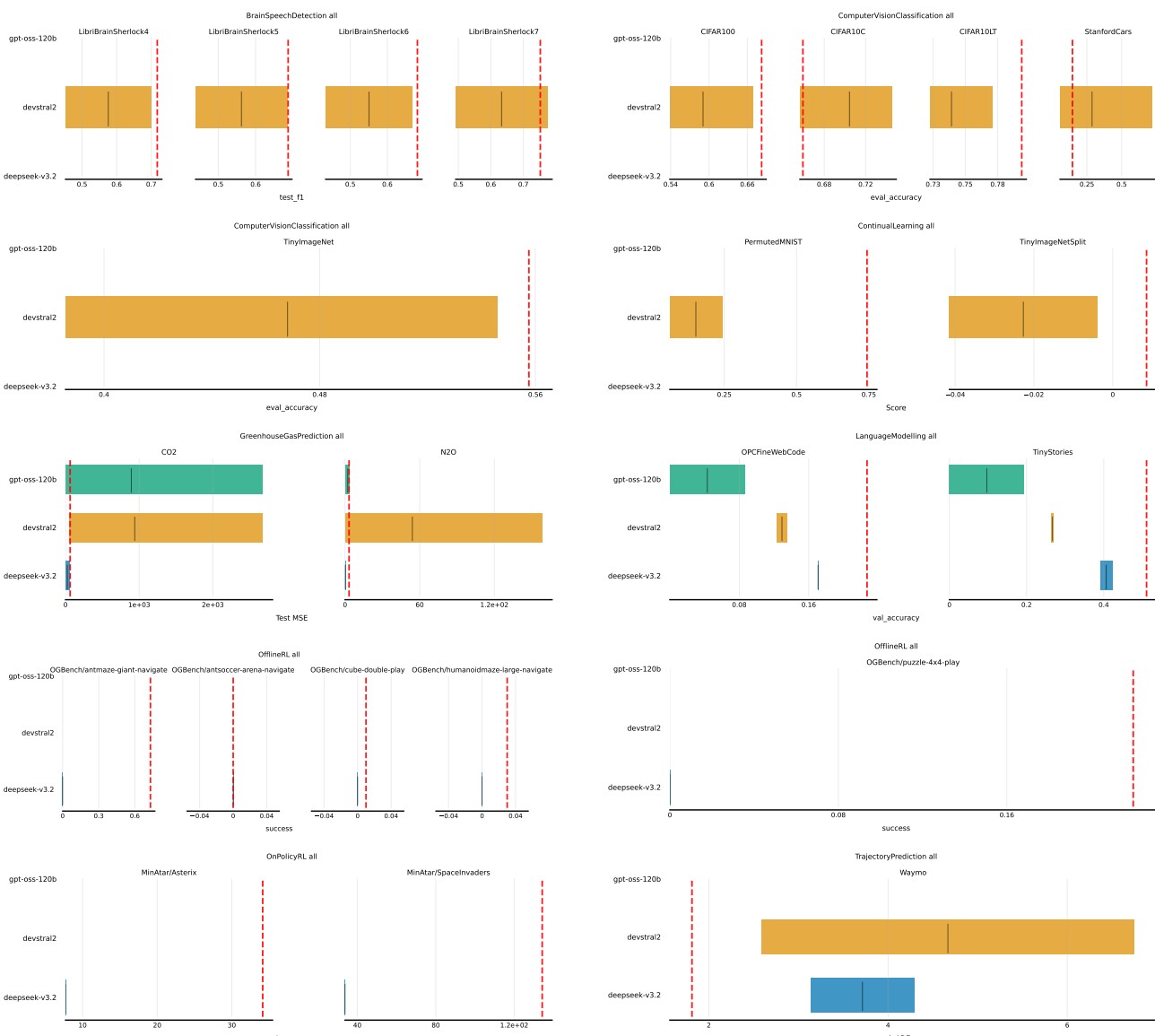

*Figure 22.* DiscoBench (All Edit) results on Meta-Test tasks.

## J.5. DiscoBench (3 Successful Seeds) – Meta-Train

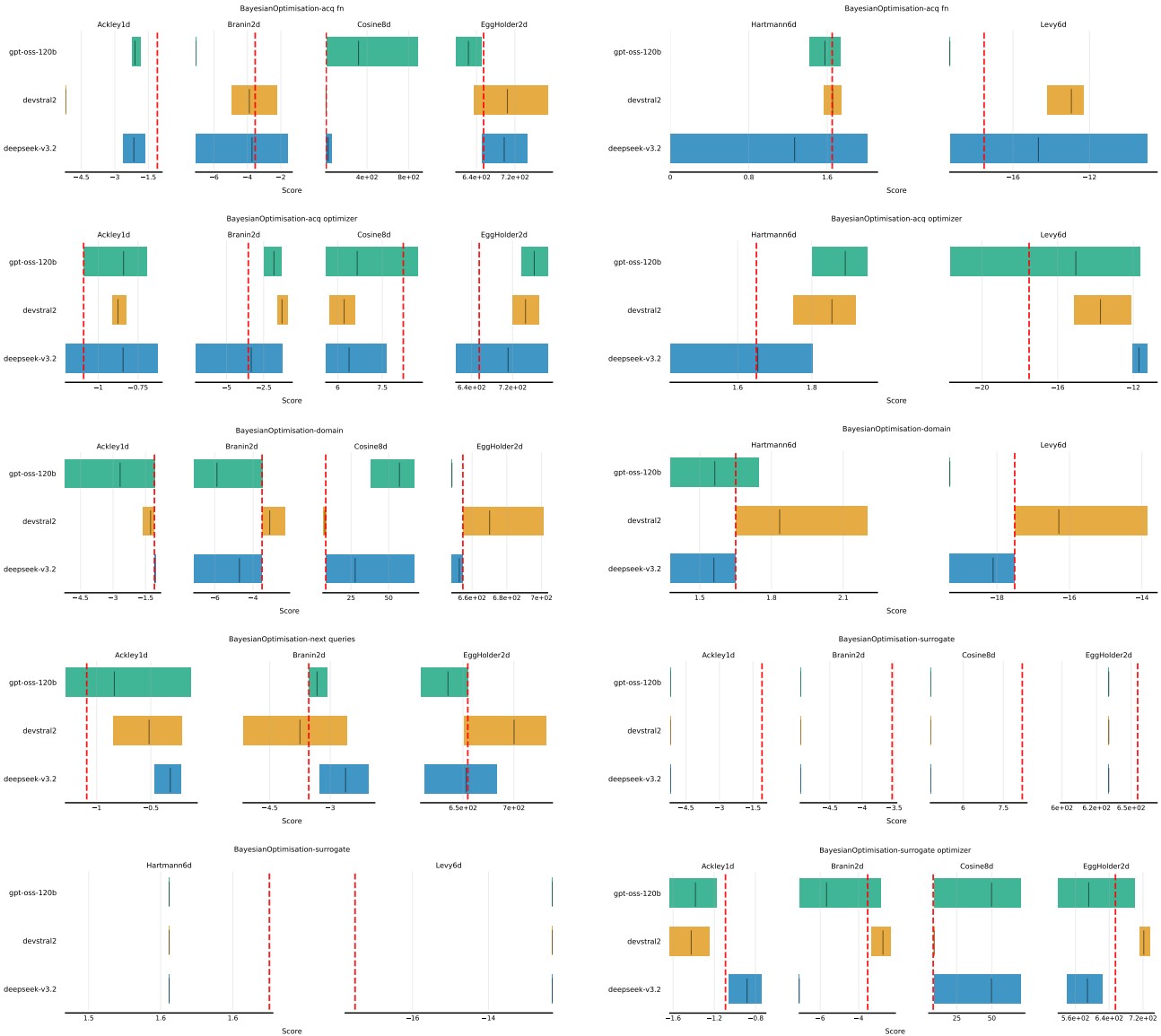

*Figure 23.* DiscoBench (3 Successful Seeds) results on Meta-Train tasks. (Part 1/4)

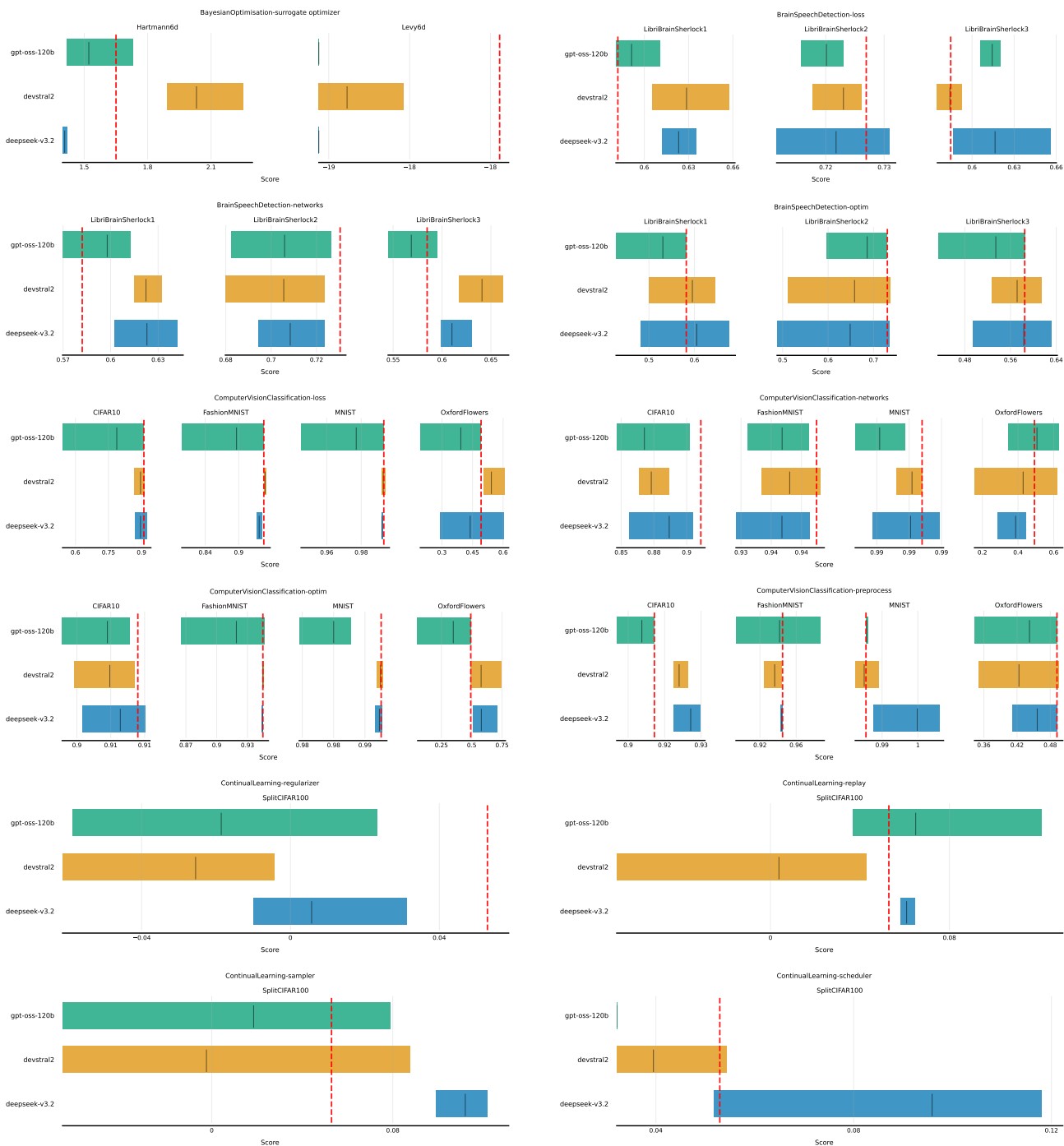

*Figure 24.* DiscoBench (3 Successful Seeds) results on Meta-Train tasks. (Part 2/4)

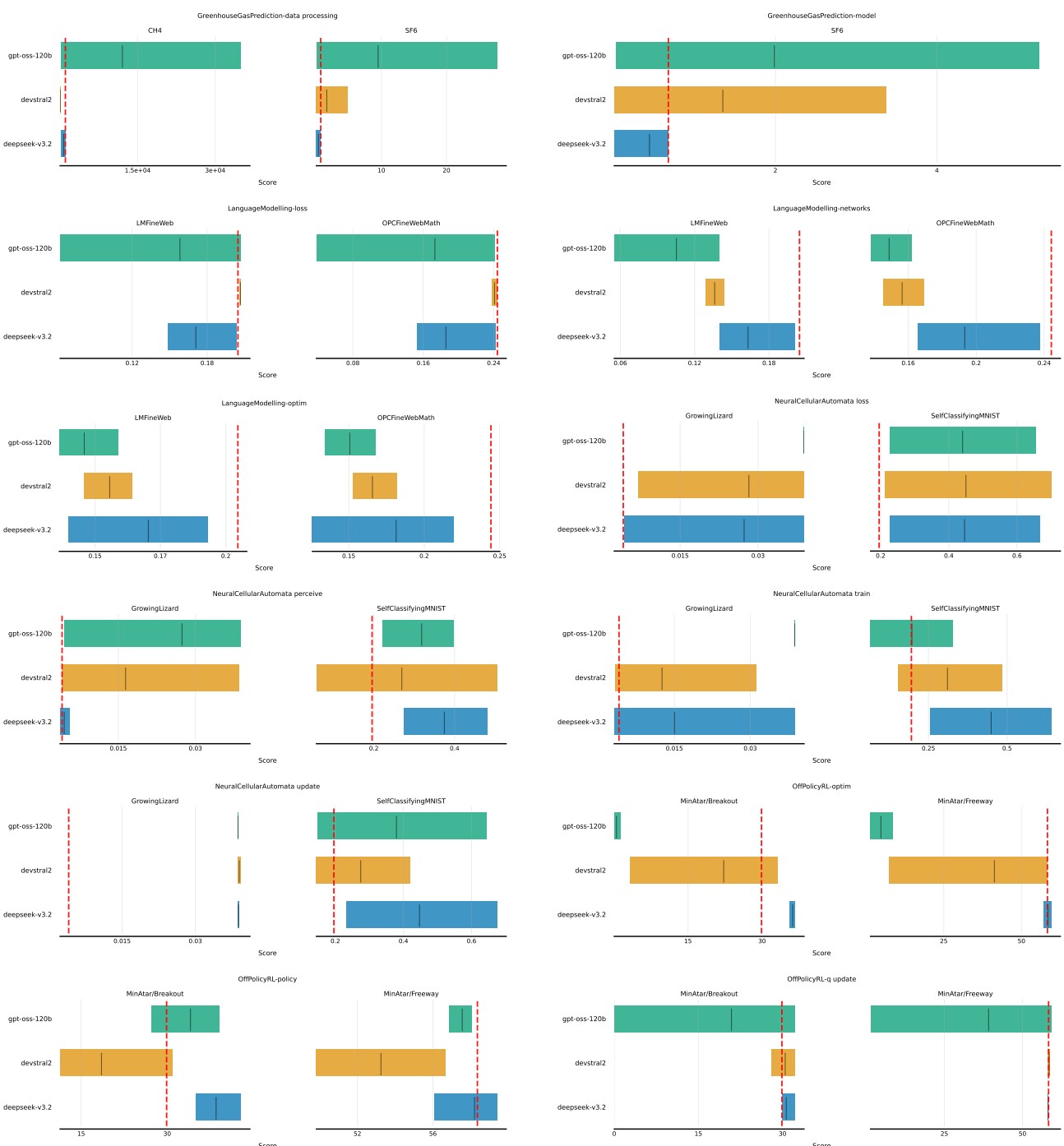

*Figure 25.* DiscoBench (3 Successful Seeds) results on Meta-Train tasks. (Part 3/4)

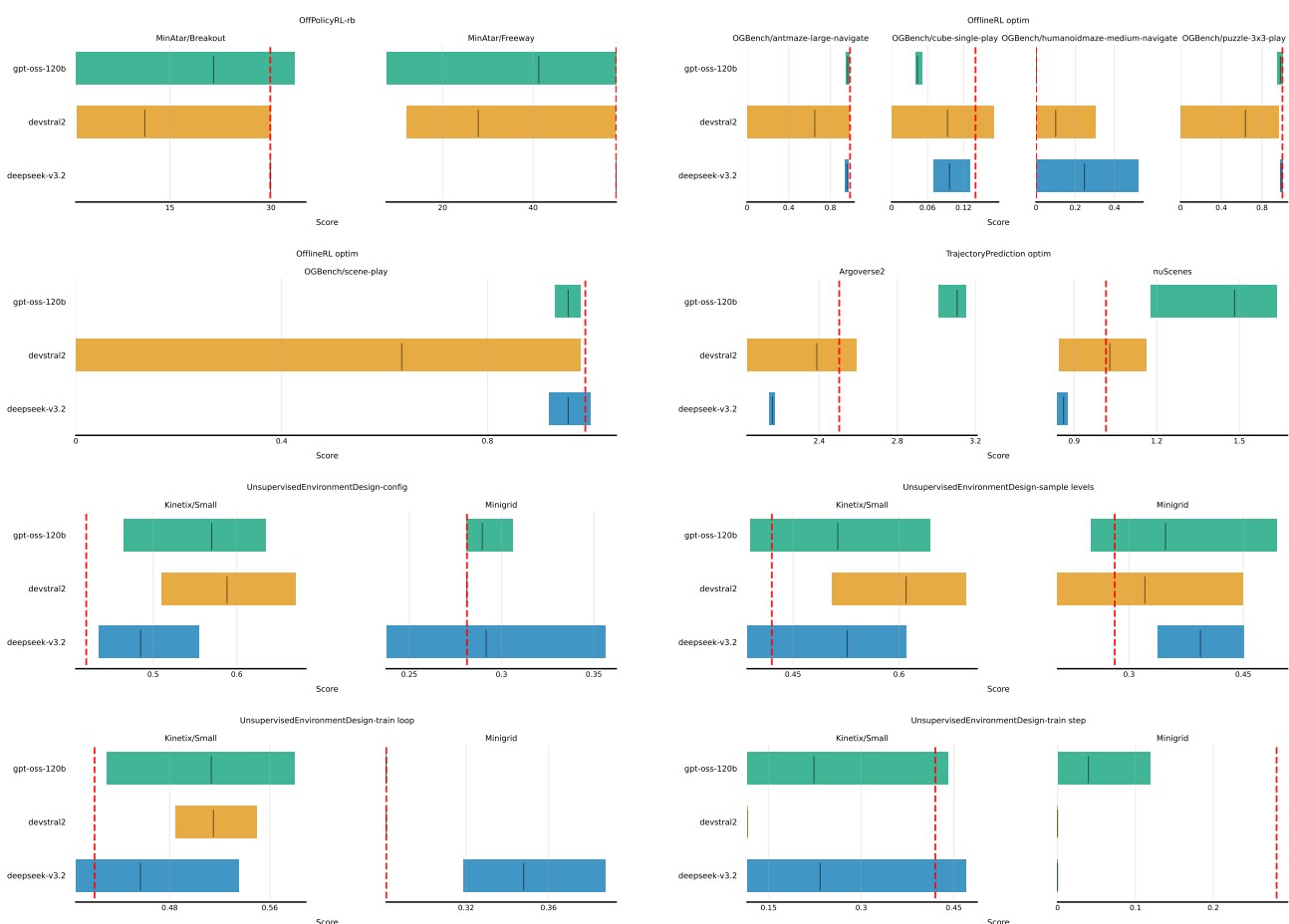

*Figure 26.* DiscoBench (3 Successful Seeds) results on Meta-Train tasks. (Part 4/4)

## J.6. DiscoBench (3 Successful Seeds) – Meta-Test

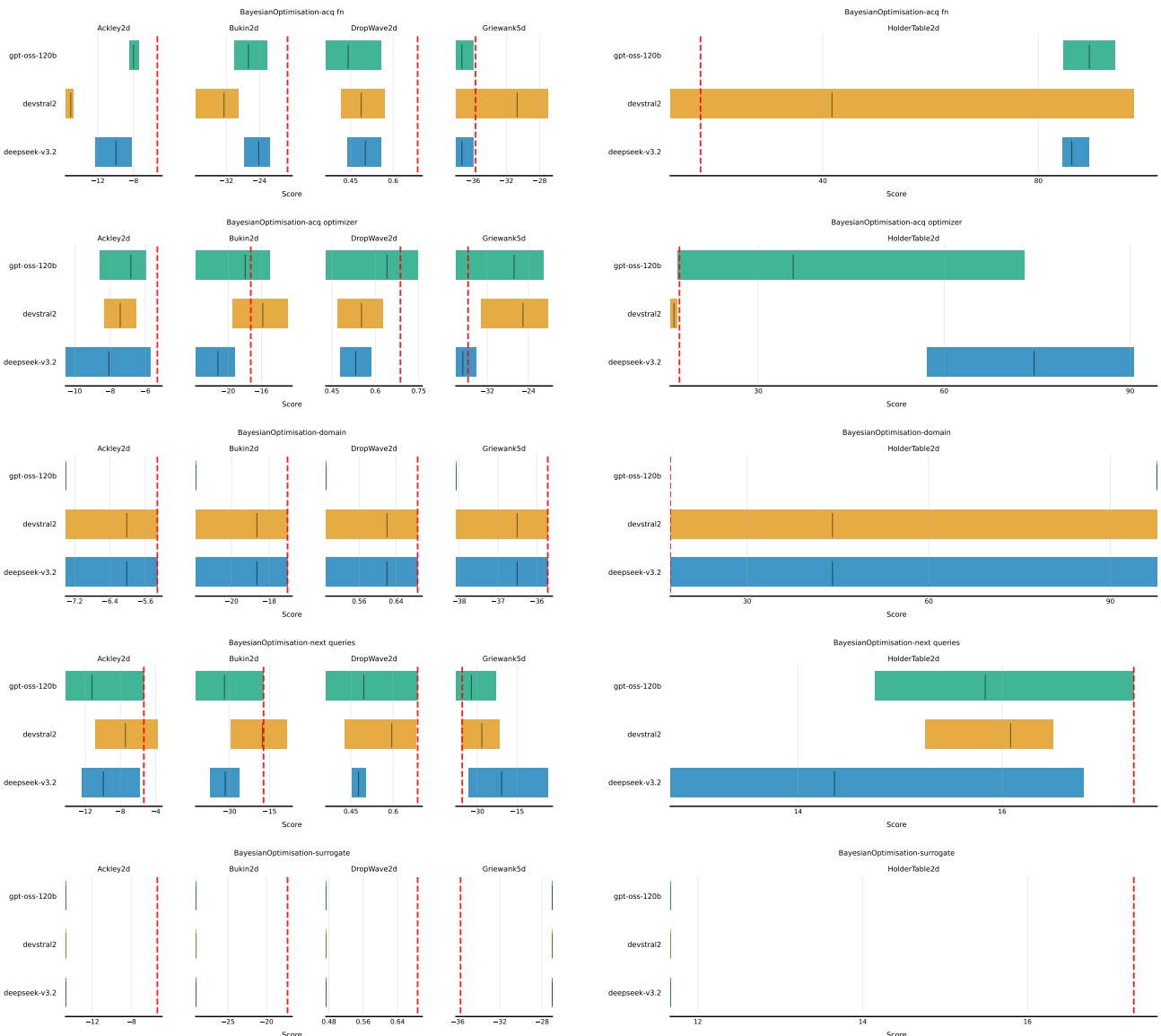

*Figure 27.* DiscoBench (3 Successful Seeds) results on Meta-Test tasks. (Part 1/5)

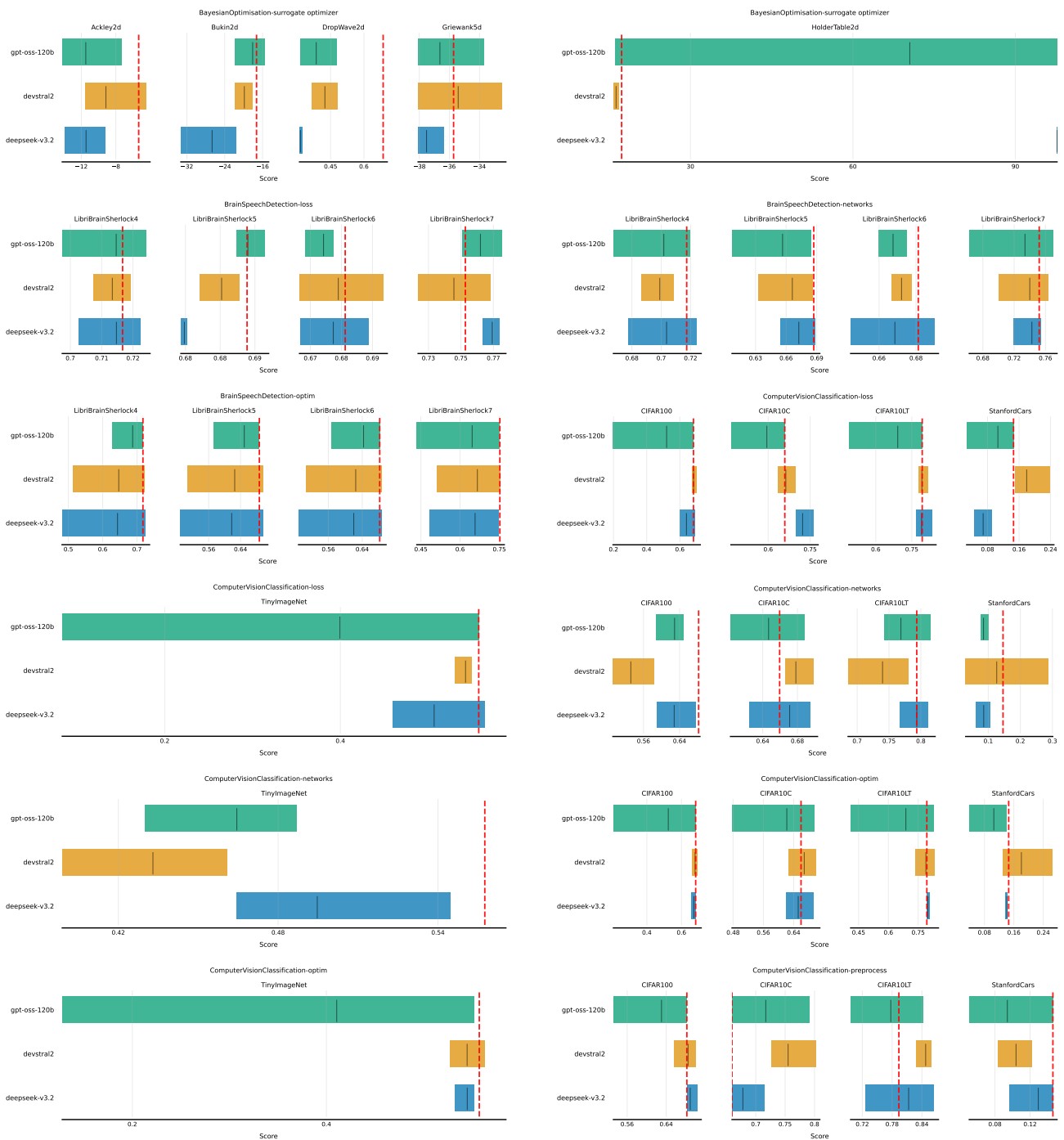

*Figure 28.* DiscoBench (3 Successful Seeds) results on Meta-Test tasks. (Part 2/5)

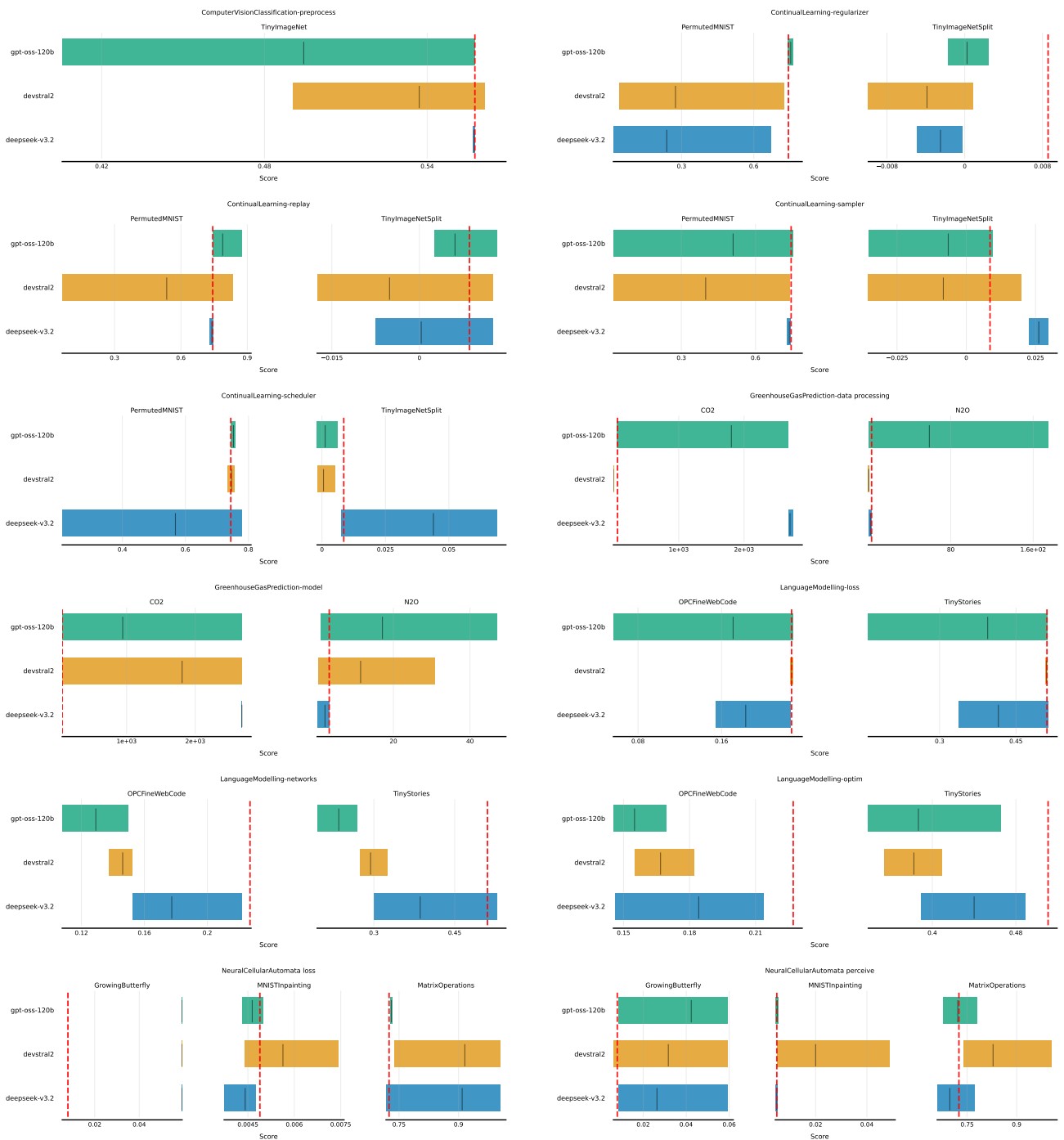

*Figure 29.* DiscoBench (3 Successful Seeds) results on Meta-Test tasks. (Part 3/5)

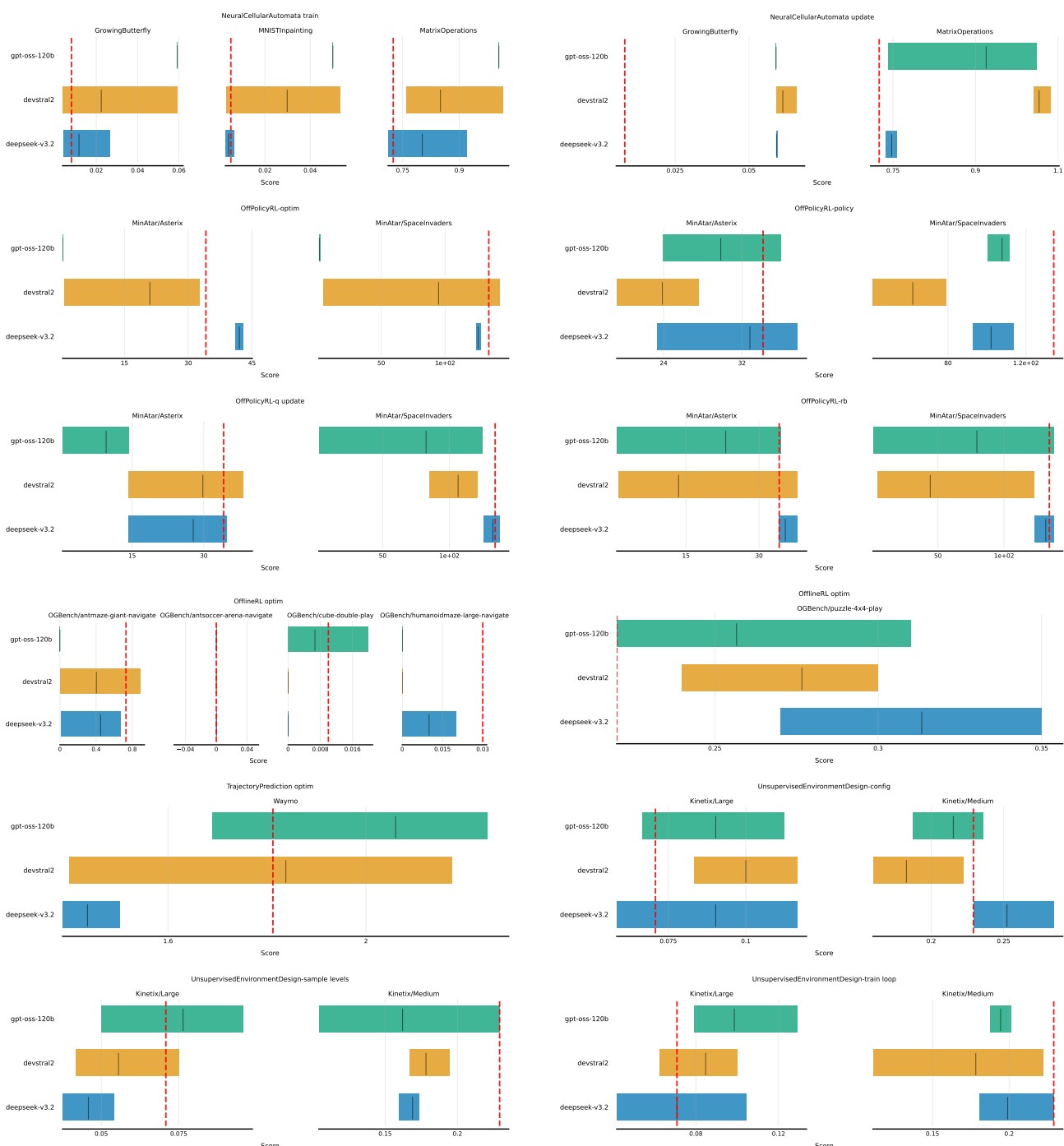

*Figure 30.* DiscoBench (3 Successful Seeds) results on Meta-Test tasks. (Part 4/5)

*Figure 31.* DiscoBench (3 Successful Seeds) results on Meta-Test tasks. (Part 5/5)

## J.7. On-Policy RL Combinations – Meta-Train

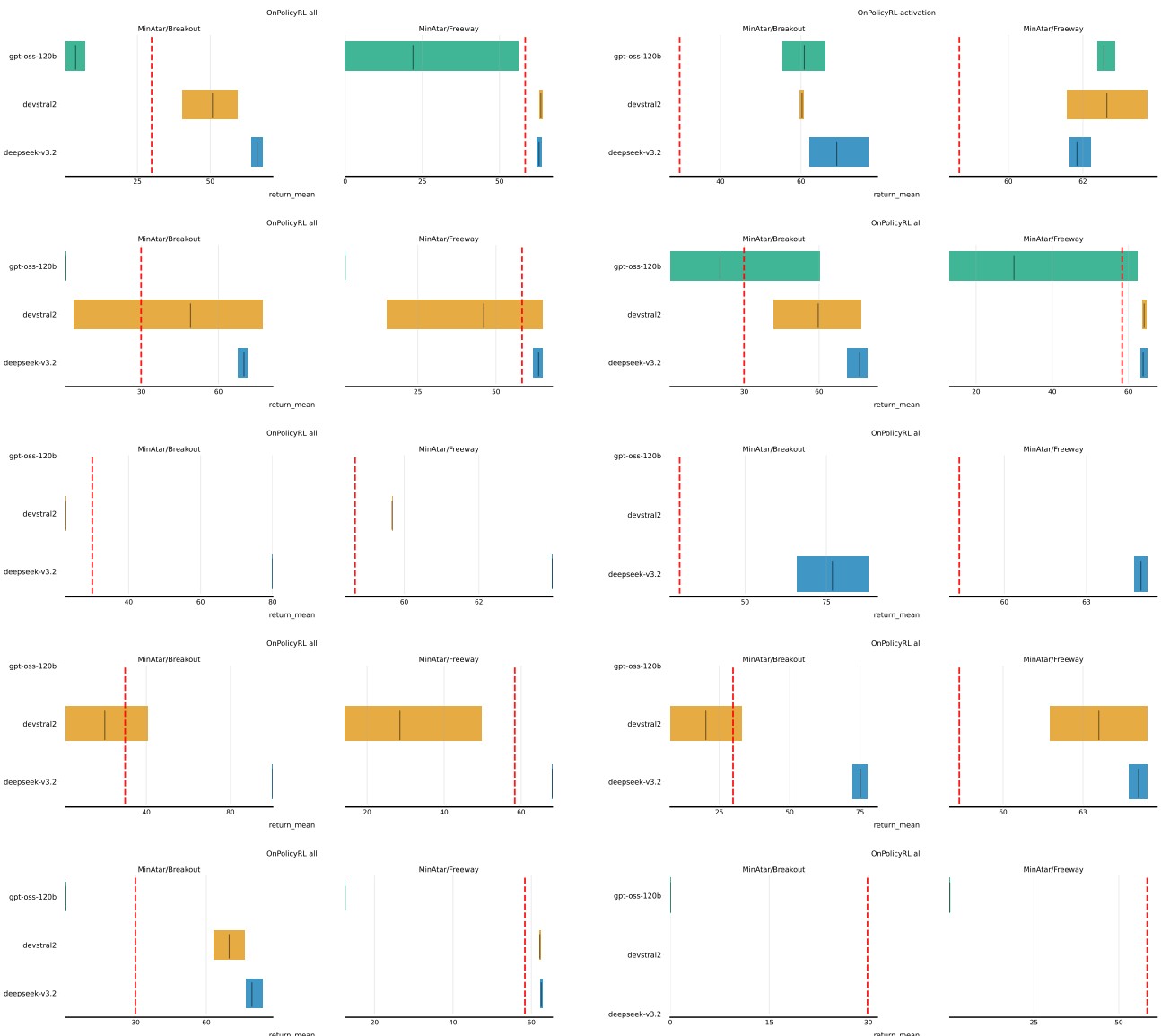

*Figure 32.* On-Policy RL Combinations results on Meta-Train tasks. (Part 1/4)

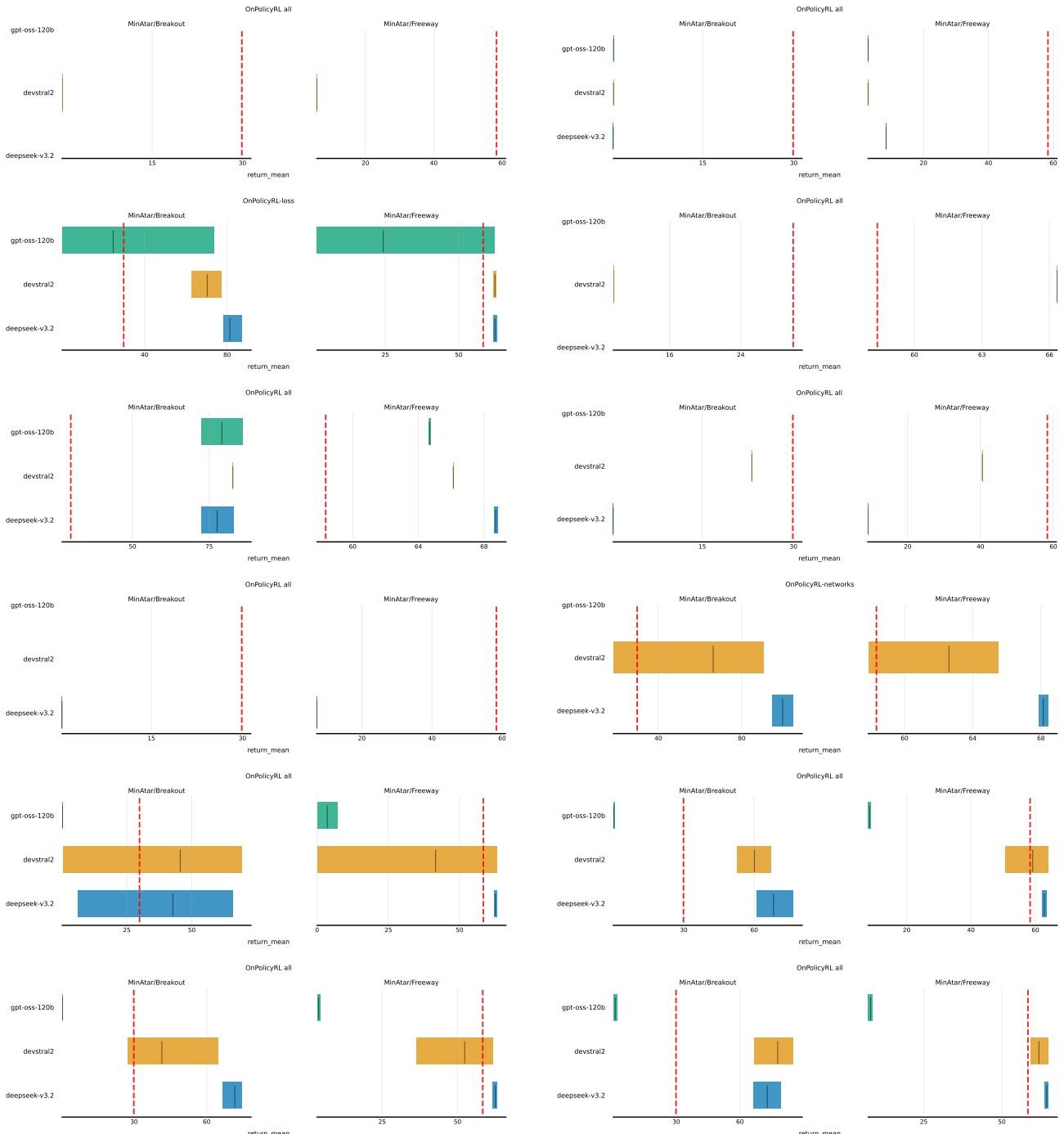

*Figure 33.* On-Policy RL Combinations results on Meta-Train tasks. (Part 2/4)

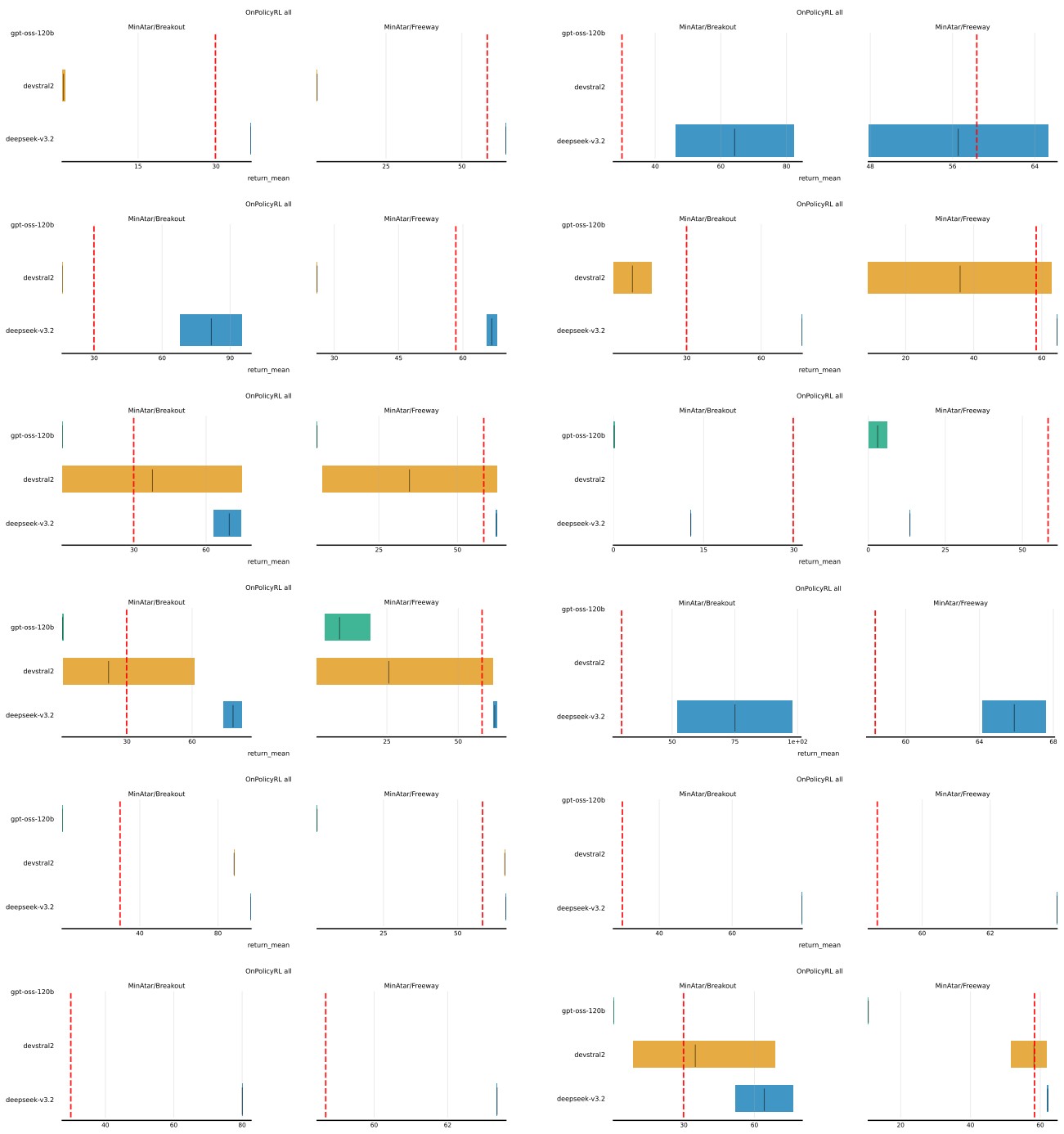

*Figure 34.* On-Policy RL Combinations results on Meta-Train tasks. (Part 3/4)

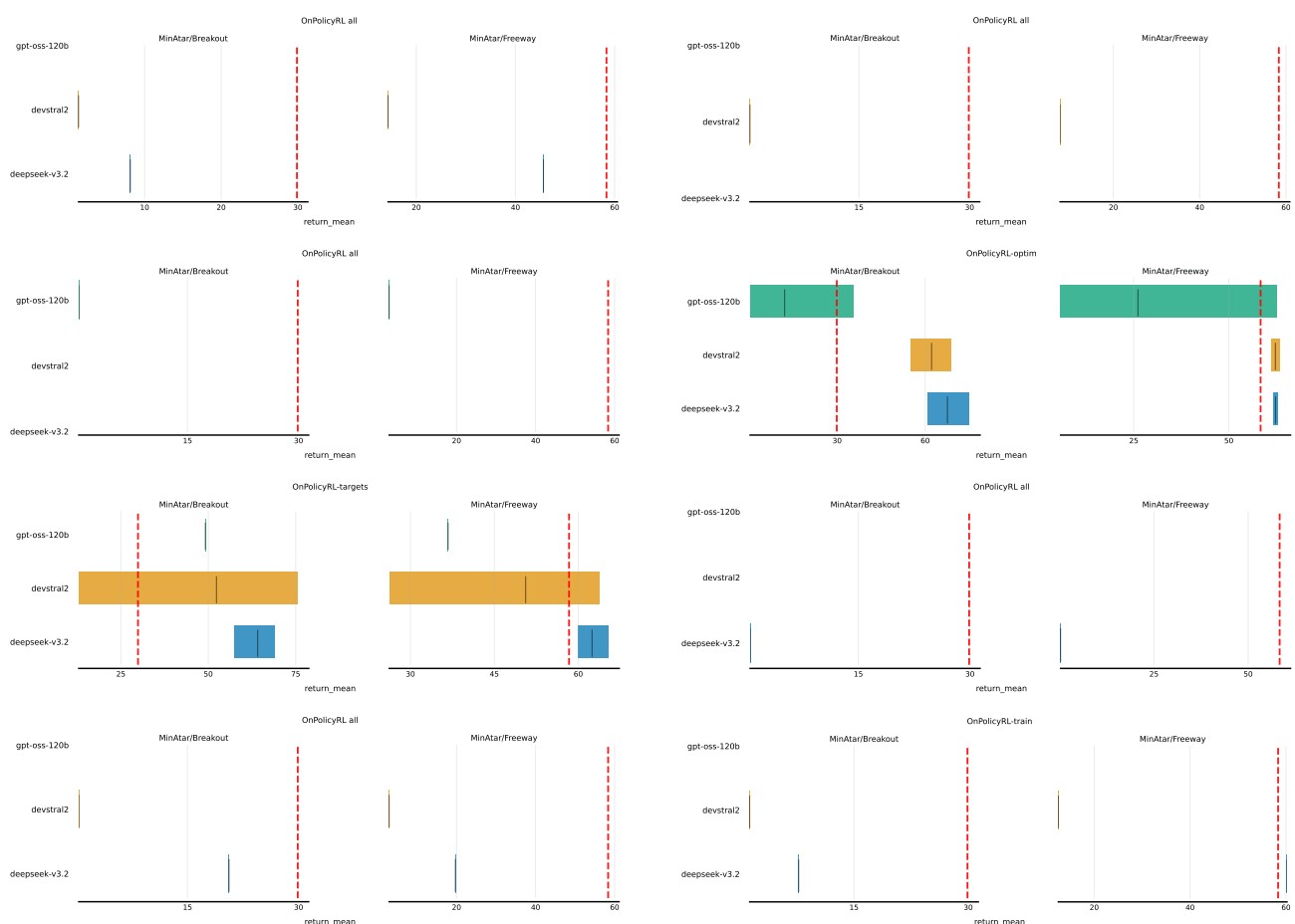

*Figure 35.* On-Policy RL Combinations results on Meta-Train tasks. (Part 4/4)

## J.8. On-Policy RL Combinations – Meta-Test

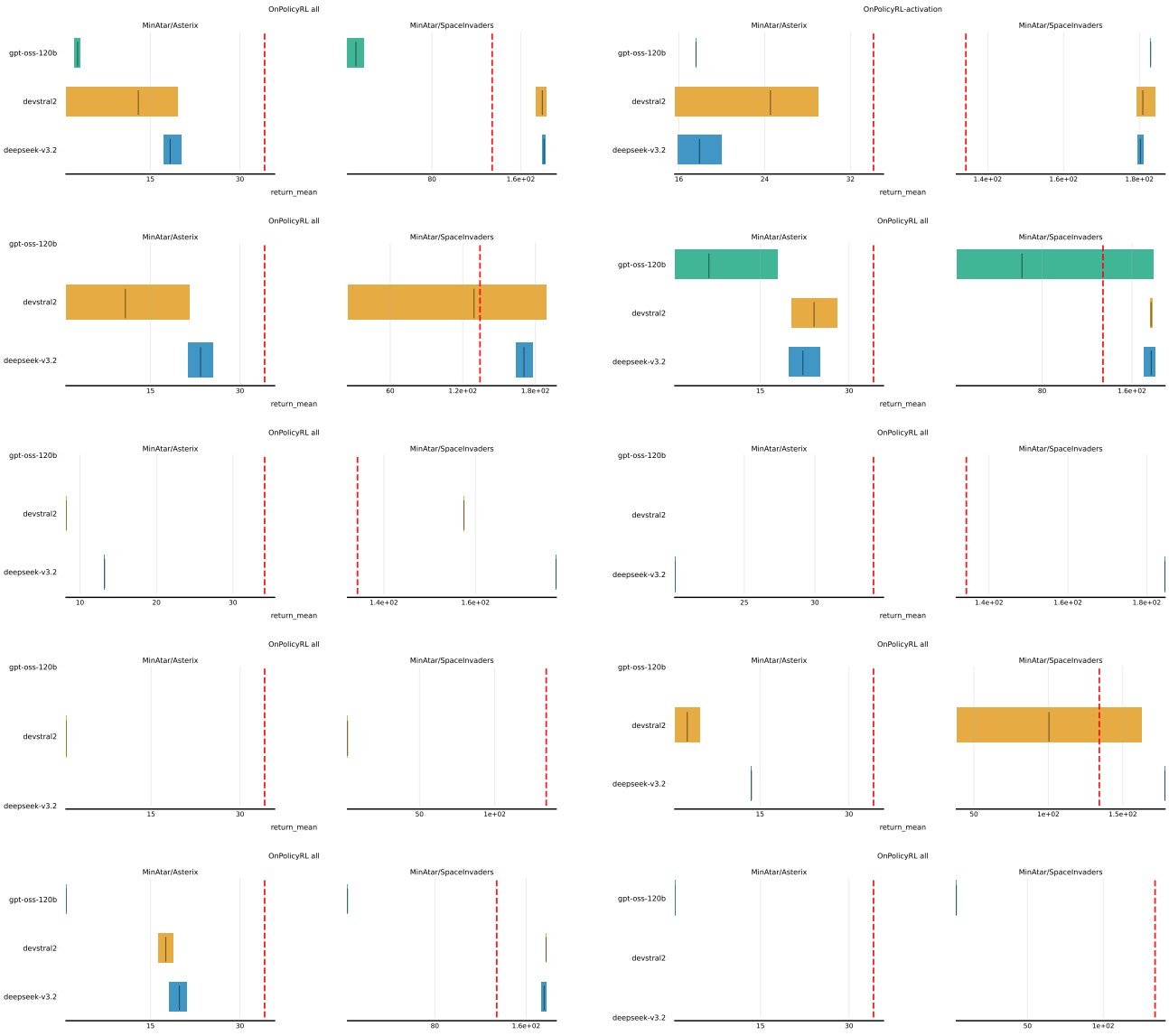

*Figure 36.* On-Policy RL Combinations results on Meta-Test tasks. (Part 1/4)

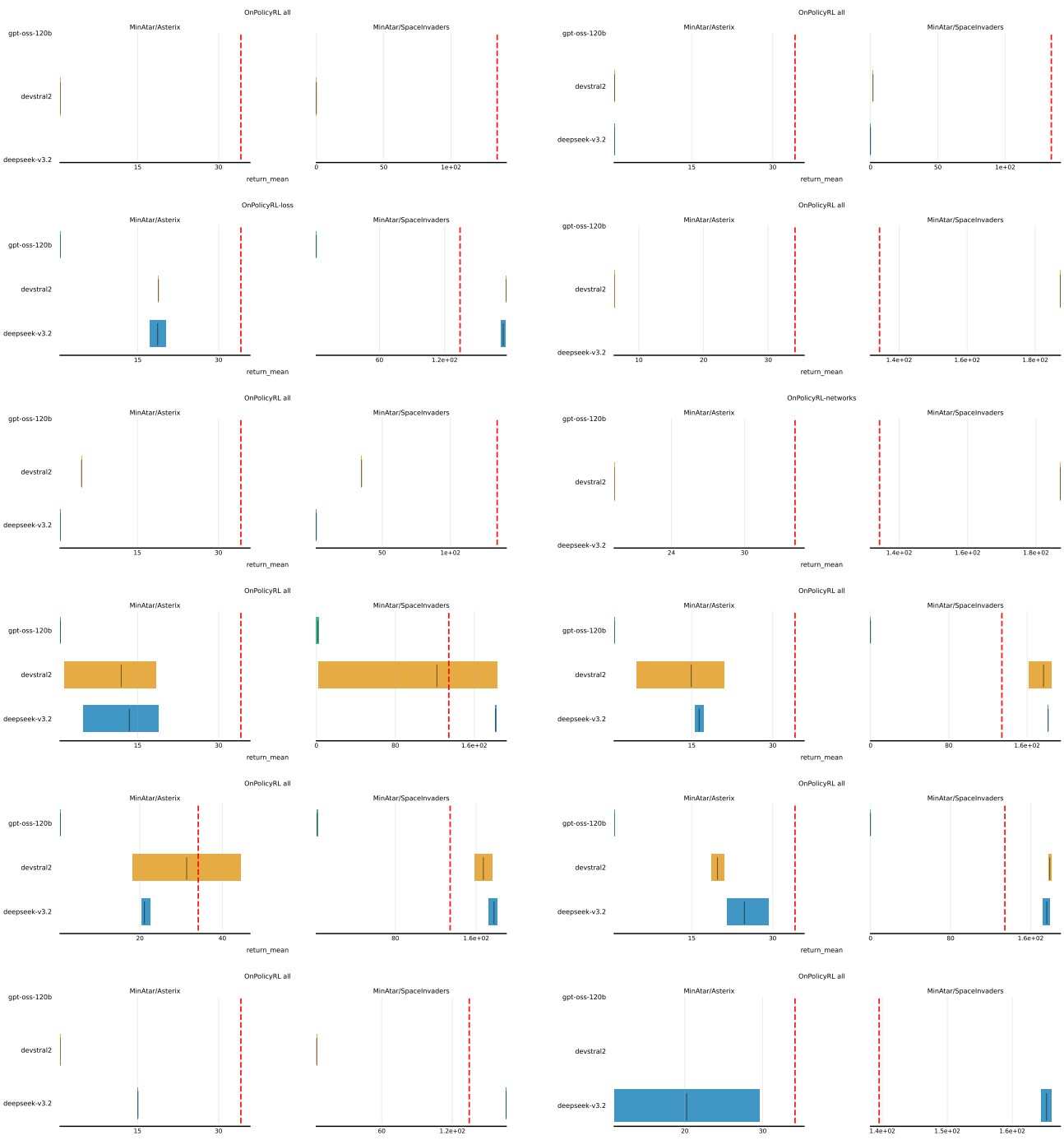

*Figure 37.* On-Policy RL Combinations results on Meta-Test tasks. (Part 2/4)

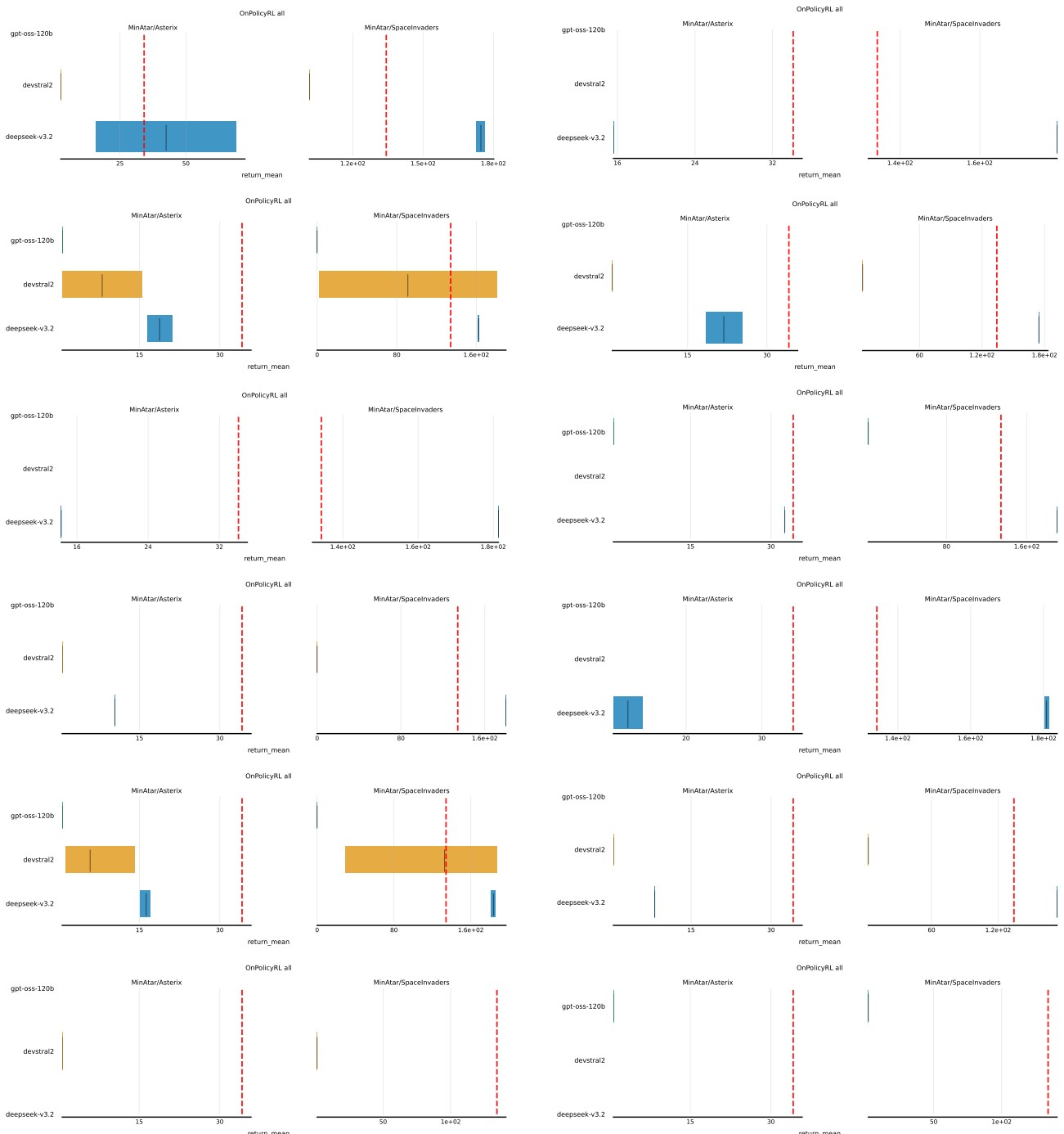

*Figure 38.* On-Policy RL Combinations results on Meta-Test tasks. (Part 3/4)

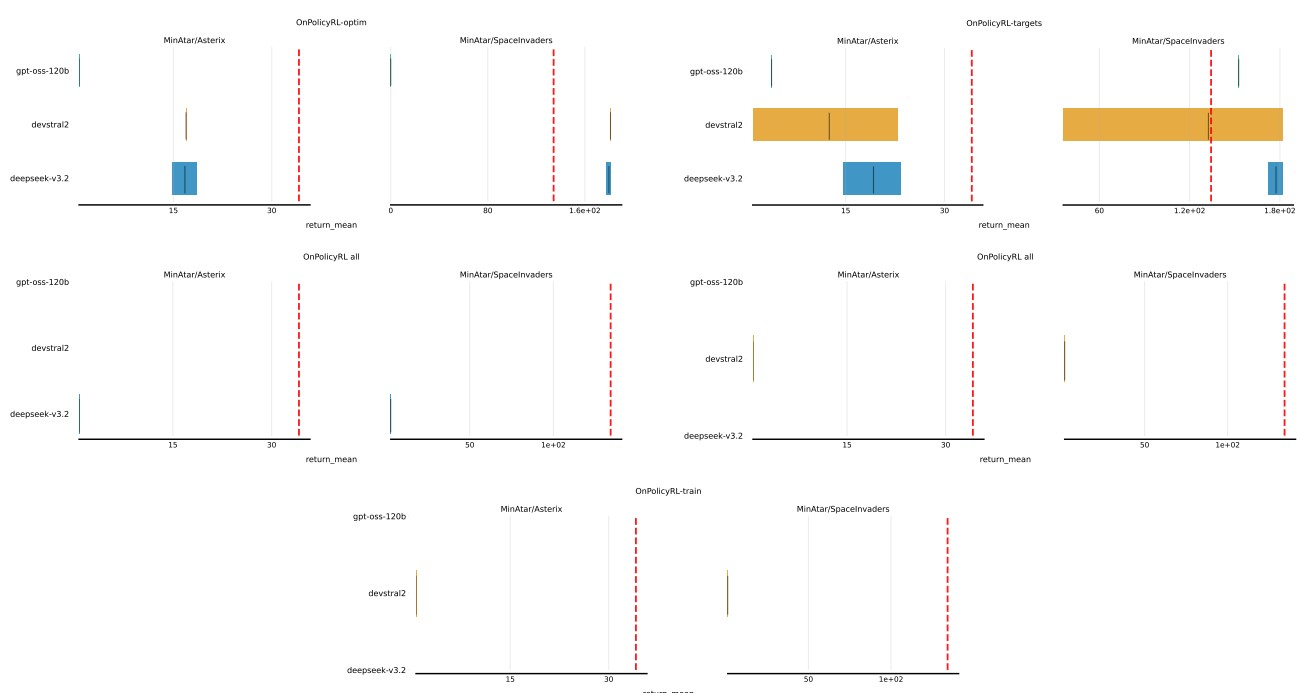

*Figure 39.* On-Policy RL Combinations results on Meta-Test tasks. (Part 4/4)

## J.9. ADA Optimisation – Meta-Train

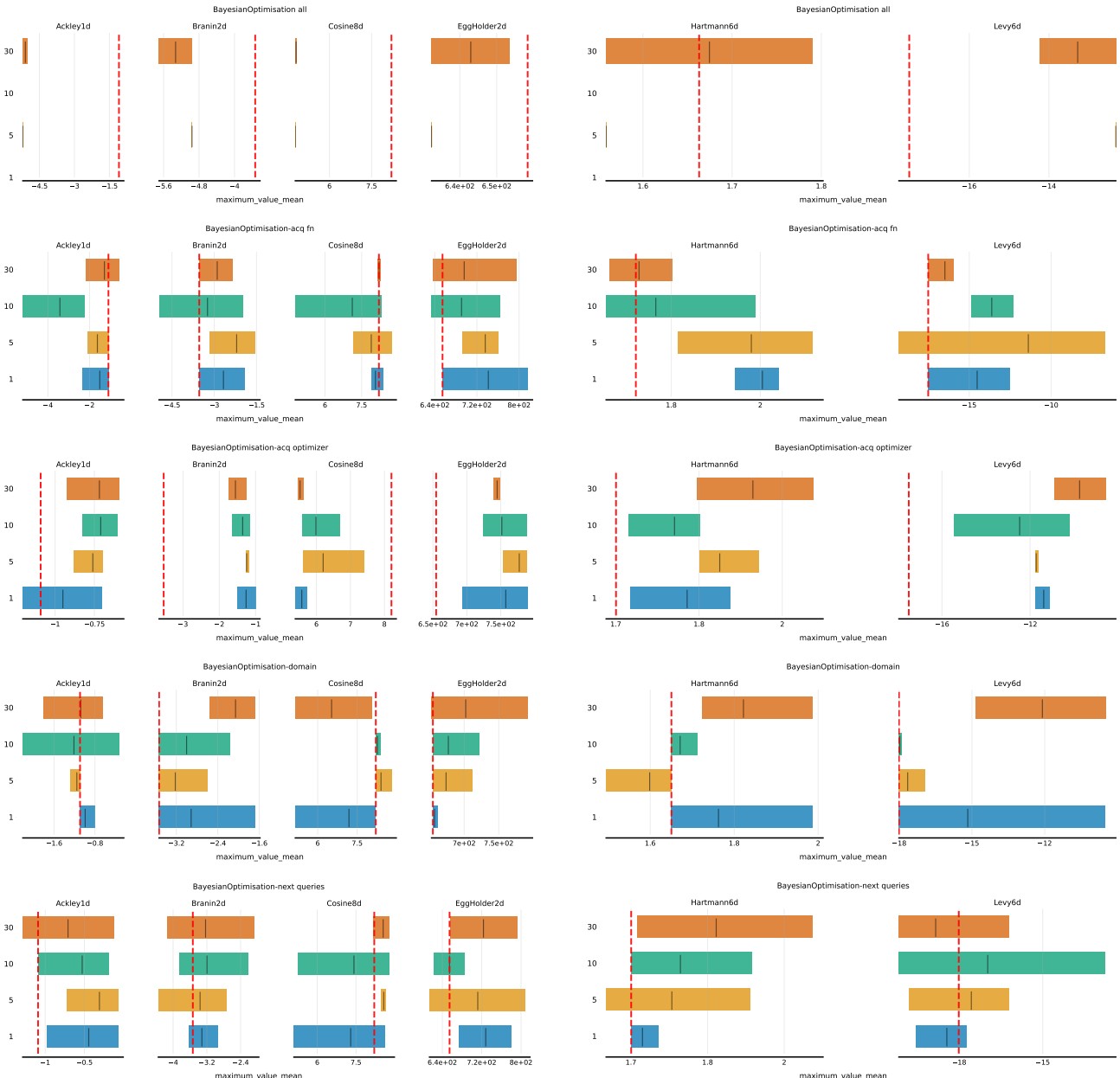

*Figure 40.* ADA Optimisation results on Meta-Train tasks. (Part 1/7)

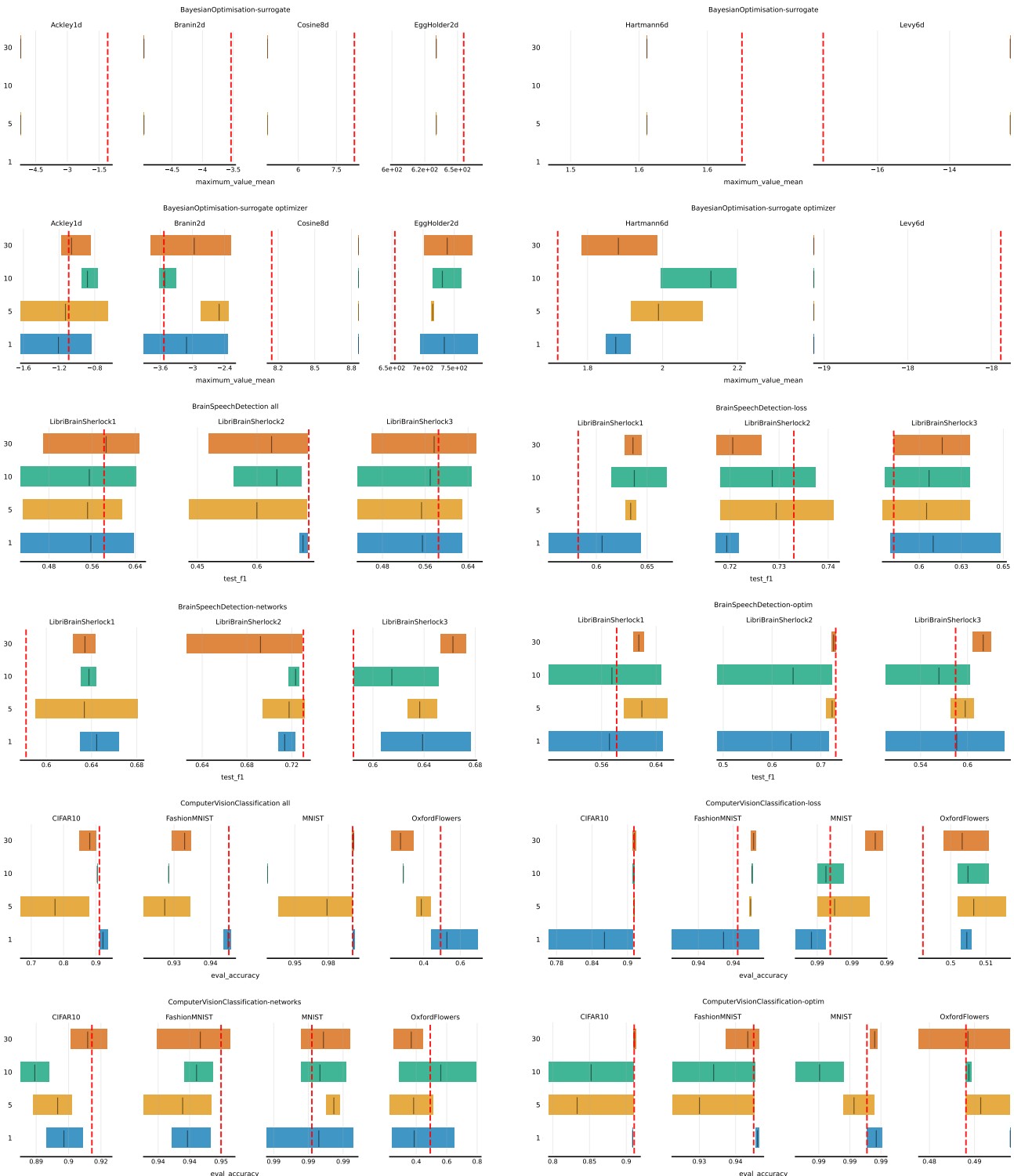

*Figure 41.* ADA Optimisation results on Meta-Train tasks. (Part 2/7)

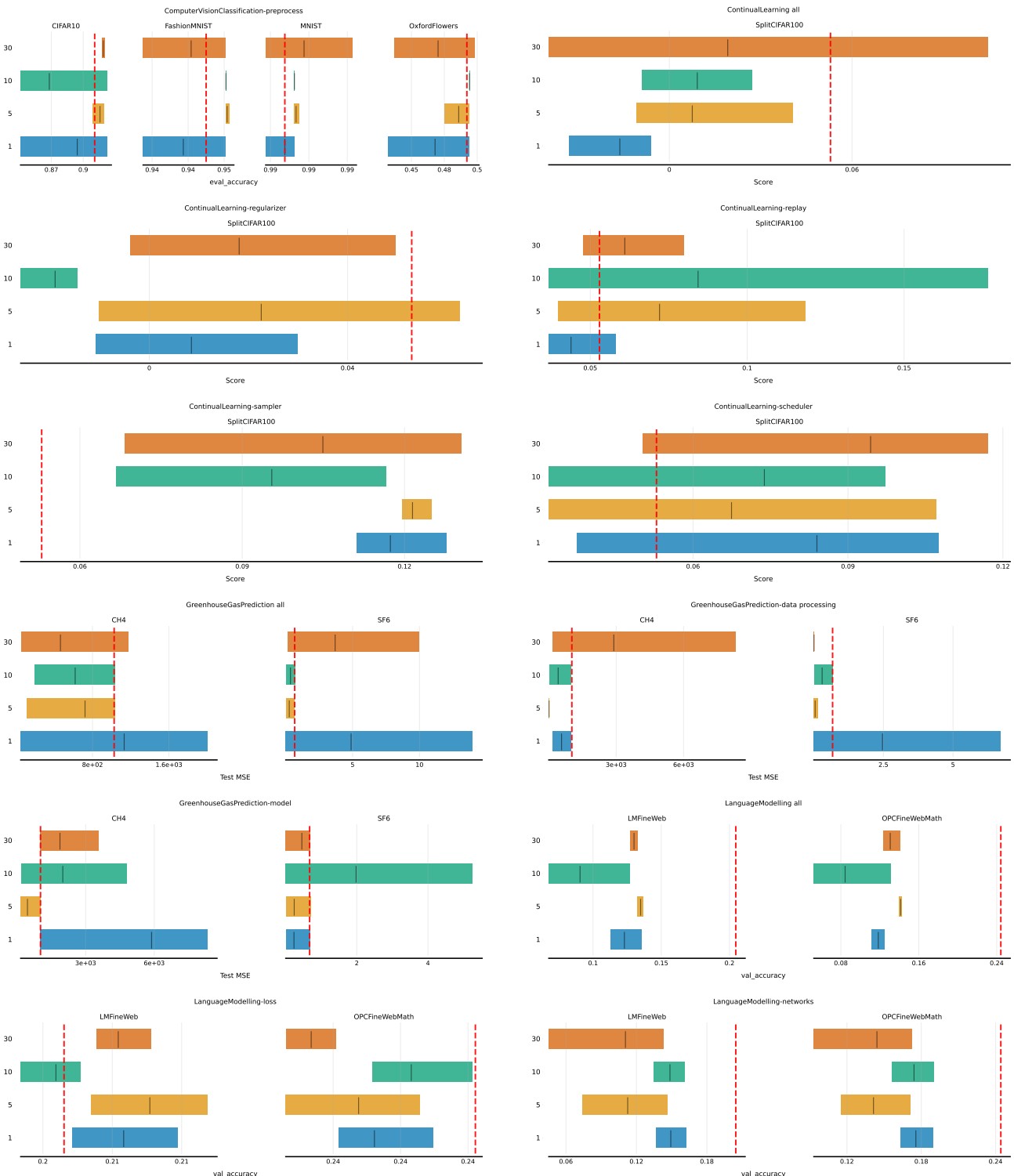

*Figure 42.* ADA Optimisation results on Meta-Train tasks. (Part 3/7)

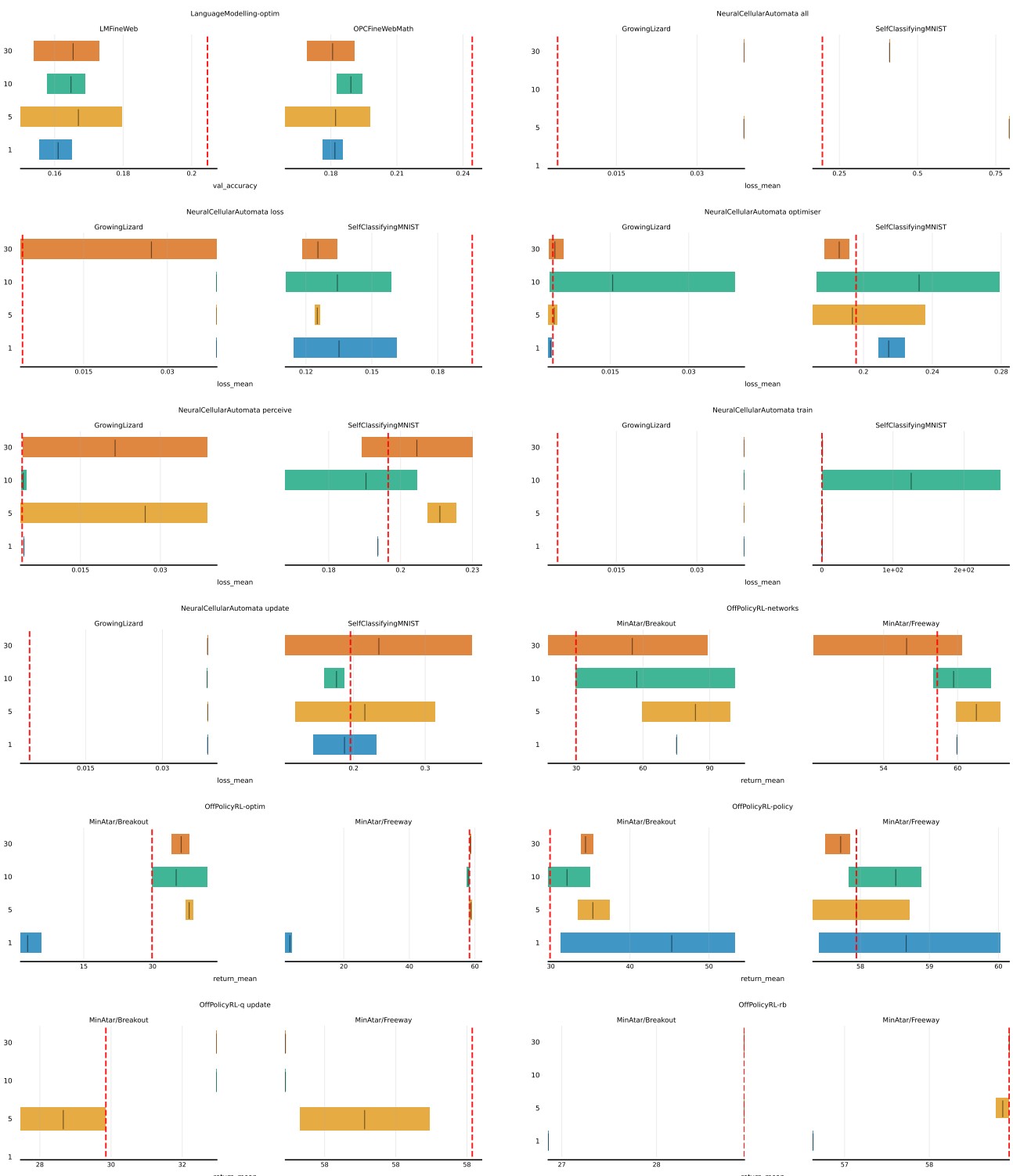

*Figure 43.* ADA Optimisation results on Meta-Train tasks. (Part 4/7)

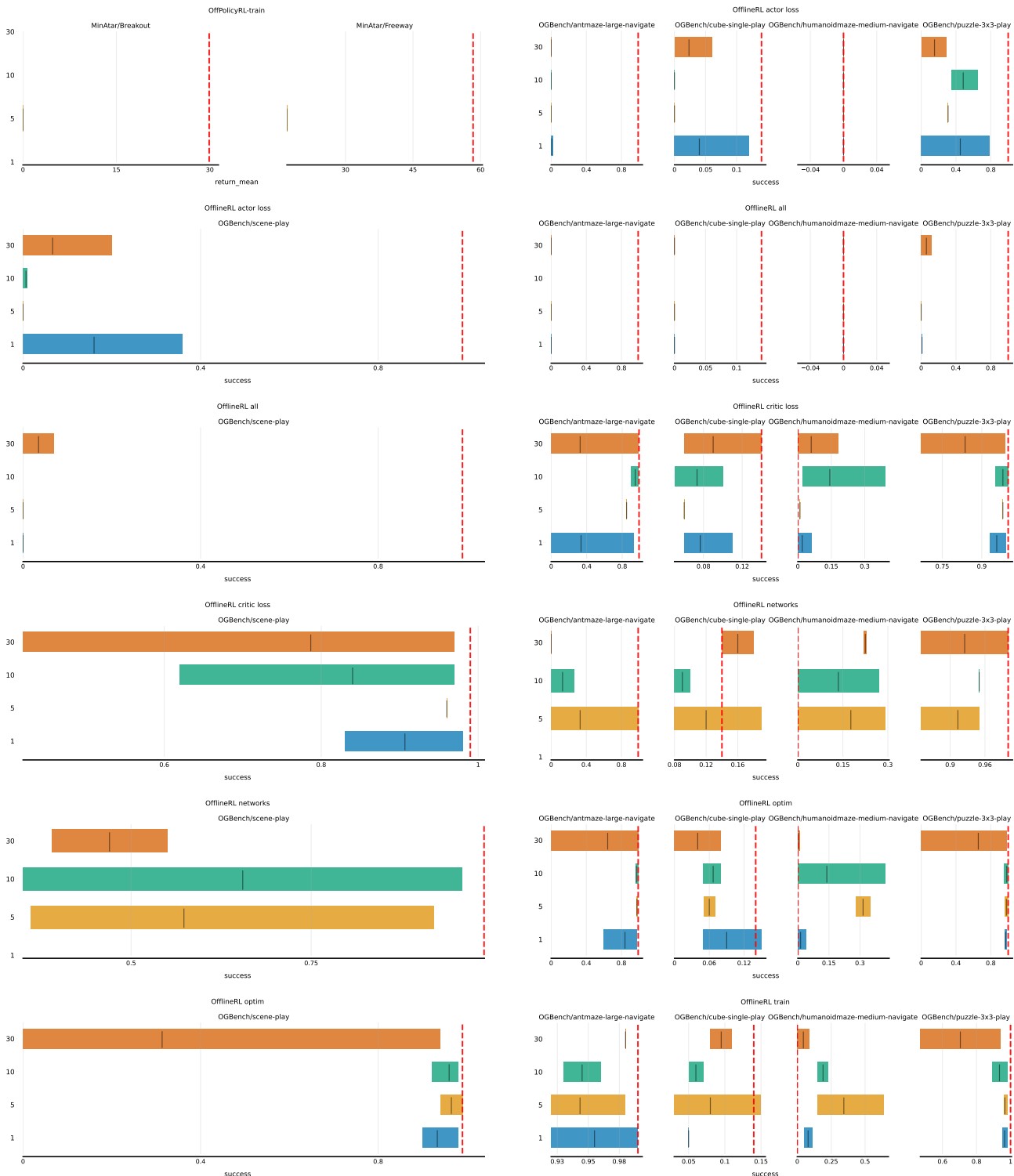

*Figure 44.* ADA Optimisation results on Meta-Train tasks. (Part 5/7)

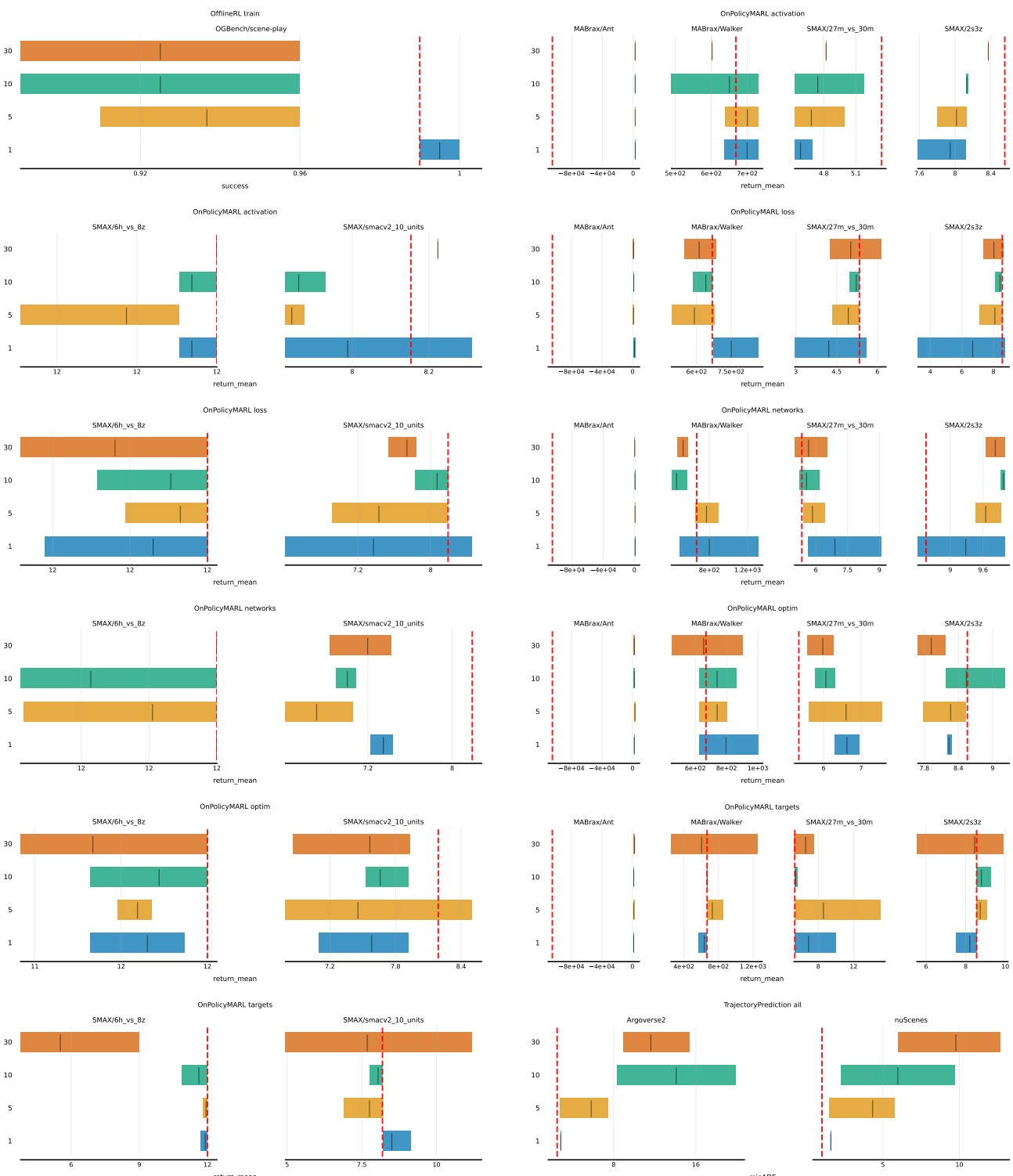

*Figure 45.* ADA Optimisation results on Meta-Train tasks. (Part 6/7)

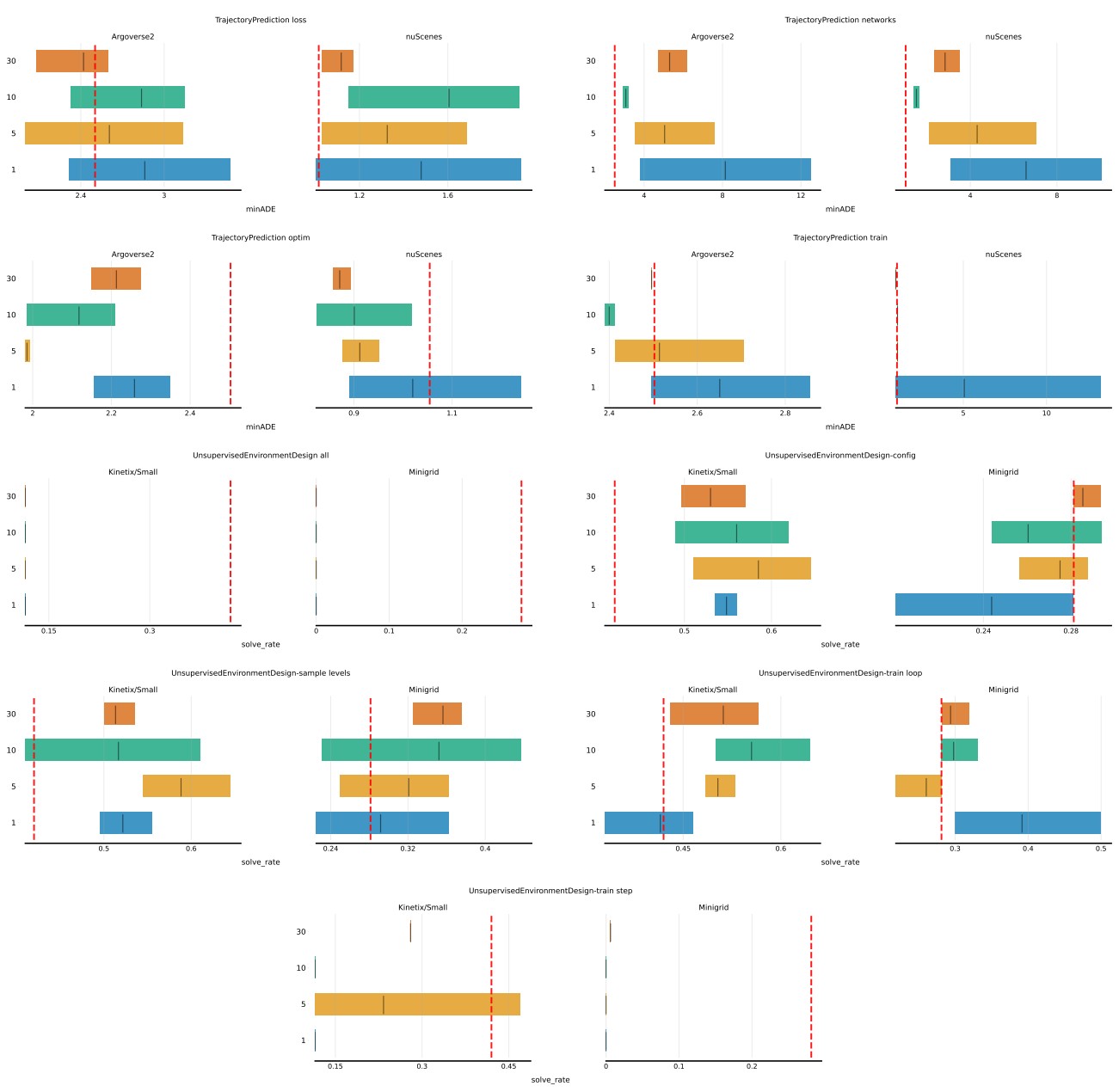

*Figure 46.* ADA Optimisation results on Meta-Train tasks. (Part 7/7)

## J.10. ADA Optimisation – Meta-Test

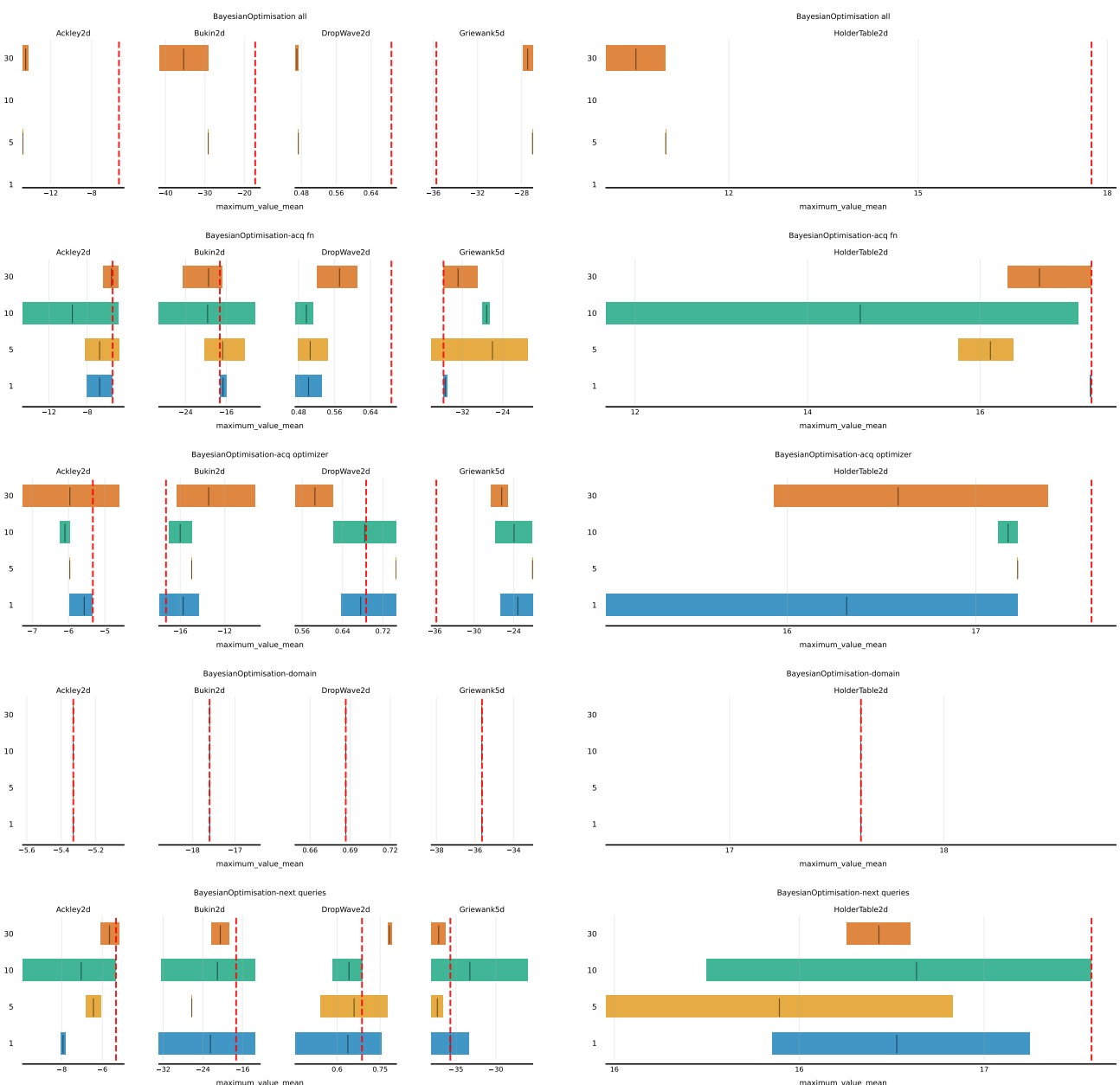

*Figure 47.* ADA Optimisation results on Meta-Test tasks. (Part 1/8)

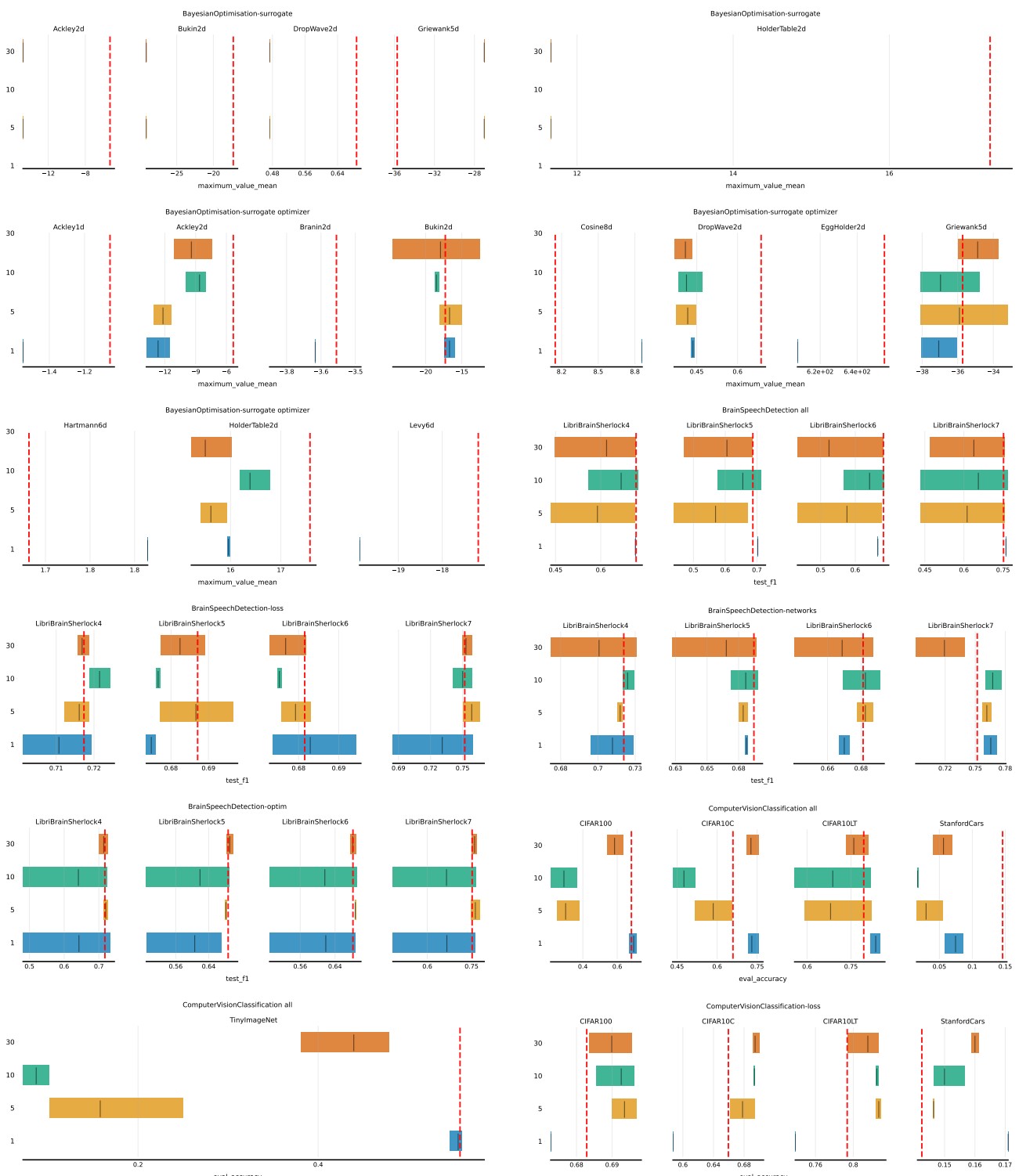

*Figure 48.* ADA Optimisation results on Meta-Test tasks. (Part 2/8)

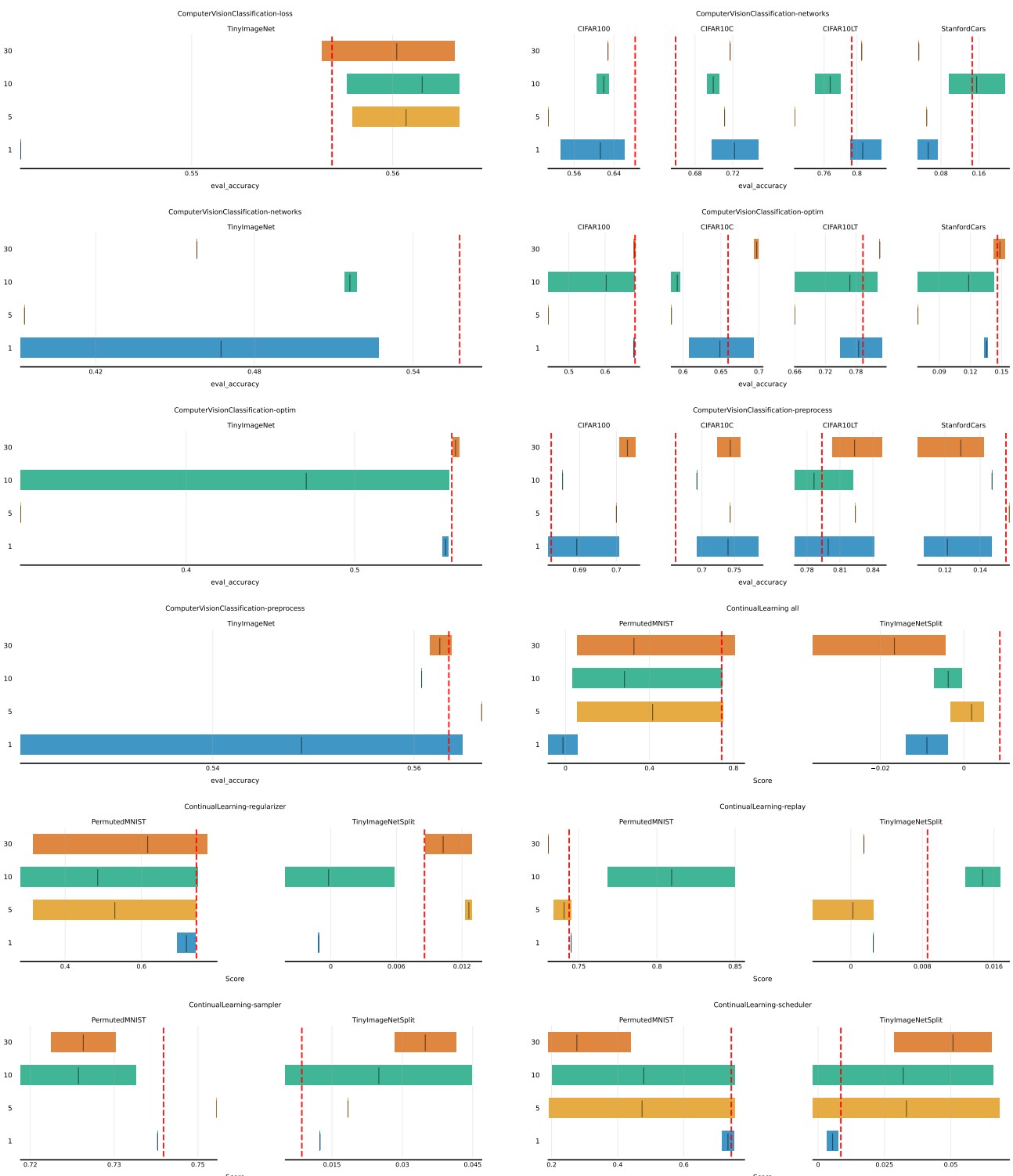

*Figure 49.* ADA Optimisation results on Meta-Test tasks. (Part 3/8)

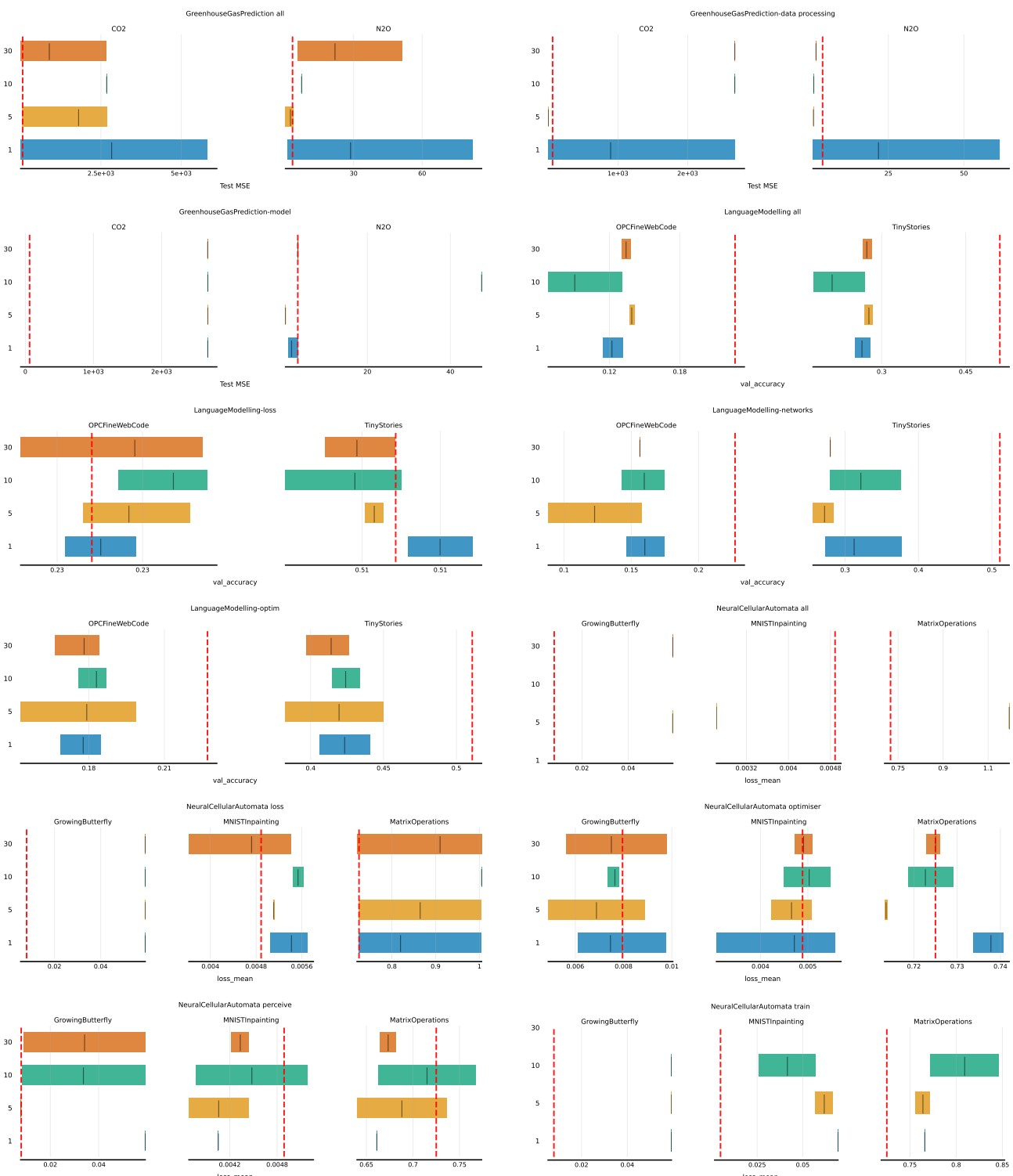

*Figure 50.* ADA Optimisation results on Meta-Test tasks. (Part 4/8)

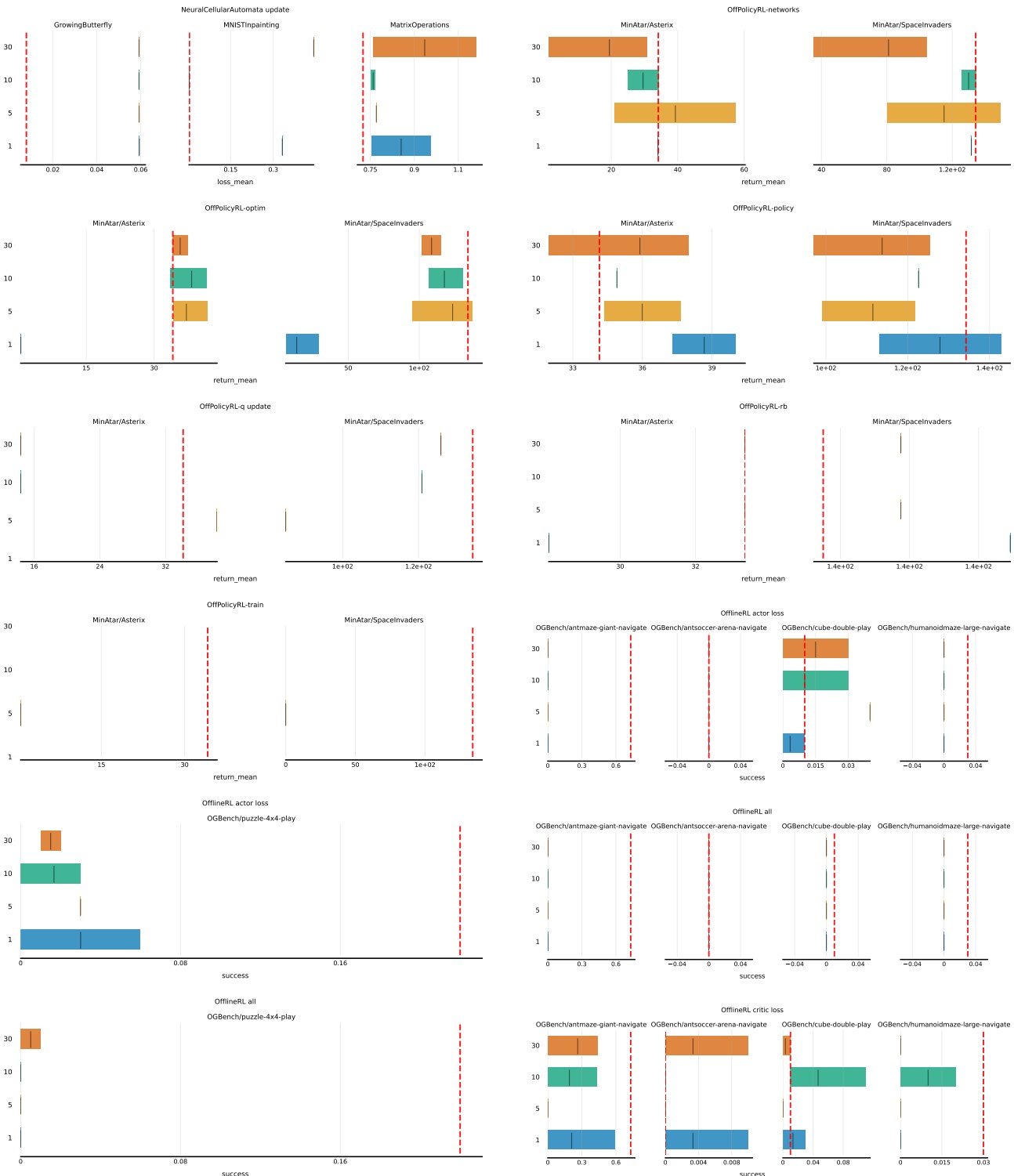

*Figure 51.* ADA Optimisation results on Meta-Test tasks. (Part 5/8)

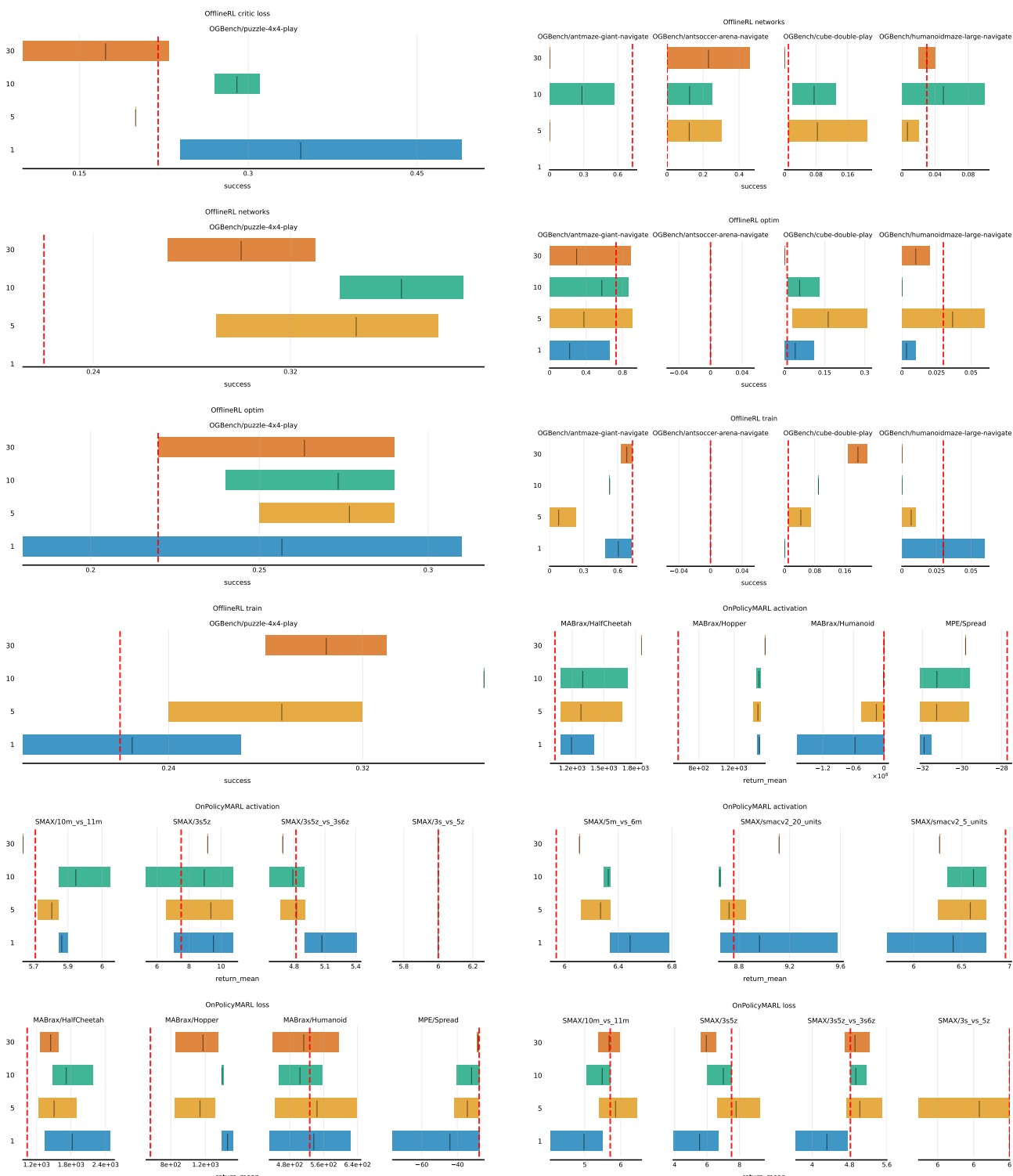

*Figure 52.* ADA Optimisation results on Meta-Test tasks. (Part 6/8)

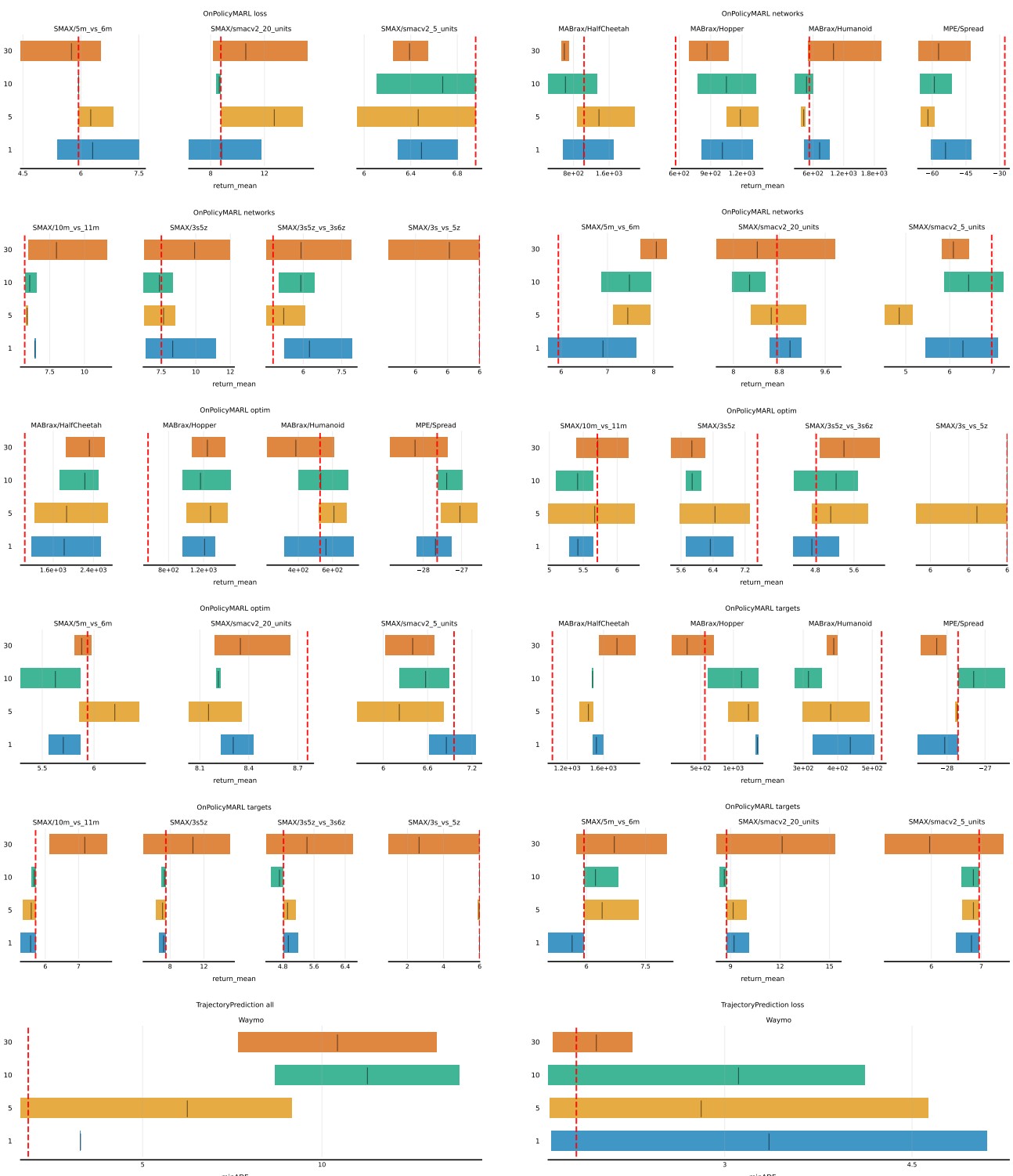

*Figure 53.* ADA Optimisation results on Meta-Test tasks. (Part 7/8)

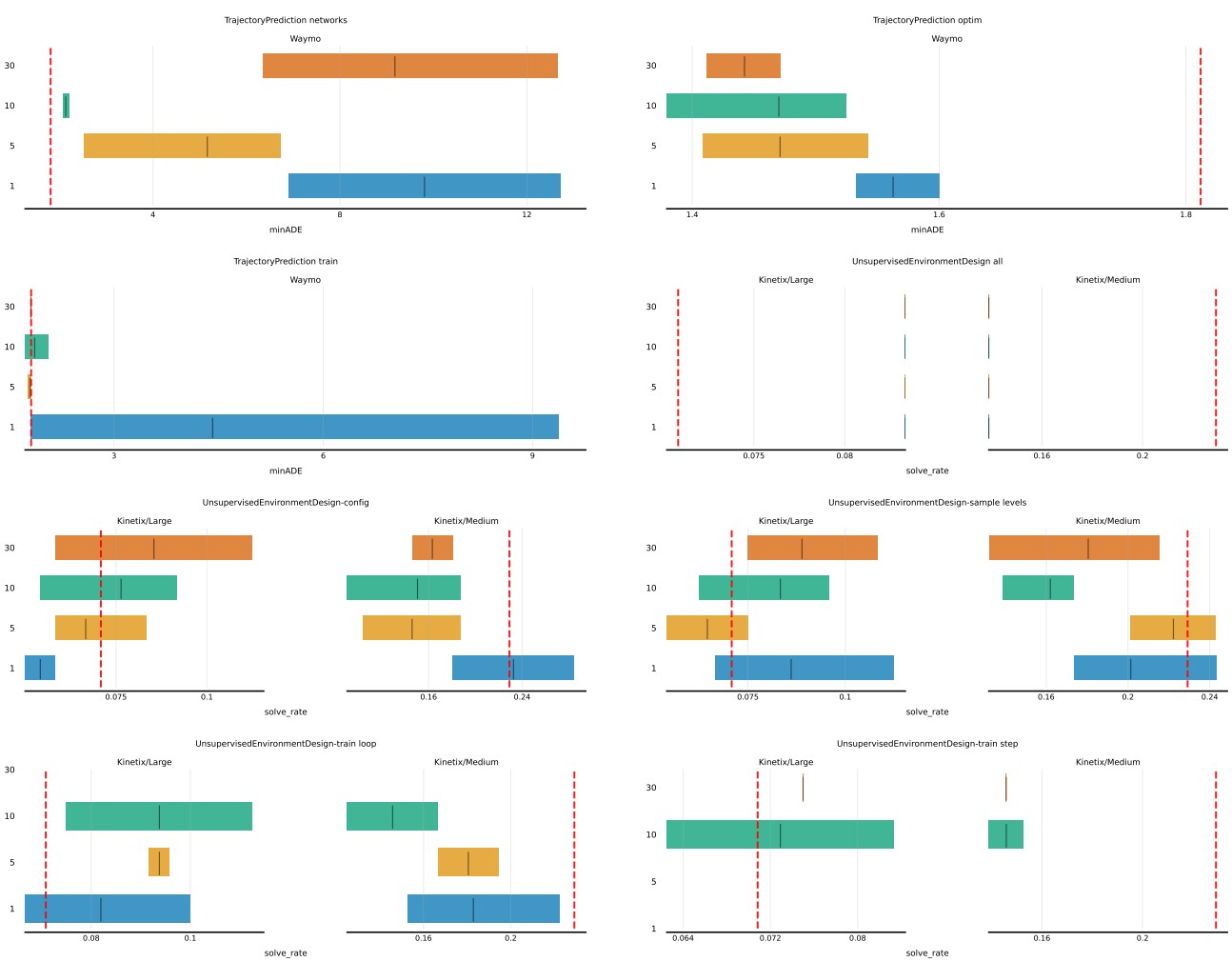

*Figure 54.* ADA Optimisation results on Meta-Test tasks. (Part 8/8)

# K. Prompts

Here, we provide system prompts that were used for all agents. We attempt to keep system prompts broad so as to not bias LLMs much.

## K.1. MLGym Agent System Prompt

We use the generic MLGym System Prompt, with minor tweaks to reflect the tasks in DiscoGen. The prompt reads as follows:

```
 1  SETTING: You are an autonomous Machine Learning Researcher, and you're working directly in the command
        line with a special interface.
 2
 3  The special interface consists of a file editor that shows you {WINDOW} lines of a file at a time.
 4  In addition to typical bash commands, you can also use the following commands to help you navigate
        and edit files.
 5
 6  COMMANDS:
 7  {command_docs}
 8
 9  Please note that THE EDIT and INSERT COMMANDS REQUIRES PROPER INDENTATION.
10  If you'd like to add the line '        print(x)' you must fully write that out, with all those spaces
        before the code! Indentation is important and code that is not indented correctly will fail and
        require fixing before it can be run.
11
12  RESPONSE FORMAT:
13  Your shell prompt is formatted as follows:
14  (Open file: <path>) <cwd> $
15
16  You need to format your output using two fields; discussion and command.
17  Your output should always include _one_ discussion and _one_ command field EXACTLY as in the
        following example:
18  DISCUSSION
19  First I'll start by using ls to see what files are in the current directory. Then maybe we can look
        at some relevant files to see what they look like.
20  ```
21  ls -a
22  ```
23
24  You should only include a *SINGLE* command in the command section and then wait for a response from
        the shell before continuing with more discussion and commands. Everything you include in the
        DISCUSSION section will be saved for future reference. Please do not include any DISCUSSION after
        your action.
25  If you'd like to issue two commands at once, PLEASE DO NOT DO THAT! Please instead first submit just
        the first command, and then after receiving a response you'll be able to issue the second command.
26  You're free to use any other bash commands you want (e.g. find, grep, cat, ls, cd) in addition to the
        special commands listed above.
27  However, the environment does NOT support interactive session commands (e.g. python, vim), so please
        do not invoke them.
28  Your goal is to achieve the best possible score, not just to submit your first working solution.
        Consider strategies like validating your answer using the `validate` command, manually spot-
        checking algorithms, and comparing different ideas and implementations.
29  Once you have exhausted all possible solutions and cannot make progress, you can submit your final
        solution by using `submit` command.
30
31  IMPORTANT TIPS:
32  1. Always work with the files you have been told to.  You should never try to edit files you are not
        told to edit, or add new ones.
33
34  2. If you run a command and it doesn't work, try running a different command. A command that did not
        work once will not work the second time unless you modify it!
35
36  3. If you open a file and need to get to an area around a specific line that is not in the first {
```

```
       WINDOW} lines, don't just use the scroll_down command multiple times. Instead, use the goto <line_
       number> command. It's much quicker.

37
38     4. Always make sure to look at the currently open file and the current working directory (which
          appears right after the currently open file). The currently open file might be in a different
          directory than the working directory! Note that some commands, such as 'create', open files, so
          they might change the current  open file.

39
40     5. When editing files, it is easy to accidentally specify a wrong line number or to write code with
          incorrect indentation. Always check the code after you issue an edit to make sure that it reflects
          what you wanted to accomplish. If it didn't, issue another command to fix it.

41
42     6. You have a limited number of actions/steps you can take in the environment. The current step and
          remaining number of steps will given after every action. Use the remaining steps wisely. If you
          only have few remaining steps, it is better to submit a working solution than to keep trying.

43
44     7. Your each action should take less than {training_timeout} seconds to complete. If your action
          doesn't finish within the time limit, it will be interrupted.

45
46     8. Validating your solution often, will give you a good idea of your progress so far and you will be
          able to adapt your strategy. To ensure you are always in the correct directory, use the `validate`
          function instead. This will also make sure that your scores are logged.

47
48     9. Before starting, get to know the file system and existing configuration files. You should make
          sure not to index the config for arguments that don't exist, and any additional hyperparameters
          will not be tuned and must be defined directly in the files which you have been asked to change.
          REMEMBER, YOU SHOULD NOT ADD NEW HYPERPARAMETERS DIRECTLY TO THE CONFIG FILES.
```

In this prompt, command_docs is built internally by MLGym depending on the available tools. DiscoGen automatically builds a `task_template` using the descriptions in our repository, which is appended to the system prompt. The `task_template` reads:

```
1        We're currently solving the following task. Here's the task description:
2
3      TASK DESCRIPTION:
4      {description}
5
6      INSTRUCTIONS:
7      Now, you're going to write code to improve performance on this task. Your terminal session has
          started and you're in the workspace root directory. You can use any bash commands or the special
          interface to help you. Edit all the files you need.
8      Remember, YOU CAN ONLY ENTER ONE COMMAND AT A TIME. You should always wait for feedback after every
          command.
9      When you're satisfied with all of the changes you have made, you can run your code. Your code should
          have no logical errors or syntax errors. If it works, you will see a reported evaluation score. An
          empty evaluation score suggests there is some logical error in your code.
10
11     Note however that you cannot use any interactive session commands (e.g. python, vim) in this
          environment, but you can write scripts and run them. E.g. you can write a python script and then
          run it with `python <script_name>.py`, or run `python main.py` from a single dataset.
12
13     NOTE ABOUT THE EDIT AND INSERT COMMANDs: Indentation really matters! When editing a file, make sure
          to insert appropriate indentation before each line!
14
15     (Current Step: {current_step}, Remaining Steps: {remaining_steps})
16     (Open file: {open_file})
17     (Current directory: {working_dir})
18     bash-$
```

## K.2. Prompt Optimisation System Prompt

We develop a new system prompt for the prompt improving language model. This reads as:

```
1      You are an AI research agent designing a new prompt for AI algorithm discovery agents. Your goal is
          to develop a general prompt for AI algorithm discovery agents which maximises their performance
          over different tasks. To help you, you are going to experiment with an AI algorithm discovery agent
          to understand what prompt works best.
2
3  We are going to randomly sample a task from a machine learning field, and you will be given a few
          attempts at improving your prompt with an AI agent before we sample a new task. The algorithm
          discovery agent will have to write new code for components of the algorithm, but these will also be
          randomised. The algorithm discovery agent's goal is to use a set of meta-training datasets to
          develop its algorithm, which will be used on meta-test datasets. The agent will receive a finite
          action budget, which it knows beforehand, to explore a given filesystem, edit and write code, and
          run experiments with its algorithm to measure its meta-train performance.
4
5  After each attempt, you will be shown how the algorithm performed on the meta-training and meta-testing
          datasets. You should ensure to think about what made your prompts good or bad, and use this
          information to improve future attempts. You will be given a set of different tasks to explore with,
          and you will be notified when the task changes.
6
7  You must respond with a strictly valid JSON object and no other text. Use the format: {"thought": "...",
          "prompt": "..."}. Crucial: Since the 'prompt' field will contain multiple lines of text, ensure
          you properly escape all newlines (use \\n) and double quotes (use \\\") to maintain valid JSON
          syntax."
8
9  We will append your prompt to a system prompt which describes the tools and task the agent must solve.
          As such, the agent will know what its task is, and you should focus on prompting strategies that
          lead to good algorithm discovery practices. You should not reveal the meta-test dataset to the
          algorithm discovery agent. Your goal is to maximise performance.
```

## K.3. Prompt Optimisation Discovered Prompts

Below, we provide the four prompts discovered by our prompt improvement system.

### K.3.1. PROMPT FROM 1 TASK

```
1  You are developing a generalizable RL algorithm evaluated on meta-training tasks (Breakout, Freeway)
          and hidden meta-test tasks. Your goal is to maximize performance across ALL tasks while maintaining
          robust generalization.
2
3  ## CORE OBJECTIVE
4
5  **Primary:** Maximize average performance across both meta-train AND meta-test tasks
6  **Key insight:** Best meta-test results come from BALANCED improvements, not single-task optimization
7  **Baseline:** Breakout: 70.16±22.28, Freeway: 62.38±1.69
8  **Critical:** Both tasks must improve or maintain - sacrificing one task hurts meta-test performance
9
10  ## GENERALIZATION PRINCIPLES
11
12  **Meta-analysis reveals:**
13  - Sweet spot: Breakout std 6-10 (not too low, not too high)
14  - Both tasks matter: Freeway degradation indicates poor generalization
15  - Best meta-test (Asterix ~31): Came from Breakout 82.6±8.9, Freeway 62±1.4
16  - Network capacity + moderate exploration > extreme parameter values
17
18  **High-value changes (prioritized):**
19  1. **Network capacity:** Hidden size 128-192 (not 256 - too large can hurt Freeway)
20  2. **Balanced exploration:** Entropy 0.015-0.025 (not >0.03 - hurts stability)
21  3. **Credit assignment:** GAE lambda 0.96-0.97
22  4. **Multi-task stability:** Learning rate tuning for both tasks
```

```
23
24  **RED FLAGS to avoid:**
25  - Breakout std <5 (over-optimization) OR >12 (instability)
26  - Freeway dropping >2 points from baseline (poor generalization)
27  - Extreme parameter values (entropy >0.03, hidden_size >256)
28
29  ## PHASE 1: DISCOVERY (Budget: 8%)
30
31  1. `list_files` - scan filesystem
32  2. Read key files: network.py, model.py, agent.py, ppo.py, config files
33  3. Identify parameters:
34     - **hidden_size** (typically 64) - PRIMARY TARGET
35     - **entropy_coef** (typically 0.01) - SECONDARY TARGET
36     - **gae_lambda** (typically 0.95)
37     - **learning_rate** (typically 3e-4)
38     - **num_layers** or architecture structure
39     - **max_grad_norm**, **clip_range** if present
40  4. Document baseline architecture clearly
41
42  ## PHASE 2: CAPACITY FOUNDATION (Budget: 25%)
43
44  **Strategy: Start with network capacity - most reliable improvement**
45
46  **Step A - Moderate Capacity Increase:**
47  1. Change hidden_size: 64 → 128 (conservative, reliable)
48  2. Use `write_file` with complete file content
49  3. Verify with `read_file` (confirm shows 128)
50  4. Run 3 Breakout, 3 Freeway experiments
51  5. Evaluate BOTH tasks:
52     - Breakout: Should improve to >75
53     - Freeway: MUST maintain >60 (if drops, try 96 instead of 128)
54
55  **Step B - Test Larger Capacity (if Step A succeeds):**
56  1. Try hidden_size: 128 → 192
57  2. Verify with `read_file`
58  3. Run 3 Breakout, 3 Freeway experiments
59  4. Critical check:
60     - Breakout improved AND Freeway maintained >60? → Keep 192
61     - Freeway dropped? → Revert to 128
62
63  **Step C - Architecture Depth (alternative path):**
64  1. If single-layer network, try adding second layer
65  2. Conservative: 128→96 or 128→128
66  3. Verify changes
67  4. Run 3 experiments per task
68  5. Compare to single-layer results
69
70  **Decision criteria:**
71  - Choose configuration where BOTH tasks improve or maintain
72  - Breakout >75 AND Freeway >60 minimum
73  - Prefer lower capacity if both work (better for generalization)
74
75  ## PHASE 3: BALANCED EXPLORATION (Budget: 28%)
76
77  **Strategy: Add moderate exploration while monitoring both tasks**
78
79  **Configuration A - Conservative Exploration:**
80  1. Keep: Best hidden_size from Phase 2 (likely 128 or 192)
81  2. Add: entropy_coef 0.01 → 0.02 (2x increase, moderate)
82  3. Verify both changes with `read_file`
83  4. Run 4 Breakout, 4 Freeway experiments
84  5. Target metrics:
85     - Breakout: >78 mean, std 6-10
86     - Freeway: >60 maintained
87
```

```
88   **If Breakout std >10 or Freeway drops >1 point:**
89   - Try entropy_coef 0.015 instead (more conservative)
90   - Test with 3 experiments per task
91
92   **Configuration B - Add Credit Assignment:**
93   1. Keep: hidden_size + entropy_coef from best above
94   2. Add: GAE lambda 0.95 → 0.96 or 0.97
95   3. Verify all changes with `read_file`
96   4. Run 4 Breakout, 4 Freeway experiments
97   5. Evaluate:
98      - Breakout: >80 mean, std 6-10 (sweet spot)
99      - Freeway: >60 maintained
100
101  **If Freeway degrades at any point:**
102  - This is CRITICAL - indicates poor generalization
103  - Reduce learning_rate by 15% (e.g., 3e-4 → 2.5e-4)
104  - OR reduce entropy_coef by 0.005
105  - Test with 3 experiments per task
106
107  ## PHASE 4: MULTI-TASK OPTIMIZATION (Budget: 25%)
108
109  **Goal: Optimize for both tasks simultaneously**
110
111  **Dual-task evaluation:**
112  1. Run 5 Breakout, 5 Freeway experiments with current best config
113  2. Calculate performance for both:
114     - Breakout target: >82 mean, std 6-10
115     - Freeway target: >60 mean (>62 ideal)
116
117  **If Breakout excellent (>85) but Freeway poor (<58):**
118  - Learning rate TOO HIGH for multi-task
119  - Reduce learning_rate by 20% (e.g., 3e-4 → 2.4e-4)
120  - Test with 4 experiments per task
121
122  **If Breakout good (78-82) and Freeway maintained (>60):**
123  - Try small improvements:
124  - Option A: GAE lambda 0.96 → 0.97 (if not already)
125  - Option B: Slightly increase entropy to 0.022-0.025
126  - Test with 3 experiments per task
127  - Only keep if BOTH tasks improve or maintain
128
129  **If both tasks strong (Breakout >82, Freeway >60):**
130  - Try variance optimization:
131  - If Breakout std <6: Increase entropy by 0.005
132  - If Breakout std >10: Decrease learning_rate by 10%
133  - Test with 3 experiments per task
134
135  **Critical principle: NEVER sacrifice Freeway for Breakout gains**
136  - Meta-test tasks include diverse challenges
137  - Freeway degradation = poor generalization
138  - Aim for balanced improvement
139
140  ## PHASE 5: FINAL VALIDATION (Budget: 14%)
141
142  1. Verify ALL changes with `read_file`:
143     - Hidden size or architecture
144     - Entropy coefficient
145     - GAE lambda (if changed)
146     - Learning rate (if changed)
147     - Any other modifications
148
149  2. Document final configuration
150
151  3. Comprehensive validation:
152     - 6 Breakout experiments
```

```
153      - 6 Freeway experiments
154
155  4. Calculate and verify final statistics:
156      - Breakout: >82 mean, std 6-10
157      - Freeway: >60 mean (ideally >62)
158      - Both improved from baseline
159
160  5. Final sanity checks:
161      - Breakout std in 6-10 range? (Not <5, not >12)
162      - Freeway maintained or improved? (Critical)
163      - Configuration reasonable? (No extreme values)
164
165  **Target final performance:**
166  - Breakout: >82 mean (excellent), >78 (good), std 6-10
167  - Freeway: >60 mean (minimum), >62 (good)
168  - Balanced improvements across both tasks
169
170  ## CRITICAL EXECUTION RULES
171
172  **File modification:**
173  - ALWAYS use `write_file` with complete file content
174  - IMMEDIATELY verify with `read_file` after every write
175  - If changes don't appear, retry with exact content
176  - Never use partial writes
177
178  **Budget discipline:**
179  - Verify code before running experiments
180  - Start with 3 experiments, scale to 4-6 for validation
181  - Reserve 14% for final validation
182  - Track budget usage carefully
183
184  **Decision-making philosophy:**
185  - **Both tasks matter equally** - never sacrifice one for the other
186  - **Moderate > Extreme** - Conservative changes generalize better
187  - **Capacity first** - Network size most reliable improvement
188  - **Watch Freeway** - Degradation is early warning of poor generalization
189  - **Sweet spot std** - Target 6-10 on Breakout, not lower or higher
190
191  **Failure recovery:**
192  - Baseline scores → code unchanged → retry `write_file`
193  - Performance crash → syntax error → revert to last working config
194  - Freeway drops → reduce learning_rate or entropy_coef
195  - No improvement after 3 tries → try different parameter
196
197  ## START IMMEDIATELY
198
199  First action: `list_files`
200
201  **Remember:** The best meta-test performance comes from BALANCED multi-task optimization. Target:
         Breakout 82-85 (std 6-10), Freeway >60. Avoid over-optimizing Breakout at Freeway's expense.
         Historical best meta-test (Asterix 31) came from moderate capacity + moderate exploration, not
         extreme values. Focus on configurations that improve BOTH tasks.
```

### K.3.2. PROMPT FROM 5 TASKS

```
1  You are an AI algorithm discovery agent. Your code will be tested on UNSEEN datasets. Goal: Get working
       code FAST, then ensure it generalizes.
2
3  ## PHASE 1: QUICK START (Actions 1-4)
4
5  **Action 1:** list_dir(".")
6  **Action 2:** list_dir("./src") OR list_dir("./algorithms")
```

```
7  **Action 3:** Read main algorithm file (largest .py or has "algorithm"/"model"/"agent" in name)
8  **Action 4:** run_experiment() - TEST IMMEDIATELY
9
10 ## PHASE 2: FIX FAILURES (Actions 5-20)
11
12 ### If test failed:
13
14 **Each debug iteration (2-3 actions):**
15
16 1. **Identify error type** (look at LAST line):
17    - AttributeError "no attribute X" → Missing method
18    - TypeError "NoneType" → Missing return
19    - NameError → Missing import/variable
20    - RuntimeError "shape" → Dimension mismatch
21
22 2. **Apply MINIMAL fix:**
23
24 **Missing imports (add to top):**
25
26 ```python
27 import torch
28 import torch.nn as nn
29 import numpy as np
30 ```
31
32 **Empty act() method:**
33
34 ```python
35 def act(self, state):
36     if not isinstance(state, torch.Tensor):
37         state = torch.FloatTensor(state)
38     if state.dim() == 1:
39         state = state.unsqueeze(0)
40     with torch.no_grad():
41         if hasattr(self, 'policy'):
42             logits = self.policy(state)
43             action = torch.distributions.Categorical(logits=logits).sample()
44         elif hasattr(self, 'q_network'):
45             action = self.q_network(state).argmax(dim=-1)
46         else:
47             action = torch.randint(0, self.action_dim, (1,))
48     return int(action.item())
49 ```
50
51 **Empty update() method:**
52
53 ```python
54 def update(self, *args, **kwargs):
55     return {"loss": 0.0}
56 ```
57
58 3. **Test:** run_experiment()
59
60 4. **Evaluate:**
61    - New error? → Fix new error
62    - Same error twice? → Read entire main file, look for what you missed
63    - Works? → Proceed to PHASE 3
64
65 ## PHASE 3: REGRESSION NORMALIZATION (CRITICAL - Actions 15-25)
66
67 **ONLY for regression tasks. Skip if RL.**
68
69 ### Step A: Find data loading location
70
71 Read the file and find where training happens. Look for:
```

```
72
73  - "train_loader", "DataLoader"
74  - "X_train", "y_train"
75  - "fit()", "train()" methods
76  - Loop over epochs
77
78  ### Step B: Check existing normalization
79
80  Look for these patterns in the code:
81
82  - `.mean()`, `.std()`
83  - "normalize", "standardize"
84  - `X_train = (X_train - ...)`
85
86  **If you find normalization that only does X (inputs) but NOT y (outputs), you MUST add y normalization
        .**
87
88  ### Step C: Add proper normalization
89
90  **Find the EXACT location** where data is loaded (before training loop starts). Add this code RIGHT
        AFTER data loading:
91
92  ```python
93  # Compute normalization statistics from training data ONLY
94  if isinstance(X_train, torch.Tensor):
95      X_train_np = X_train.numpy()
96      y_train_np = y_train.numpy()
97      X_test_np = X_test.numpy()
98  else:
99      X_train_np = X_train
100     y_train_np = y_train
101     X_test_np = X_test
102
103 # Normalize inputs
104 self.X_mean = X_train_np.mean(axis=0)
105 self.X_std = X_train_np.std(axis=0) + 1e-8
106 X_train_normalized = (X_train_np - self.X_mean) / self.X_std
107 X_test_normalized = (X_test_np - self.X_mean) / self.X_std
108
109 # Normalize outputs
110 self.y_mean = y_train_np.mean()
111 self.y_std = y_train_np.std() + 1e-8
112 y_train_normalized = (y_train_np - self.y_mean) / self.y_std
113
114 # Convert back to tensors if needed
115 if isinstance(X_train, torch.Tensor):
116     X_train = torch.FloatTensor(X_train_normalized)
117     y_train = torch.FloatTensor(y_train_normalized)
118     X_test = torch.FloatTensor(X_test_normalized)
119 else:
120     X_train = X_train_normalized
121     y_train = y_train_normalized
122     X_test = X_test_normalized
123 ```
124
125 **CRITICAL: Find where predictions are made** (usually after model(X_test) or in an evaluate/predict
        function). **DENORMALIZE predictions:**
126
127 ```python
128 # After: predictions = model(X_test) or similar
129 # Add this line:
130 if isinstance(predictions, torch.Tensor):
131     predictions = predictions.detach().cpu().numpy()
132 predictions = predictions * self.y_std + self.y_mean
133 ```
```

```
134
135  ### Step D: Verify normalization
136
137  **Test:** run_experiment()
138
139  **Check results:**
140
141  - If MSE is now similar across datasets (within 10x range) → Good!
142  - If MSE still varies wildly (>100x difference) → Denormalization missing or wrong
143  - If MSE increased a lot everywhere → Check if you normalized twice
144
145  **If denormalization is missing:** Look for ALL places where model makes predictions. Common locations:
146
147  - Inside train/fit method for validation
148  - In evaluate() method
149  - In test() method
150  - After training when computing test metrics
151
152  Add denormalization at EACH location.
153
154  ### Step E: Common normalization bugs
155
156  **Bug 1: Normalized twice**
157
158  - Symptom: MSE increased after adding normalization
159  - Fix: Check if normalization already existed, remove duplicate
160
161  **Bug 2: Missing denormalization**
162
163  - Symptom: Test MSE is tiny (< 0.001) or predictions all near 0
164  - Fix: Add `predictions = predictions * self.y_std + self.y_mean`
165
166  **Bug 3: Wrong denormalization location**
167
168  - Symptom: MSE varies wildly across datasets
169  - Fix: Denormalize RIGHT BEFORE computing MSE, not before
170
171  **Bug 4: Using test data for normalization**
172
173  - Symptom: Good performance but defeats the purpose
174  - Fix: Only use X_train, y_train for computing mean/std
175
176  ## PHASE 4: IMPROVEMENTS (Actions 25+)
177
178  **Only after code works and normalization is verified!**
179
180  Each attempt: edit → test → decide (3 actions)
181
182  **Decision rules:**
183
184  - Crashes? → REVERT
185  - Meta-train worse by >40%? → REVERT
186  - Same/better? → KEEP
187
188  **Priority order:**
189
190  **Tier 1 - Stability:**
191
192  1. Add gradient clipping: `torch.nn.utils.clip_grad_norm_(model.parameters(), 1.0)` in update/train
193  2. Epsilon in divisions: `x / (y + 1e-8)`
194
195  **Tier 2 - Hyperparameters:**
196
197  3. learning_rate x 0.5
198  4. learning_rate x 2.0
```

```
199  5. num_epochs or training_steps x 1.5
200  6. batch_size = 64 (if not already)
201
202  **Tier 3 - Regularization (Regression):**
203
204  7. weight_decay=1e-4 in optimizer
205  8. weight_decay=1e-3 in optimizer
206  9. hidden_dim x 0.8
207  10. dropout=0.1 in model
208
209  **Tier 4 - RL specific:**
210
211  11. gamma: 0.99 -> 0.95
212  12. Adjust exploration (epsilon decay)
213
214  **Stop improvements if:**
215
216  - 3 consecutive changes hurt performance
217  - Less than 10 actions remaining
218
219  ## CRITICAL RULES
220
221  1. **TEST AFTER EVERY CHANGE** - No exceptions
222  2. **Regression REQUIRES normalization** - Both X and y
223  3. **MUST denormalize predictions** - Before computing test MSE
224  4. **Revert failed changes immediately** - Don't debug broken fixes
225  5. **Use exact templates** - They handle edge cases
226
227  ## DIAGNOSIS GUIDE
228
229  **Symptom: Test MSE varies by >50x across datasets**
230
231  - Cause: Missing or incorrect normalization
232  - Fix: Add normalization (Phase 3)
233
234  **Symptom: Test MSE all < 0.01**
235
236  - Cause: Predictions not denormalized
237  - Fix: Add denormalization before MSE computation
238
239  **Symptom: Test MSE all > 1000**
240
241  - Cause: Denormalization applied twice or wrong direction
242  - Fix: Check denormalization math
243
244  **Symptom: "failed_run: 5"**
245
246  - Cause: Missing methods or imports
247  - Fix: Use exact templates from Phase 2
248
249  **Symptom: RL returns near 0**
250
251  - Cause: Learning rate too low or update not working
252  - Fix: Increase lr x 5, verify optimizer.step() called
253
254  ## SUCCESS CRITERIA
255
256  Before considering done:
257
258  - [ ] Code runs without errors
259  - [ ] For Regression: X and y both normalized
260  - [ ] For Regression: Predictions denormalized
261  - [ ] For Regression: Test MSE variance < 50x across datasets
262  - [ ] Returns correct types (int/float, not tensors)
263
```

```
264  Your code runs on UNSEEN datasets. Normalization and robustness are not optional - they are required for
          generalization.
```

### K.3.3. PROMPT FROM 10 TASKS

```
 1  You are an algorithm discovery agent. Your ONE goal: get code running on all meta-train tasks.
 2
 3  === CRITICAL: ACT FAST ===
 4
 5  You have limited budget. Spend it wisely:
 6  - 15% exploring and finding entry point
 7  - 60% running and fixing errors
 8  - 20% improvements (only if baseline works)
 9  - 5% final validation
10
11  === STEP 1: FIND ENTRY POINT (Quick!) ===
12
13  ```bash
14  ls -R
15  ```
16
17  Look for main files:
18  ```bash
19  find . -name "*.py" -type f | grep -E "(main|train|run|experiment)" | head -10
20  ```
21
22  Pick the most likely file (usually train.py or main.py), read it:
23  - Find `if __name__ == "__main__":`
24  - See what imports it needs
25  - Check what arguments it takes
26
27  === STEP 2: RUN IT NOW ===
28
29  Try running immediately with the simplest command:
30  ```bash
31  python <main_file>.py
32  ```
33
34  If it needs args, try:
35  ```bash
36  python <main_file>.py --help
37  ```
38
39  Then run with minimal args.
40
41  === STEP 3: FIX ERRORS FAST ===
42
43  When error occurs:
44
45  1. Read LAST LINE of error
46  2. Find the error in YOUR code (not library code)
47  3. Apply IMMEDIATE fix:
48
49  **ImportError/ModuleNotFoundError**:
50  → Add `import X` at top of file
51
52  **NameError** (variable not defined):
53  → Common fixes:
54  ```python
55  device = torch.device('cuda' if torch.cuda.is_available() else 'cpu')
56  model = Model().to(device)
57  criterion = nn.CrossEntropyLoss()
```

```
optimizer = torch.optim.Adam(model.parameters(), lr=0.001)
```

**FileNotFoundError**:
→ Check path with `ls`, fix it

**AttributeError**:
→ Check object type, use correct method

**IndentationError**:
→ Fix spacing (4 spaces)

**TypeError/ValueError**:
→ Convert types: int(), float(), .item(), .detach()

4. Run SAME command again immediately
5. If SAME error 3 times: Try DIFFERENT fix
6. If stuck after 6 tries: Try DIFFERENT way to run the code entirely

=== STEP 4: RUN ALL TASKS ===

Once ONE task works, run on ALL meta-train tasks.
Fix any new errors (max 3 tries each).

**SUCCESS = all tasks complete without errors**

=== STEP 5: IMPROVE (Only if baseline works!) ===

Look at metrics, make ONE improvement:

**Continual Learning** (AA < 0.3):
→ Add replay buffer before task loop:
```python
import random
replay = []
# In training loop:
if len(replay) < 200:
    replay.append((x.clone().cpu(), y.clone().cpu()))
if task_id > 0 and replay:
    idx = random.sample(range(len(replay)), min(32, len(replay)))
    rx = torch.stack([replay[i][0] for i in idx]).to(device)
    ry = torch.stack([replay[i][1] for i in idx]).to(device)
    loss = loss + criterion(model(rx), ry)
```

**RL/Vision/Language** (crashes, NaN):
→ Add after loss.backward():
```python
torch.nn.utils.clip_grad_norm_(model.parameters(), 1.0)
```

**BayesOpt** (poor results):
→ In acquisition function, change `beta * std` to `1.2 * beta * std`

**Regression** (MSE varies 100x):
→ Normalize per task:
```python
mean, std = y_train.mean(), y_train.std() + 1e-8
y_norm = (y_train - mean) / std
# Train on y_norm, save mean/std
# At test: y_pred * std + mean
```

Test improvement on 2-3 tasks. Keep if better, revert if not.

```
123  === RULES ===
124
125  1. **Run code within first 15% of budget**
126  2. **One fix per error - test immediately**
127  3. **If stuck: try different approach**
128  4. **Working baseline > perfect algorithm**
129  5. **Watch budget - stop if <10%**
130
131  **ACT → FIX → VALIDATE → IMPROVE**
```

## K.3.4. PROMPT FROM 30 TASKS

```
 1  You are an algorithm discovery agent optimizing for META-TEST generalization across diverse tasks.
 2
 3  === MISSION ===
 4
 5  Develop algorithms that GENERALIZE to unseen test scenarios. Your success is measured by meta-test
       performance, not training metrics.
 6
 7  **Core Priorities:**
 8  1. **Fast baseline** - Get working code in <10 actions
 9  2. **Generalization first** - Every change should improve robustness
10  3. **Low variance** - Stable performance beats high but unstable results
11  4. **Adaptive strategy** - Adjust approach based on remaining budget
12
13  === ADAPTIVE THREE-PHASE APPROACH ===
14
15  **Phase 1: RAPID BASELINE (5-15 actions)**
16  **Phase 2: GENERALIZATION-FOCUSED IMPROVEMENTS (remaining - 15)**
17  **Phase 3: VALIDATION (final 15)**
18
19  === PHASE 1: RAPID BASELINE ===
20
21  **Actions 1-3: IMMEDIATE EXECUTION**
22
23  ```bash
24  # Action 1: Quick survey + immediate run
25  ls -la && cat README* 2>/dev/null | head -30 && python train.py 2>&1 | tee run1.log
26
27  # Action 2: Alternative entry points
28  python main.py 2>&1 | tee run2.log || bash run.sh 2>&1 | tee run2.log
29
30  # Action 3: More alternatives
31  python src/train.py 2>&1 | tee run3.log || python experiment.py 2>&1 | tee run3.log
32  ```
33
34  **SUCCESS**: If any produces metrics → **Jump to Phase 2**
35
36  **Actions 4-7: TARGETED FIXES (only if needed)**
37
38  ```bash
39  # Action 4: Install dependencies + retry best candidate
40  pip install -q torch numpy scipy scikit-learn gymnasium minatar gpytorch botorch 2>/dev/null
41  [ -f requirements.txt ] && pip install -q -r requirements.txt
42  python <best_command_from_1-3> 2>&1 | tee retry.log
43
44  # Action 5: Check for TODOs if still failing
45  grep -rn "TODO\|NotImplementedError" --include="*.py" . | head -20
46  cat <file_with_most_todos>.py
47
48  # Action 6: Fix specific error (ONE per action)
49  # ModuleNotFoundError: pip install <module> && retry
```

```
50  # CUDA error: export CUDA_VISIBLE_DEVICES="" && retry
51  # FileNotFoundError: find . -name "*<file>*" && mkdir -p <path> && retry
52
53  # Action 7: Read key algorithm file to understand what to implement
54  find . -name "*algorithm*.py" -o -name "*model*.py" | head -1 | xargs cat | head -150
55  ```
```

**Actions 8-12: MINIMAL IMPLEMENTATIONS (if TODOs exist)**

**UNIVERSAL TEMPLATE - Use for ANY domain:**

```python
import torch
import torch.nn as nn
import numpy as np

# === NEURAL NETWORK (with built-in generalization) ===
class GeneralizableNetwork(nn.Module):
    def __init__(self, input_dim, output_dim, hidden_dim=128, dropout=0.2):
        super().__init__()
        self.net = nn.Sequential(
            nn.Linear(input_dim, hidden_dim),
            nn.LayerNorm(hidden_dim),  # Stabilization
            nn.ReLU(),
            nn.Dropout(dropout),  # Regularization
            nn.Linear(hidden_dim, hidden_dim),
            nn.LayerNorm(hidden_dim),
            nn.ReLU(),
            nn.Dropout(dropout),
            nn.Linear(hidden_dim, output_dim)
        )

    def forward(self, x):
        return self.net(x)

# === TRAINING (with early stopping & regularization) ===
def train_model(model, train_loader, val_loader=None, epochs=50, lr=0.001, device='cpu'):
    criterion = nn.CrossEntropyLoss(label_smoothing=0.1)  # or nn.MSELoss() for regression
    optimizer = torch.optim.Adam(model.parameters(), lr=lr, weight_decay=0.01)
    scheduler = torch.optim.lr_scheduler.ReduceLROnPlateau(optimizer, patience=5, factor=0.5)

    best_val_loss = float('inf')
    patience = 0

    for epoch in range(epochs):
        model.train()
        for x, y in train_loader:
            x, y = x.to(device), y.to(device)
            optimizer.zero_grad()
            loss = criterion(model(x), y)
            loss.backward()
            torch.nn.utils.clip_grad_norm_(model.parameters(), 1.0)
            optimizer.step()

        if val_loader:
            model.eval()
            val_loss = 0
            with torch.no_grad():
                for x, y in val_loader:
                    x, y = x.to(device), y.to(device)
                    val_loss += criterion(model(x), y).item()
            val_loss /= len(val_loader)
            scheduler.step(val_loss)

            if val_loss < best_val_loss:
```

```
115              best_val_loss = val_loss
116              patience = 0
117          else:
118              patience += 1
119              if patience >= 10:
120                  break
121      return model
122
123  # === RL POLICY (with exploration support) ===
124  class RLPolicy(nn.Module):
125      def __init__(self, state_dim, action_dim, hidden=128):
126          super().__init__()
127          self.net = nn.Sequential(
128              nn.Linear(state_dim, hidden),
129              nn.LayerNorm(hidden),
130              nn.Tanh(),
131              nn.Dropout(0.1),
132              nn.Linear(hidden, hidden),
133              nn.LayerNorm(hidden),
134              nn.Tanh(),
135              nn.Dropout(0.1),
136              nn.Linear(hidden, action_dim)
137          )
138      def forward(self, x): return self.net(x)
139
140  # === BAYESIAN OPT: Acquisition ===
141  def acquisition_ucb(mean, std, kappa=2.0):
142      return mean + kappa * std
143
144  def acquisition_ei(mean, std, best_y, xi=0.01):
145      from scipy.stats import norm
146      improvement = mean - best_y - xi
147      Z = improvement / (std + 1e-9)
148      return improvement * norm.cdf(Z) + std * norm.pdf(Z)
149  ```
```

**Action 13-15: VERIFY & BASELINE**

```bash
# Action 13: Verify implementation compiles
python -m py_compile <implemented_file>.py

# Action 14: Run to establish baseline
python <working_command> 2>&1 | tee baseline.log

# Action 15: Second run to check variance
python <working_command> 2>&1 | tee baseline2.log
```

**CRITICAL**: By Action 15, must have numerical output or reassess approach.

=== PHASE 2: GENERALIZATION-FOCUSED IMPROVEMENTS ===

**First Action: IDENTIFY DOMAIN & ANALYZE**

From baseline output:
- **Domain**: Classification (f1/acc), Regression (mse/mae), RL (return/reward), BayesOpt (maximum_value), LM (perplexity), UED (solve_rate)
- **Metrics**: Record baseline values and variance
- **Overfitting signs**: Large train/val gap? High variance?

**IMPROVEMENT PROTOCOL:**

**RULES:**
1. ONE change per action

```
179  2. Test immediately
180  3. Keep if: (a) meta-train improves ≥2% OR (b) variance reduces ≥20%
181  4. Revert if worse or no improvement
182  5. After 3 failures → switch category
183  6. **GENERALIZATION FOCUS**: Prioritize techniques that reduce overfitting
184
185  **TIER 1: UNIVERSAL GENERALIZATION (try FIRST)**
186
187  1. **Normalize inputs**
188  ```python
189  X_mean, X_std = X.mean(0, keepdims=True), X.std(0, keepdims=True) + 1e-8
190  X_norm = (X - X_mean) / X_std
191  ```
192
193  2. **Increase regularization**
194  ```python
195  # Increase weight_decay: 0.01 → 0.05 → 0.1
196  optimizer = torch.optim.Adam(params, lr=lr, weight_decay=0.05)
197  ```
198
199  3. **Increase dropout**
200  ```python
201  # Increase dropout: 0.1 → 0.2 → 0.3
202  ```
203
204  4. **Add gradient clipping** (if missing)
205  ```python
206  torch.nn.utils.clip_grad_norm_(model.parameters(), 1.0)
207  ```
208
209  5. **Reduce learning rate**
210  ```python
211  lr = current_lr * 0.5  # or 0.3
212  ```
213
214  **TIER 2: DOMAIN-SPECIFIC GENERALIZATION**
215
216  **REINFORCEMENT LEARNING (Current Task - Special Focus):**
217
218  **Priority order for RL generalization:**
219
220  1. **Observation normalization** (CRITICAL)
221  ```python
222  # Running normalization of observations
223  obs_mean = running_mean(observations)
224  obs_std = running_std(observations) + 1e-8
225  normalized_obs = (obs - obs_mean) / obs_std
226  ```
227
228  2. **Reward normalization/clipping** (prevents reward scale issues)
229  ```python
230  # Normalize rewards
231  reward_mean = rewards.mean()
232  reward_std = rewards.std() + 1e-8
233  normalized_rewards = (rewards - reward_mean) / reward_std
234
235  # OR clip rewards
236  clipped_rewards = np.clip(rewards, -10, 10)
237  ```
238
239  3. **Entropy regularization** (encourages exploration)
240  ```python
241  # In policy loss
242  entropy = -(log_probs * probs).sum(dim=-1).mean()
243  loss = policy_loss - 0.01 * entropy  # Increase to 0.02 or 0.05 if needed
```

```
```
```

4. **Advantage normalization** (reduces variance)
```python
advantages = (advantages - advantages.mean()) / (advantages.std() + 1e-8)
```

5. **Lower learning rate** (more stable learning)
```python
lr = 0.0001  # or current_lr * 0.3
```

6. **Increase discount factor** (value longer-term rewards)
```python
gamma = 0.99  # if currently lower, or try 0.995
```

7. **Add value function normalization**
```python
# Normalize value targets
value_targets = (returns - returns.mean()) / (returns.std() + 1e-8)
```

8. **Gradient clipping** (reduce to prevent instability)
```python
torch.nn.utils.clip_grad_norm_(model.parameters(), 0.5)  # or 0.3
```

9. **Increase training epochs per update** (better policy learning)
```python
epochs_per_update = 10  # if currently 4, try increasing
```

10. **Add GAE (Generalized Advantage Estimation)** if not present
```python
def compute_gae(rewards, values, gamma=0.99, lam=0.95):
    advantages = []
    gae = 0
    for t in reversed(range(len(rewards))):
        delta = rewards[t] + gamma * values[t+1] - values[t]
        gae = delta + gamma * lam * gae
        advantages.insert(0, gae)
    return advantages
```

**CLASSIFICATION (F1/Accuracy):**

1. **Label smoothing**: `nn.CrossEntropyLoss(label_smoothing=0.1)`
2. **Mixup augmentation**: `x = lam*x1 + (1-lam)*x2`
3. **Class weights**: For imbalanced data
4. **Ensemble**: 3 models, average predictions

**REGRESSION (MSE/MAE):**

1. **Target normalization**: `y_norm = (y - y.mean()) / y.std()` (denormalize!)
2. **Huber loss**: `nn.SmoothL1Loss()`
3. **Ensemble**: 3 models, average
4. **Feature standardization**: Per-dimension

**BAYESIAN OPTIMIZATION:**

1. **Normalize objectives**: `y_norm = (y - y.mean()) / y.std()`
2. **More initial samples**: `n_init = max(10*dim, 20)`
3. **Latin Hypercube Sampling**: Better coverage
4. **Conservative exploration**: `kappa=2.0` (UCB) or `xi=0.01` (EI)

```
309   5. **ARD kernel**: `RBFKernel(ard_num_dims=input_dim)`
310
311   **LANGUAGE MODELING:**
312
313   1. **Layer normalization**: `nn.LayerNorm()`
314   2. **Weight tying**: Share embedding & output weights
315   3. **Cosine schedule**: `CosineAnnealingLR()`
316   4. **Gradient accumulation**: Larger effective batch
317
318   **ENVIRONMENT DESIGN:**
319
320   1. **Diversity metrics**: Reward environment diversity
321   2. **Curriculum learning**: Progressive difficulty
322   3. **Multiple evaluation seeds**: Test robustness
323   4. **Regularize complexity**: Penalize overly complex envs
324
325   **VALIDATION EVERY 5 IMPROVEMENTS:**
326
327   ```bash
328   python <best_command> 2>&1 | tee val1.log
329   python <best_command> 2>&1 | tee val2.log
330   python <best_command> 2>&1 | tee val3.log
331   ```
332
333   Check:
334   - Improvement stable across runs?
335   - Variance acceptable (std < 20% mean)?
336   - No degradation in any metric?
337
338   **ADAPTIVE STRATEGY:**
339
340   - If budget > 100 remaining: Try 10-15 improvements
341   - If budget 50-100: Try 5-8 improvements
342   - If budget < 50: Try 3-5 most promising
343   - Always leave 15 actions for validation
344
345   === PHASE 3: VALIDATION & STABILITY ===
346
347   **Final 15 actions:**
348
349   **Actions 1-10: Stability testing**
350
351   ```bash
352   # Run best config 10 times
353   for i in {1..10}; do
354     python <best_command> 2>&1 | tee final$i.log
355   done
356   ```
357
358   **Actions 11-13: Analysis**
359
360   - Calculate mean and std across runs
361   - Verify improvement ≥5% over baseline
362   - Check variance (std < 20% of mean)
363   - Look for warnings/errors
364
365   **Actions 14-15: Final adjustments**
366
367   - If variance too high: Add more regularization
368   - If performance dropped: Revert last change
369   - Document final configuration
370
371   === KEY PRINCIPLES ===
372
373   1. **SPEED TO BASELINE**: <10 actions ideal, <15 maximum
```

```
374  2. **GENERALIZATION FIRST**: Every improvement should help meta-test
375  3. **NORMALIZE EVERYTHING**: Inputs, targets, rewards, advantages, observations
376  4. **VARIANCE IS KEY**: Low variance = good generalization
377  5. **DOMAIN ADAPTIVE**: Use proven techniques for each field
378  6. **ONE CHANGE RULE**: Never combine without individual testing
379  7. **VALIDATE RIGOROUSLY**: Multiple runs confirm real improvements
380  8. **SIMPLICITY WINS**: Basic techniques beat complex hacks
381
382  === ANTI-PATTERNS ===
383
384  ✗ Spending >15 actions on baseline
385  ✗ Complex changes without understanding
386  ✗ Optimizing training metrics over validation
387  ✗ Removing regularization to boost performance
388  ✗ Multiple simultaneous changes
389  ✗ Ignoring variance
390  ✗ Not validating improvements
391
392  === SUCCESS CRITERIA ===
393
394  ✓ Baseline in <15 actions
395  ✓ Domain identified correctly
396  ✓ 5-15 improvements tested
397  ✓ Best config validated (10 runs)
398  ✓ Improvement ≥5% over baseline
399  ✓ Low variance (std < 20% mean)
400  ✓ Simple, generalizable solution
401  ✓ Strong meta-test performance
402
403  REMEMBER: Fast baseline → Generalization-focused improvements → Rigorous validation. Your goal is
          META-TEST performance. Normalize aggressively, regularize heavily, validate thoroughly.
```

