# OpenReview forum: "Procedural Generation Of Algorithm Discovery Tasks in Machine Learning"
_ICML.cc/2026/Conference — ICML 2026 regular_

### Official Review · Reviewer_iseH · 2026-03-06

**Soundness:** 2
**Presentation:** 2
**Significance:** 3
**Originality:** 3
**Overall Recommendation:** 3
**Confidence:** 3

**Summary:**

This paper introduces DiscoGen, a procedural generator for algorithm discovery tasks in machine learning. The authors argue that existing algorithm discovery benchmarks suffer from structural issues, including a lack of proper meta-train/meta-test separation, limited task diversity and scale, risks of data contamination, and reliance on manually curated task suites. To address this, DiscoGen leverages a modular design to generate over 80 million tasks across diverse ML subfields. Each generated task consists of an objective along with meta-train and meta-test datasets, aiming to enable principled evaluation of algorithm discovery agents (ADAs). Based on DiscoGen, the authors construct DiscoBench, a fixed subset of tasks for evaluation. The paper further demonstrates a case study where procedurally generated tasks are used to optimize prompts for an ADA, and discusses potential future research directions enabled by DiscoGen.

**Compliance With Llm Reviewing Policy:**

Affirmed.

**Final Justification:**

While I thank the authors for their response, the method’s formalization and methodological grounding remain insufficient. The evaluation lacks the necessary baselines to isolate the generator's impact, and concerns regarding the benchmark's robustness persist. Given that these core issues require substantial revision, I maintain my score.

**Key Questions For Authors:**

**1. Clarification of "Procedural Generation":** Is DiscoGen purely combinatorial, or does it contain learnable or adaptive components? If there are no learnable parameters, how does DiscoGen differ fundamentally from a large rule-based task enumerator? Beyond the combinatorial explosion of $m$, $d$, and $b$ (modules, datasets, backends), what specific algorithmic or learnable components make DiscoGen "procedural" in the sense of RL environment generators? Is there any adaptive curriculum, difficulty control, or agent-conditioned generation?

**2. Experimental Logic and Alignment:** Since the main contribution of the paper is the procedural task generator (DiscoGen), could the authors clarify why the evaluation primarily focuses on comparing LLM performance on DiscoBench tasks? Are there analyses that directly evaluate the properties of the generated tasks themselves (e.g., diversity, difficulty distribution, or novelty relative to existing task suites), or comparisons with alternative task-generation baselines? Such evaluations would help better isolate the contribution of the generator.

**3. Overfitting Mitigation:** How exactly does DiscoGen mitigate overfitting beyond increasing task count? Could the authors provide empirical evidence that ADAs trained on DiscoGen generalize better to unseen distributions than when trained on existing benchmarks?

**4. Meta-Train / Meta-Test Separation:** What is the precise definition of the meta-train/meta-test separation in DiscoGen? Is the split defined at the dataset level, module level, distribution level, or through held-out generative configurations? Additionally, how large are the meta-train and meta-test sets in terms of the number of tasks? Providing a more formal specification of this split would help clarify the claimed evaluation guarantees.

**5. Benchmark Scope:** Could the authors justify why DiscoBench is limited to a test-set role, and how users should handle the training distribution to ensure fairness across different discovery agents? If DiscoGen is the primary contribution, what is the rationale for framing DiscoBench as a benchmark rather than evaluating directly on held-out regions of the generator’s task distribution?

**Limitations:**

The paper discusses motivations and future directions in the Impact Statement section, but it does not clearly and systematically articulate the limitations of the current generator design. For example, potential limitations could include the reliance on combinatorial construction, the absence of adaptive or learning-based task generation, the possible narrowness of the predefined modules, and the risk of generating trivial or less meaningful task combinations.

**Strengths And Weaknesses:**

## Strengths

**1. Scale and Diversity:** The ability to generate more than 80 million tasks across various domains (RL, Supervised Learning, etc.) is a significant engineering contribution that addresses the data scarcity in algorithm discovery.

**2. Meta-Train/Meta-Test Distinction:** The explicit separation of meta-train and meta-test datasets within generated tasks helps mitigate risks of data contamination and overestimating agent performance.

**3. Modular Design:** The method is implemented in a practical, containerized, file-system-based setup. The use of modular components (optimizers, losses, networks) allows for high flexibility in defining what an agent needs to discover.

## Weaknesses

Despite the interesting high-level idea, there are several substantial concerns regarding soundness, clarity, positioning, and alignment between claims and evidence.

### (A) Soundness

**1. Procedural Generation Formulation.**

While the paper leans heavily on the term "procedural generation" (motivated by RL), the actual implementation appears to be a **combinatorial enumeration** of pre-defined modules and datasets. There is a lack of mathematical formalization; the method section provides only one formula for task counting. It is unclear if there are any learnable parameters or adaptive mechanisms in the generator itself.

**2. Misalignment Between Evaluation and Central Claims.**

The experiments primarily compare the performance of different LLM-based agents on DiscoBench tasks. However, performance differences between models do not directly validate the quality or superiority of the proposed task generation methodology. Since the central contribution of the paper is the procedural generation method (DiscoGen), the evaluation should ideally include comparisons with alternative task-generation strategies or baselines (e.g., manually constructed task suites, simpler combinatorial generators, or existing algorithm discovery benchmarks). Without such comparisons, it is difficult to assess whether the proposed generation process provides meaningful advantages beyond producing a large number of tasks. Furthermore, the "Prompt Optimisation" case study appears more like an engineering demonstration of how the generated tasks can be used, rather than a direct validation of the task generation mechanism itself.

**3. Overfitting Mitigation Without Empirical Evidence.**

The paper argues that large-scale procedural generation helps prevent overfitting and enables open-ended learning. However, there is no rigorous empirical comparison that ADAs trained on DiscoGen generalize better than when trained on existing benchmarks. The connection between scale and generalization is asserted but not convincingly demonstrated.

**4. Ambiguity in Meta-Train/Meta-Test Separation.**

The paper emphasizes the importance of meta-train/meta-test separation to avoid leakage and enable principled evaluation. However, the exact granularity of this separation is not clearly defined. For example, how many tasks are included in the meta-train and meta-test sets, and how are they partitioned (e.g., at the module level, dataset level, backend level, or via held-out generative configurations)? A more formal and precise description of the split would help clarify the claimed evaluation guarantees. In addition, the statement that DiscoBench is not subject to data contamination because the paper has not yet been released (Page 2, Lines 071-072) does not appear to be a methodological safeguard, and further clarification would be helpful.

**5. Narrow Definition of the Benchmark.**

The authors define **DiscoBench** solely as a test set (a subset of DiscoGen). However, a robust benchmark should encompass the entire pipeline (train/val/test). This narrow scope limits the utility of the benchmark for training generalized discovery agents.

### (B) Presentation

**1. Overemphasis on Meta-Learning Terminology.**

The discussion of inner-loop/meta-loop optimization is lengthy and sometimes redundant. It restates known meta-learning concepts in the Introduction section. Meanwhile, the actual mechanics of task generation receive comparatively less formal treatment.

**2. Conceptual Ambiguity of "Procedural Generation."**

The paper repeatedly invokes reinforcement learning and procedural content generation literature (page 1, Line 022-023; page 2, Line 088-092; Page 3, Line 138-143 Related Work; Page 3, Line 137-141), but the analogy is mostly conceptual. Unlike many RL PCG systems, DiscoGen does not appear to adapt task generation based on agent feedback. The conceptual borrowing feels under-justified.

**3. Structural and Editorial Issues.**

The paper frequently introduces complex terminology (e.g., "autocurricula" and "open-ended learning") without providing sufficient technical detail to substantiate these concepts in the context of the proposed method. There are also several notation inconsistencies (e.g., the reuse of the symbol m in Page 5, Lines 236-238 and Page 6, Lines 304-306) and formatting issues in figure references (Page 38, Lines 2041 and 2043). In addition, the overall structure could be improved. Some sections rely heavily on descriptive text rather than rigorous analysis, and related discussions appear multiple times across the paper (e.g., Sections 3, 4.3, and 6), which weakens the clarity of the narrative. Certain sections (e.g., prompt optimization, full task prompts in appendix) occupy disproportionate space relative to the core technical contribution.

### (C) Significance

The problem addressed is important. If successfully realized, a scalable and principled task generator for algorithm discovery could meaningfully impact research on autonomous ML systems.

However, in its current form, the work appears more like a large engineering framework for enumerating task variations rather than a fundamentally new paradigm for algorithm discovery evaluation. The lack of strong empirical validation of the claimed benefits limits its immediate impact.



### (D) Originality

The idea of applying procedural generation concepts to algorithm discovery is interesting and provides a potentially useful perspective for constructing large-scale task suites. However, the generator itself appears to rely primarily on combinatorial construction across predefined modules, datasets, and backends, and the paper does not introduce a new learning-based generative mechanism. In addition, although the work draws conceptual inspiration from procedural generation in reinforcement learning, this connection is not instantiated in the proposed methodology. As a result, the originality of the paper appears to lie more in the application and framing rather than in the introduction of new algorithmic or theoretical developments.

---

> ### Author Rebuttal · Authors · 2026-03-30
>
> Dear iseH,
>
> Thank you for your review, and recognising DiscoGen's **principled design**.
> # Procedural Generation (A1, B2, D, Q1)
> It is important to distinguish between “PCG” (generation) and “autocurricula” (adaptive generation), distinct concepts conflated in your review.
>
> Canonical PCG RL environments (e.g., Nethack, ProcGen and MiniGrid) **do *not* include a learnable component**, as outlined in Togelius et al. PCG in Games; PCG is only defined as *content* developed algorithmically by a manually-constructed engine. PCG is often driven by combinatorics (e.g., combinations of “wall locations”), with the goal of providing a smooth distribution to optimise over for generalisation (L116). Making PCG adaptive/learnable falls under the umbrella of autocurricula/UED, which is **a separate concept to PCG**. Autocurricula is proposed as future research in Section 6.2.
>
> Re: mathematical formalisation, we **derive the size of DiscoGen’s support** in Appendix D, and explore statistical redundancy analysis from L229R. We welcome suggestions for additional formalisation to strengthen the work.
> # Evaluation/Evidence (A2, A3, C, D, Q2, Q3)
> Table 3 shows that **optimising over more random DiscoGen tasks monotonically improves ADA performance in a held-out set of tasks**; direct verification that the generator’s task diversity improves generalisation. We complement Table 3 with analysis of how scale mitigates overfitting in Section H (quantitatively) and Appendix L.3 (prompts).
>
> Furthermore, we *directly* evaluate DiscoGen’s task properties: rank-correlation analysis produces a Fisher-Z mean Spearman over tasks of 0.4, suggesting meaningful signal and low redundancy (L238); Appendix G provides redundancy/diversity analysis over different domains/modules, and justifies the meta-train/meta-test distinction; and Appendix F explores how difficult different module combinations are within the same domain. Collectively, these results are **direct empirical verification of our core claimed benefits**, justifying “the task generation mechanism itself”. Please see our response to catX for further justification.
>
> Empirical comparison with other benchmarks is non-trivial: they do not have distinct ADA optimisation (DiscoGen) and evaluation (DiscoBench) sets, nor are they designed for ADA optimisation. We **qualitatively analyse structural differences between DiscoGen and other task suites in Sections 3 and 4** (e.g., scale, meta-train/meta-test). **We will include a table comparing scale, diversity, success rates (as in Section 5.1) and other properties across different task suites**.
>
> Since **DiscoGen is the first generator of its kind**, a meaningful alternative baseline for task generation is non-obvious. Combined with the above results, DiscoGen is a significant contribution (PCG for algorithm discovery, 80M+ tasks, meta-train/meta-test, empirical verification of ADA optimisation) rather than just an “originality [...] in application and framing”.
> # Definitions and Ambiguity (A4, A5, Q4, Q5)
> Meta-train/meta-test is only about which *datasets* are seen by ADAs during discovery (Fig 1, Sections 1&4). Whereas an ADA can experiment with meta-train datasets, meta-test datasets are held-out to evaluate an algorithm’s performance only. In Example 1, the ADA develops its loss function and architecture in Breakout and Freeway; these are subsequently evaluated in a rebuilt codebase (to prevent evaluation hacking) for SpaceInvaders and Asterix.
>
> Because meta-test datasets are unknown to the agent during discovery, and PCG means each task varies, DiscoGen helps mitigate data contamination. While the paper’s unpublished state ensures the validity of our current analysis, we propose a private DiscoBench API as a longstanding solution (Section 8).
>
> The full system has a (meta-meta-)train/test split: DiscoBench is a set of 47 meta-meta-test tasks for evaluating an ADA optimised in DiscoGen (meta-meta-train). This mirrors PCG environments like ProcGen, where a generator acts as the training set and hand-designed levels act as a held-out evaluation, without a fixed validation suite. We will expand our explanation to clarify this (L103:105).
> # Presentation & Limitations
> We will expand the mechanics of task generation for clarity, and provide a cleaner, documented codebase at: https://anonymous.4open.science/r/discogen-anon-rebut-2380/.
>
> We use meta-learning terminology, grounded in prior work, to establish our 3 levels of optimisation. We will, however, also define the terms you raise.
>
> The **prompt optimisation section is a core contribution**, and the first example of meta-meta-learning (ADA optimisation) for agents to our knowledge; it will be renamed to reflect this.
>
> We will add that modules can be expanded in certain domains to our limitations, but emphasise that DiscoGen **never generates trivial task combinations** (L240 & Appendix D.2). We also couldn’t find the notation/formatting issues; could you please clarify so we can address them?

---

> > ### Author Rebuttal · Reviewer_iseH · 2026-04-03
> >
> > Thank you for the rebuttal. While the authors provide useful clarifications and propose revisions, several key concerns remain only partially addressed. For example:
> >
> > - **Procedural Generation and Novelty.** The rebuttal correctly clarifies that procedural generation (PCG) does not inherently require learnable or adaptive components, and that combinatorial, rule-based generation can be consistent with the PCG paradigm. However, this does not resolve the core concern. The cited PCG literature (Togelius et al., *PCG in Games*) explicitly includes learning-based approaches (PCGML), so the absence of learnable components is not itself sufficient to justify the formulation's depth. More importantly, it remains unclear whether DiscoGen goes beyond a combinatorial enumeration of predefined modules, datasets, and backends. The response does not clearly identify what distinguishes DiscoGen from a basic rule-based task enumerator, nor does it provide evidence that such combinations yield non-trivial or emergent diversity. In addition, the claimed “smooth distribution” for generalization is not clearly substantiated.
> >
> > - **Evaluation and Evidence.** The additional analyses (e.g., scaling experiments and task statistics) show that increasing the number of DiscoGen tasks improves downstream performance. However, the central issue of evaluation misalignment remains. The experiments primarily demonstrate improvements within the same generator distribution, without isolating the contribution of the task generation mechanism itself. In particular, the lack of comparisons with alternative task-generation strategies or simpler baselines (e.g., a manually curated suite of equivalent size or a random-combination baseline) makes it difficult to assess whether the gains arise from the proposed procedural design rather than scale alone. Similarly, the claim of improved generalization is only indirectly supported, without controlled comparisons against existing benchmarks.
> >
> > - **Meta-Train/Test Separation and Benchmark Scope.** The rebuttal clarifies that the meta-train/meta-test split is defined at the dataset level and provides concrete examples, which resolve the ambiguity in the evaluation protocol. The design of DiscoBench as a fixed evaluation suite is also reasonably justified. However, it remains unclear whether the selected evaluation tasks are sufficiently representative of the full task distribution and how selection bias was avoided when curating these specific 47 tasks.
> >
> > As these issues concern the core aspects of the method and its evaluation, they cannot be fully addressed within the rebuttal and would require substantial revision.

---

> > > ### Author Response · Authors · 2026-04-08
> > >
> > > Dear iseH,
> > >
> > > Thank you for your response and the rigour of your engagement. Please see below.
> > > # Procedural Generation
> > > We appreciate your **acknowledgement that DiscoGen is a valid PCG approach**. To clarify: **we *never* suggested DiscoGen was PCGML**, and prior PCG methods (e.g., ProcGen) have been accepted to ICML without that extension.
> > >
> > > As established, DiscoGen is a combinatorial, rule-based generator, and your response agrees that this is "consistent with the PCG paradigm". However, such a system is not necessarily "basic", given the non-triviality of its design and implementation.
> > > # Diversity
> > > Our paper and rebuttals demonstrate that DiscoGen has non-trivial diversity:
> > >
> > > - Fisher-Z transformed mean Spearman Correlation over tasks is ~0.4 (line 238R). There are correlations and anti-correlations for tasks with different modules in the same, and different, domains.
> > > - Appendix E shows how success rates and max achieved performance, aggregated over multiple seeds and models, change when **sweeping over all module combinations** in On-Policy RL. The *max achieved performance* (for 3 models x 3 seeds) is **always** higher for two modules than one, and the success rate decreases as the number of editable modules increase, indicating breadth in discovery.
> > > - Appendix G demonstrates the difference between meta-train and meta-test datasets, showing how datasets contribute diversity.
> > > - Scaling in Section 7 demonstrates that each task contributes non-redundant information.
> > >
> > > "Smooth distribution" is used as in other discrete PCG settings, and is intuitive; there are large changes between very different tasks, but tasks can also be changed incrementally by substituting a dataset or module. This idea of optimising over a "distribution" comes from prior PCG literature (e.g., Kinetix Section 4, Matthews et al., 2025). This forms the crux of our argument: PCG enables optimising an ADA for tasks in DiscoGen’s support, which is very general (modular ML codebase, meta-train (discovery) and meta-test (evaluation) datasets), as shown in Section 7 for unseen DiscoBench tasks. Appendix E **empirically verifies this smoothness**: just changing modules smoothly affects ADA performance.
> > >
> > > The concept of PCG as a distribution is not novel, is well-substantiated in prior literature, and is intuitive to understand. We will clarify this in the paper.
> > > # Evaluation and Evidence
> > > While it is true that performance improvements are shown within the generator distribution, this is **standard practice in prior PCG literature** (e.g., ProcGen, Kinetix, XLand, XLand-MiniGrid, etc.). That said, all datasets, code-bases and modules used in DiscoGen/DiscoBench are standard in each field.
> > >
> > > *Task generation in Section 7 already uses a "random-combination"*. In prior PCG literature, this approach (“domain randomisation”) is commonly used for sampling. It is non-obvious what a "manually curated suite" could be: given DiscoBench is itself manually curated, and we want to optimise an ADA to generalise to new tasks, it would be impossible to select tasks without unintentionally succumbing to selection bias. Random sampling is, therefore, the most simple and sound approach.
> > >
> > > Regarding the suggestion that **it [is] difficult to assess whether the gains arise from the proposed procedural design rather than scale**; the two are intrinsically linked, given that **scale over tasks directly arises from the PCG design of DiscoGen**.
> > > # Representativeness of DiscoBench
> > > Demonstrating that evaluation tasks are representative of a full distribution goes far beyond the expectation of other PCG works; for example, it is unclear whether the 38 registered environments in XLand-MiniGrid are representative of its distribution. However, the 47 DiscoBench tasks covers every (domain,module) pair and an all-module task per domain, which we believe is highly representative of the generator's support.
> > >
> > > Instead, evaluation tasks in PCG are typically considered based on *human-interest*. One reason we selected these extremes of module combinations is that they reflect the two poles of prior algorithm discovery/meta-learning. "Change one" is like conventional meta-learning literature, which learns one component of an algorithm. "Change all" is like more recent work in automated research.
> > > # Selection Bias
> > > We describe how evaluation tasks are selected in line 302:314. **Every single domain and module is represented in DiscoBench**, and curation of the tasks is systematic. We include a task where each module is “editable” and all others are fixed. For every domain, we also include an evaluation task where all modules are "editable". We are upfront about an explicit selection bias in line 312:314 (meta-train datasets are quick-to-run), but besides this, datasets are selected randomly.
> > >
> > > In light of the concessions made in your response, and if you feel we have addressed a number of your concerns, we kindly request that you consider increasing your score.
> > >
> > > Thank you,
> > >
> > > Authors

---

### Official Review · Reviewer_dsmj · 2026-03-08

**Soundness:** 2
**Presentation:** 1
**Significance:** 1
**Originality:** 1
**Overall Recommendation:** 2
**Confidence:** 3

**Summary:**

The current benchmark on algorithm discovery methods is limited because the data contains contamination issues, and the tasks in the benchmark are all similar. To address the issue of data contamination, this paper proposes to procedurally generate tasks to evaluate algorithm discovery systems. The results showed that the existing algorithm discovery methods fall short at the proposed benchmark.

**Compliance With Llm Reviewing Policy:**

Affirmed.

**Ethical Review Flag:**

Flag this paper for an ethics review.

**Key Questions For Authors:**

See weakness

**Limitations:**

Limitations are not addressed. I suggest including the limitations about the scope of datasets and tasks that the procedural generation methods can cover.

**Strengths And Weaknesses:**

- Soundness: The idea of providing a procedurally generated benchmark is good, but because of the unclear presentation, I'm unsure if this paper's proposed method can actually tackle the data contamination issue. Also, the future works mentioned in Section 6 were not grounded on any preliminary results, and the connection between the concept of procedural generation is unclear.
- Presentation:
 - How are the tasks procedurally generated? Section 4.1 defers the implementation details to Appendix D.1 but Appendix D.1 also didn't explain the implementation details well. Procedural generation is the main contribution of this paper, so it is necessary to present the implementation details of the procedural generation process. My understanding now is that the procedural generation is simply done by randomly sampling predefined datasets, tasks, and modules, which doesn't seem to be able to mitigate the data contamination issue since all of them are public and manually defined.
- Significance: The problem is important but at its current form, I'm unsure about the significance of the proposed algorithm
- Originality: The insight on the limitation of current algorithm discovery benchmark is reasonable based on the summary from the related work section and the introduction section, while I'm not sure about the details of the other benchmark.

---

> ### Author Rebuttal · Authors · 2026-03-30
>
> Dear dsmj,
>
> Thank you for your review. We are glad that you **recognise the value of our idea**, and respond below.
>
> # Summary
>
> We would like to clarify our contribution, which was possibly unclear and **may have been mischaracterised**. While we mention diversity and contamination, we outline a number of issues with existing task suites in practice, and the majority of the paper **is dedicated to issues of limited scale and poor evaluation methodology** (i.e., not using meta-train/meta-test). Contamination *resistance* (not solving) is discussed as only one benefit of DiscoGen. Our results also extend to meta-meta learning (ADA Optimisation); we will update the Section 7 title to clarify that this is a key result.
>
> # Presentation
>
> Below, we reiterate the PCG process in DiscoGen. We will expand our PCG explanations in the paper to reflect this.
>
> Section 4.1 provides a high level of what procedural generation relies upon. A task depends on:
> 1. The task domain
> 2. The modules which are/aren’t editable (editable modules are what an agent discovers)
> 3. The meta-train datasets
> 4. The meta-test datasets
> 5. The backend
>
> These are like the “parameters” of procedural generation in a CMDP; like the position of obstacles and goals in a gridworld.
>
> We provide implementation details in Appendix D.1 and through our open-source codebase (which we have polished at https://anonymous.4open.science/r/discogen-anon-rebut-2380/). As in D.1, PCG works as follows for these parameters:
> 1. Parameters are selected manually, randomly or via curriculum (e.g., Example 1). In PCG research, it is **common to randomly sample during training (here: ADA Optimisation) and manually during evaluation**.
> 2. These parameters are passed to make_files() in DiscoGen, which automatically builds a new directory for the task by:
>     - Copying non-module files in the domain’s codebase (e.g., environment wrappers in RL), and evaluation logic.
>     - Loading, respectively, *editable* (empty) or *fixed* versions of each module file.
>     - Writing a description.md file, including details specific to the config.
> 3. An agent discovers performant implementations for the editable modules.
> 4. make_files() is called again, for the meta-test datasets. This deletes non-module code and rebuilds it from scratch to prevent cheating.
>
> Please let us know if we can clarify further.
>
> # Contamination (Presentation/Soundness)
>
> The main objective of procedural generation in DiscoGen is to provide a *smooth distribution over which to optimise* (L115-117 & 138-140, right), to produce generalist ADAs.
>
> That said, PCG in DiscoGen also *partially mitigates* contamination. Since tasks are generated, an agent does not know which datasets its algorithm will be evaluated on. Please note that we **do not claim to *solve* contamination in our paper**. In fact, we claim the opposite: "While DiscoBench, like other benchmarks, is susceptible to data contamination” (L105). We propose a private “API” for DiscoBench in the future, using non-publicised meta-test datasets.
>
> # Future Works
>
> Section 6 is framed as speculative future work (“high level proposals [...] to serve as inspiration”). **All ideas are grounded in the stated advantages of DiscoGen**, and introduced as such, but we will reestablish these justifications in the text.
>
> While not all proposals have preliminary results, which would be uncommon, **each proposal is based on prior literature**. Nevertheless, 6.1, 6.2 and 6.3 are linked to our results (6.1: understanding how ADAs fail in DiscoBench, 6.2: controllable task complexity, 6.3: ADA optimisation in Section 7).
>
> # Significance & Originality
>
> The introduction of PCG to algorithm discovery, and more principled task design, is a **significantly novel contribution**. As demonstrated throughout the paper, these are crucial for enabling *ADA optimisation* (e.g., monotonic improvement with number of tasks), and principled ADA evaluation.
>
> As the review states “**I’m unsure about the significance**”, our “**insight […] is reasonable**” and “**I’m not sure about the details of the other benchmark**”, we are unsure why these sections were scored 1. Nonetheless, we hope we have alleviated your concerns.
>
> # Limitations
>
> We discuss a number of limitations in our paper, including in Section 8 (Future work), discussing contamination risks throughout the paper, and in our impact statement (page 9). Though we raise (L430) that “expanding the number of domains, modules and datasets [...] would enable even more diverse generation”, we will clarify that DiscoGen tasks are not exhaustive of a field and that *some* currently have limited modules.
>
> # Additional Response
>
> This review has flagged our paper for ethics review. We provide a detailed Impact Statement, include AI safety tasks into DiscoGen (ModelUnlearning), and do not believe our work violates the ICML ethics guidelines. Despite this, we are committed to addressing any issues; could the reviewer please clarify their ethical concerns?

---

> > ### Author Rebuttal · Reviewer_dsmj · 2026-04-04
> >
> > Thank you for the response. I think the rebuttal can't fully address my concern. These update should be presented in the paper at submission time.

---

> > > ### Author Response · Authors · 2026-04-08
> > >
> > > Dear dsmj,
> > >
> > > Thank you for your response, which we would like to push back on. Below, we demonstrate that every point made in our rebuttal was already present in the submitted paper. We provide line references to help verification for the reviewer and Area Chair. Please note, these were similarly (though not as extensively) referenced throughout our rebuttal.
> > >
> > > - The explanation of task definitions is in Section 4.1, using the exact language used in our rebuttal.
> > > - The methods for making file systems is described in Appendix D.1. This covers the same set of content as our rebuttal, besides explicitly mentioning the "make_files" file name.
> > > - Our open-source codebase is included as a hyperlink from the abstract (in the pdf).
> > > - Establishing issues in existing task suites is the subject of Section 3.1.
> > > - The fact that contamination resistance (not resolution, see below) is only a small part of the advantages of DiscoGen is covered throughout Section 4.3.
> > > - An example set of parameters is provided in Example 1.
> > > - DiscoGen providing a smooth distribution is stated in line 115:117 & 138R:140R.
> > > - We state that DiscoGen does not solve contamination in line 105:107, line 257:264 (right), line 297, line 319:321, line 439.
> > > - In Section 6, every suggestion is partnered with a description of a design decision/advantage present in DiscoGen.
> > > - The fact that 6.1, 6.2, and 6.3 are linked to our results is established clearly by the context of the results (failure modes on line 325R:338, learning to discover linking to Section 7, difficulty variations discussed on line 314R:317R).
> > > - The novelty of using PCG in algorithm discovery is established throughout the main text, in the title of the paper ("Procedural Generation of Algorithm Discovery Tasks in Machine Learning"), and in line 031R:034R.
> > > - The importance of PCG for ADA optimisation is established over Section 7.
> > > - The limitations of DiscoGen is established throughout the paper, including all the above references for contamination, a future work section in Section 8, and a full-page Impact Statement discussing limitations in Section 9.
> > >
> > > The only updates we suggest in our rebuttal are very minor; adding that the name of the function for procedurally generating a task is called "make_files", and that DiscoGen tasks are not completely exhaustive of all datasets/modules in a field (which is already implied by the counts in Table 1 and full list of modules and datasets in Appendix A). None of these proposed changes affect any of the core methodology, content or results of the paper.
> > >
> > > In light of this information, we would appreciate if the reviewer can acknowledge any remaining concerns they may have following our rebuttal. We would kindly request that, if our rebuttal has alleviated the reviewers issues by highlighting where issues are already covered in the paper, the reviewer considers increasing their overall and section-level scores
> > >
> > > Thank you,
> > >
> > > Authors

---

### Official Review · Reviewer_catX · 2026-03-11

**Soundness:** 3
**Presentation:** 3
**Significance:** 3
**Originality:** 3
**Overall Recommendation:** 4
**Confidence:** 3

**Summary:**

This paper introduces DiscoGen, a novel approach for generating new algorithm discovery tasks in different machine learning fields. The authors also present a corresponding benchmark, named DiscoBench, which is built using this approach. This benchmark allow achieving more effective validation of novel methods for algorithm discovery. It is useful tool and benchmark, however emphirical confirmation of benchmarks efferctiveneess for algorithms comparion is quite limited.

**Compliance With Llm Reviewing Policy:**

Affirmed.

**Final Justification:**

This paper introduces DiscoGen, a novel approach for generating new algorithm discovery tasks in different machine learning fields. The authors also present a corresponding benchmark, named DiscoBench, which is built using this approach.

Strengths:

- The paper proposes a practical approach for generating new tasks to benchmark methods for algorithm discovery.
- It considers multiple machine learning domains, including image classification, reinforcement learning, Bayesian optimization, and others.
- A pipeline for incorporating tasks from new areas is proposed.
- The benchmark is novel and there is not direct analogues.
- The open-source code is available.

Wearnesses:
- Empirical comparison of the practical efficacy of validation of algorithm discovery methods between DiscoBench and other alternative benchmarks.
- DiscoGen is not supplemented with results of evlauation of strong baselines or state-of-the-art (SOTA) methods on generated tasks.
- Lack of comparison with human-designed tasks

While the authors explained the motivation behind these aspects during the rebuttal and answered the main questions, I think the score correctly reflects the impact  and level of implementation of the paper

**Key Questions For Authors:**

- How does the complexity of tasks in DiscoBench correspond to that of human-generated tasks? Are there any experiments in this direction?
- What is the exact number of tasks in the final setup of DiscoBench? Although an analysis is provided in Table 3, the final size of the benchmark remains unclear, making it difficult to estimate the cost of a single run.
- Where the code for DiscoGen located in the repository? I tried to locate it within make_files.py, but it is difficult to separate DiscoGen from DiscoBench in order to use it independently. Alos, the file discobench/eval_code.py is empty. Why?

**Limitations:**

Yes, the limitations and directions for future improvement are discussed properly.

**Strengths And Weaknesses:**

Strengths:

- The paper proposes a practical approach for generating new tasks to benchmark methods for algorithm discovery.
- It considers multiple machine learning domains, including image classification, reinforcement learning, Bayesian optimization, and others.
- A pipeline for incorporating tasks from new areas is proposed.
- The benchmark is novel and there is not direct analogues.
- The open-source code is available.

Wearnesses:
- There is no empirical comparison of the practical efficacy of validation of algorithm discovery methods between DiscoBench and other alternative benchmarks for more narrow fields. There is a lack of empirical confirmation for the claims made in Sections 3.3 and 4.3.
- DiscoGen is not supplemented with results of evlauation of strong baselines or state-of-the-art (SOTA) methods on generated tasks.
Table 2 includes only simple LLM baselines evaluated on the benchmark. If the main impact of the paper is the benchmark itself, then SOTA solutions for algorithm discovery (such as AlphaEvolve/OpenEvolve, noted in Section 2.1, or other appropriate tools) should also be evaluated on it.

---

> ### Author Rebuttal · Authors · 2026-03-30
>
> Dear catX,
>
> Thank you for your review. We appreciate your recognition of **novelty in DiscoGen**, its **breadth in coverage** and its **practicality**. We respond below:
> # Lack of Confirmation for Claims
> We would like to clarify how **all claims in Section 3/4 are well justified**. Please see our response to iseH for further justification:
> - **Principled evaluation**: We show that only considering meta-train leads to flawed analysis and inflated claims about an algorithm's quality (Table 2, 3, 5, Appendix G).
> - **Diversity**: We explore diversity in Appendix G (note, we have corrected an incorrect ref on L357), and justify claims with citations in the paper. We will also cite the new AIRSBench (Lupidi et al. 2026), which also makes this claim (and is itself limited).
> - **Initialisation**: This is defended by prior literature, both about LLMs (generally) and algorithm discovery research (MLGym). To enable further analysis, we have now added the ability for DiscoGen tasks to start from *either* empty *or* baseline implementations.
> - **Easy Expansion**: The size of DiscoGen’s support demonstrates this. In fact, to expand DiscoGen after submission, we have added a small number of extra modules to OnPolicyRL, which has quadrupled its support, as well as adding four more domains (e.g., Driving Trajectory Prediction and Offline RL).
> - **Problem Saturation**: We justify this with prior literature, though will reference MLGym as it includes known-to-be-saturated tasks. We verify DiscoGen is not saturated (Section 5), and that tasks can be made easier or harder (Appendix E).
> - **Data Contamination**: Intuitively, we argue that combining PCG with a held out meta-test set reduces risks of contamination. However, we take care to highlight that it is still an issue of DiscoBench (static), and propose mitigations in Section 8.
> # Comparison With Other Benchmarks
> All design decisions are motivated empirically or through prior work, as above. However, empirical comparisons between DiscoGen/DiscoBench and other task suites is non-trivial given ours are designed for a different end (meta-meta-optimisation). We do compare DiscoBench success rates to MLGymBench (line 302), and similarly find that agents underperform human baselines. Given the extra space afforded by camera ready, we **will include a table to expand our comparison between DiscoGen and other task suites in the main text**.
> # Missing Strong Baselines
> We see our primary contribution in this work as enabling *procedural generation* (DiscoGen) for ADA optimisation; we will retitle Section 7 to clarify it is a **core contribution** showing ADA optimisation. DiscoBench is one component of this, enabling ADA evaluation like specific mazes in MiniGrid.
>
> While extra baselines in DiscoBench would benefit our evaluation, testing more complex (and thus, expensive) methods is beyond our means as an academic lab. For context, our MLGym agents took 80-100 “actions”, corresponding to \~10 algorithm proposals. This costs \~\\$0.50 per task with open-source models and c. \\$100 for 24 (rented) H200 hours. Evolutionary methods use *significantly more attempts*; ShinkaEvolve makes 80-500 proposals and costs between \~\\$6-\\$40, according to the paper; 100 times more expensive! Given our evaluation also needs much more compute (\~\\$800/task for 80 proposals), this analysis is prohibitive. We will discuss this in the paper as a limitation and future work, but still note that **our current analysis demonstrates that DiscoBench is discriminative between agents**.
>
> We considered testing closed-source LLMs. Besides reproducibility issues, Claude 4.5 Sonnet is \~40x more expensive/token than Deepseek V3.2; again, prohibitively costly.  However, we have verified that Claude Code *can successfully complete* DiscoGen tasks; since DiscoGen *only* builds file systems, **it is strictly agent-agnostic**.
> # Questions
> **Task Complexity**: Tasks in DiscoBench are human-*designed*, like mazes in a gridworld. DiscoBench tasks have fixed structure (one module at a time, or all modules) for fixed meta-train/meta-test splits. In fact, all tasks in DiscoGen are somewhat “human” as they are built by our manually designed engine (as discussed in lines 465 and 475 of our Impact Statement).
>
> **DiscoBench Size**: The total number of DiscoBench tasks is $\sum_{domains} m_{d}+N_{domains}$. For the current iteration of DiscoGen, this gives 47 (Model Unlearning only has 1 module, so its *DiscoBench Single* and *DiscoBench All* tasks are the same). We will clarify this in the paper.
>
> **Code**: We apologise about confusion in the codebase; DiscoGen's logic is in make_files.py. While we originally uploaded the *exact* code used in our experiments, we provide a cleaner, better documented repository here: **https://anonymous.4open.science/r/discogen-anon-rebut-2380/**. This code has been *slightly* expanded since submission.
>
> eval_code.py was empty and accidentally left in the old code. It is not in the new codebase.

---

> > ### Author Rebuttal · Reviewer_catX · 2026-03-31
> >
> > Thank you for providing clarifications and improving the open-source repository.
> >
> > I agree with the arguments about the computational cost of additional experiments. However, in my opinion, it is quite important for a benchmark to provide metrics for SOTA solutions that can be used for future comparisons by the benchmark's users. This also raises questions about the real-world applicability of the benchmark, since evaluating state-of-the-art candidates on it can be too expensive.
> >
> > Additionally, the argument that "tasks in DiscoBench are human-designed, like mazes in a gridworld" is not very convincing. While the tasks are "based on human-selected codebases," it is still not clear whether they are comparable to benchmark tasks designed by humans from scratch.
> >
> > So, my score remains the same, since I think it is appropriate for the paper.

---

> > > ### Author Response · Authors · 2026-04-08
> > >
> > > Dear catX,
> > >
> > > Thank you for your engagement in our rebuttal, and we appreciate your **positive recommendation for the paper**. We have highlighted a couple of points below which may help clarify your remaining concerns.
> > >
> > > # Missing Baselines
> > >
> > > While evaluating SOTA solutions in DiscoBench would help us better understand their current capabilities, we would like to reemphasise that the core contribution of the work lies not in benchmarking current agents, but in providing a **procedural generator of tasks which can be used for optimising ADAs**. As stated, we will retitle Section 7 to reemphasise that ADA optimisation via PCG is a **core contribution** rather than just a case study.
> > >
> > > As such, we feel it worth highlighting that other PCG papers generally follow a similar structure: evaluate a commonly used (but unlikely to be SOTA) algorithm to demonstrate a proof-of-concept. For example, many PCG papers only evaluate with simple RL algorithms:
> > > - ProcGen - PPO and Rainbow
> > > - Xland-Minigrid - Recurrent PPO
> > > - Kinetix - PPO only
> > >
> > > # High Cost
> > >
> > > High costs are a significant limitation across the field of agentic research (we discuss this in line 488R:493R). As such, we wanted to briefly mention the baselines used in earlier ADA benchmarks (note - these papers do not focus on PCG/ADA optimisation, and their *only* contribution is in benchmarking current agents). As a tree-search agent, AIDE is not evolutionary (and is not always better than a ReAct agent in the examples below). We will clarify evaluating tree-search agents as a potential future research direction in our Future Work.
> > > - MLAgentBench: Simple ReAct agent - Academia
> > > - MLEBench: AIDE and ReAct - OpenAI
> > > - RE-Bench: AIDE and ReAct - METR
> > > - MLGymBench: The same agent we used (ReAct agent) - Meta
> > > - AIRS-Bench: One shot and a greedy tree-search agent - Meta
> > >
> > > By releasing DiscoGen, one hope is that the cost of this research can be brought down significantly by, for example, enabling ADA optimisation with smaller models. We also verified that DiscoBench tasks can be run on different types of hardware, in Appendix C.4 (e.g., 2080Ti, 3080 and L40s).
> > >
> > > It is worth acknowledging that evolutionary baselines, while effective in easy-to-evaluate problems (e.g., those seen in AlphaEvolve), are somewhat at odds with research in algorithm discovery for some machine learning domains. For example, a system which needs to evaluate hundreds or thousands of loss functions to discover a SOTA algorithm for training an LLM will be impractical, no matter the available resources. This represents a bifurcation of promising research directions for different fields, and we will highlight this in the paper.
> > >
> > > # Human Tasks
> > >
> > > We're sorry that we didn't clear this up fully. Tasks are not *just* based on human-selected codebases; they are, genuinely, human-designed. We manually selected the "DiscoBench Single" and "DiscoBench all" settings as extremes which are most "human-like", based on prior literature. As below, these are exactly the same kinds of problems that are used in non-procedurally generated automated discovery/meta-learning research. We do mention that our failure rates are similar to a human-designed task suite (MLGymBench (line 301R)), but will make this more concrete.
> > >
> > > "DiscoBench Single" mimics meta-learning problems where one component of an algorithm is replaced by a learned one; for example, DiscoPoP (Lu et al., 2024) asks an LLM to optimise a loss function in some dataset, and the How Should We Meta-Learn RL algorithms (Goldie et al., 2025) uses hand-designed problems set up like the On-PolicyRL tasks in DiscoBench (for an example from that paper: discover an optimiser given its input/output interface in optim.py, using one or more RL environments to help discovery, and evaluating the algorithm in other RL environments). "DiscoBench all" reflects the structure of tasks seen in agent-based automated research, where an agent is tasked with editing nearly all components of an algorithm (though, sometimes, agents start from a full implementation); besides the meta-train/meta-test distinction, this is similar to MLGymBench or MLE-Bench.
> > >
> > > ---
> > >
> > > We hope we have helped to clarify some additional points, and thank you again for your engagement throughout the review process.
> > >
> > > Thank you,
> > >
> > > Authors

---

### Decision · Program_Chairs · 2026-04-30

**Decision:**

Accept (regular)

**Comment:**

Disclaimer: As requested by the authors, we carefully considered their feedback and have invited an additional reviewer to evaluate this paper. This new review (copied below), together with the AC's own evaluation and the current reviews, form the core content of this meta-review:

Summary: In this paper, the authors focus on designing DiscoGen, a procedural framework for generating large-scale algorithm-discovery tasks in machine learning. The new framework aims to address limitations and challenges in existing benchmarks, including  limited diversity, data contamination, and flawed evaluation protocols.

The paper studies an important concept: How to systematically scale and diversify tasks for training and evaluating algorithm-discovery agents (ADAs). It also introduces DiscoBench, a set of tasks for evaluating ADAs and providing a controlled evaluation suite, although it inherits some limitations of fixed benchmarks.

The authors did a good job addressing the main concerns of the reviewers, while some are still left open. However, the overall evaluation is that this paper has its merits and it's worth presenting them at the conference.

## The emergency review

In this paper, the authors focus on designing DiscoGen, a procedural framework for generating large-scale algorithm-discovery tasks in machine learning. The new framework aims to address limitations and challenges in existing benchmarks, including  limited diversity, data contamination, and flawed evaluation protocols.

The paper studies an important concept: how to systematically scale and diversify tasks for training and evaluating algorithm-discovery agents (ADAs). It also introduces DiscoBench, a set of tasks for evaluating ADAs and providing a controlled evaluation suite, although it inherits some limitations of fixed benchmarks.

Related to the strengths of the paper:

-  The paper proposes a novel, scalable framework (DiscoGen) for generating algorithm-discovery tasks that addresses key limitations of prior benchmarks.

-  It introduces a principled evaluation design with explicit meta-train/meta-test separation, which improves experimental rigor and reduces risks of overfitting and data contamination.

-  The framework demonstrates high flexibility and extensibility, supporting multiple ML domains and millions of task configurations, enabling broader research exploration.

-  It also provides details on experimental designs, implementing DiscoGen and evaluating on DiscoBench, and all relevant experimental results.

Related to the weaknesses of the paper:

-  While DiscoBench is useful, there some concerns that it may still inherit benchmark-related issues (e.g., potential contamination or limited representativeness), reducing its long-term robustness.

-  It would be great if the paper could show a clear superiority of ADAs trained with DiscoGen over existing methods.
Despite these concerns, the work opens promising research directions in automated ML and agent-based algorithm design, particularly in exploring generalization and open-ended learning.

Recommendation: Weak Accept